# A model of human neural networks reveals NPTX2 pathology in ALS and FTLD

Marian Hruska-Plochan[1], Vera I. Wiersma[1,13], Katharina M. Betz[1,2,3,13], Izaskun Mallona[1,2,3,13], Silvia Ronchi[4,12,13], Zuzanna Maniecka[1], Eva-Maria Hock[1], Elena Tantardini[1], Florent Laferriere[1], Sonu Sahadevan[1], Vanessa Hoop[2], Igor Delvendahl[2], Manuela Pérez-Berlanga[1], Beatrice Gatta[1], Martina Panatta[1], Alexander van der Bourg[5], Dasa Bohaciakova[6], Puneet Sharma[7,8], Laura De Vos[1], Karl Frontzek[9], Adriano Aguzzi[9], Tammaryn Lashley[10,11], Mark D. Robinson[2,3], Theofanis Karayannis[5], Martin Mueller[2], Andreas Hierlemann[4] & Magdalini Polymenidou[1✉]

Human cellular models of neurodegeneration require reproducibility and longevity, which is necessary for simulating age-dependent diseases. Such systems are particularly needed for TDP-43 proteinopathies[1], which involve human-specific mechanisms[2–5] that cannot be directly studied in animal models. Here, to explore the emergence and consequences of TDP-43 pathologies, we generated induced pluripotent stem cell-derived, colony morphology neural stem cells (iCoMoNSCs) via manual selection of neural precursors[6]. Single-cell transcriptomics and comparison to independent neural stem cells[7] showed that iCoMoNSCs are uniquely homogenous and self-renewing. Differentiated iCoMoNSCs formed a self-organized multicellular system consisting of synaptically connected and electrophysiologically active neurons, which matured into long-lived functional networks (which we designate iNets). Neuronal and glial maturation in iNets was similar to that of cortical organoids[8]. Overexpression of wild-type TDP-43 in a minority of neurons within iNets led to progressive fragmentation and aggregation of the protein, resulting in a partial loss of function and neurotoxicity. Single-cell transcriptomics revealed a novel set of misregulated RNA targets in TDP-43-overexpressing neurons and in patients with TDP-43 proteinopathies exhibiting a loss of nuclear TDP-43. The strongest misregulated target encoded the synaptic protein NPTX2, the levels of which are controlled by TDP-43 binding on its 3′ untranslated region. When NPTX2 was overexpressed in iNets, it exhibited neurotoxicity, whereas correcting NPTX2 misregulation partially rescued neurons from TDP-43-induced neurodegeneration. Notably, NPTX2 was consistently misaccumulated in neurons from patients with amyotrophic lateral sclerosis and frontotemporal lobar degeneration with TDP-43 pathology. Our work directly links TDP-43 misregulation and NPTX2 accumulation, thereby revealing a TDP-43-dependent pathway of neurotoxicity.

TDP-43 protein accumulates in affected neurons from patients with neurodegenerative diseases, including amyotrophic lateral sclerosis (ALS) and frontotemporal lobar dementia[1,9] (FTLD). TDP-43 is an essential RNA-binding protein[10] that is tightly autoregulated via binding to its own mRNA[11,12]. In normal cells, TDP-43 is predominantly nuclear and directly controls the processing of hundreds of its RNA targets[11,13]. Conversely, TDP-43 forms pathological aggregates in disease, which are neurotoxic per se, featuring a potency that correlates with disease duration in patients with FTLD[14–16]. Moreover, the aggregates trap newly synthesized TDP-43, leading to nuclear clearance and loss of its normal function[1,17,18]. This has detrimental consequences, as it leads to broad splicing misregulation[11,19], including the inclusion of cryptic exons in specific TDP-43 RNA targets[20] such as STMN2[3,5] and UNC13A[2,4]. Both of these RNA targets are neuronal and human-specific, and their levels were found to be significantly reduced in brains of patients with TDP-43

[1]Department of Quantitative Biomedicine, University of Zurich, Zurich, Switzerland. [2]Department of Molecular Life Sciences, University of Zurich, Zurich, Switzerland. [3]SIB Swiss Institute of Bioinformatics, University of Zurich, Zurich, Switzerland. [4]Department of Biosystems Science and Engineering, ETH Zürich, Basel, Switzerland. [5]Brain Research Institute, University of Zurich, Zurich, Switzerland. [6]Department of Histology and Embryology, Faculty of Medicine, Masaryk University Brno, Brno, Czech Republic. [7]Department of Chemistry, Biochemistry and Pharmaceutical Sciences, University of Bern, Bern, Switzerland. [8]NCCR RNA and Disease Technology Platform, Bern, Switzerland. [9]Institute of Neuropathology, University of Zurich, Zurich, Switzerland. [10]Queen Square Brain Bank for Neurological diseases, Department of Movement Disorders, UCL Institute of Neurology, London, UK. [11]Department of Neurodegenerative Disease, UCL Institute of Neurology, London, UK. [12]Present address: MaxWell Biosystems AG, Zurich, Switzerland. [13]These authors contributed equally: Vera I. Wiersma, Katharina M. Betz, Izaskun Mallona, Silvia Ronchi. ✉e-mail: magdalini.polymenidou@uzh.ch

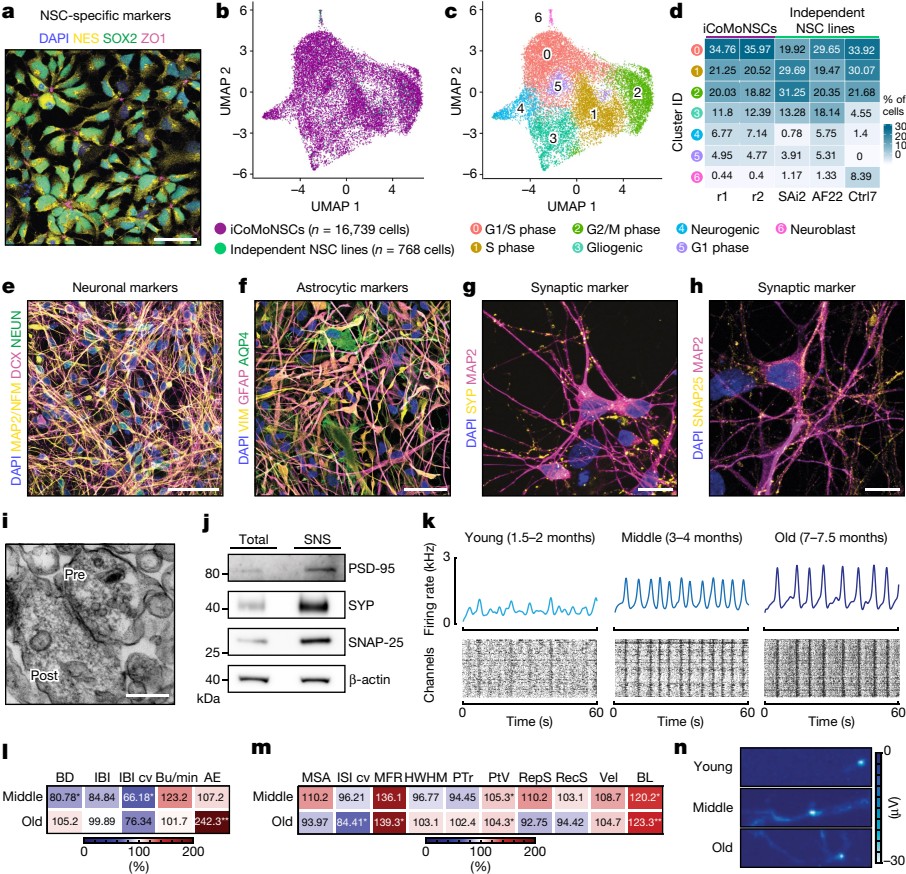

**Fig. 1 | iCoMoNSC neurons form functional networks. a**, Immunofluorescence detection of NSC markers in iCoMoNSCs (representative image from two repeats). **b**,**c**, Uniform manifold approximation and projection (UMAP) of iCoMoNSCs integrated with cells from three independent NSC lines (**b**) and corresponding annotated clusters (**c**). **d**, Percentage of cell distribution per sample across clusters. r1 and r2 indicate replicates 1 and 2, respectively. **e**,**f**, Immunofluorescence of 1.5-month-old human neural cultures stained with neuronal (**e**) or astrocytic (**f**) markers. Synaptic marker immunofluorescence at 3 months for SYP (**g**) and SNAP-25 (**h**). **e**–**h**, Representative images from two repeats. **i** Electron microscopy of SNS, showing pre- and postsynaptic compartments. **j**, western blot of SNS fractions. **i**,**j**, 3-month-old culture. One experiment. **k**, Population spike time histograms (top) and raster plots (bottom) of human neurons (representative data visualization). **l**,**m**, Heat maps of percentage changes in network (**l**) or single-cell and subcellular (**m**) metrics. Number of chips used: 10 (middle stage) and 10 (old), normalized to 7 (young). AE, active electrodes; BD, burst duration; BL, branch length; Bu/min, bursts per minute; HWHM, half width half maximum; IBI, inter-burst interval; IBI cv, IBI coefficient of variation; ISI cv, inter-spike interval coefficient of variation; MFR, mean firing rate; MSA, mean spike amplitude; PTr, peak-to-trough ratio; PtV, peak-to-valley; RecS, recovery slope; RepS, repolarization slope; Vel, velocity. **n**, HD-MEA electrical image or 'footprint' showing the 2D spatial distribution maps of the electrical activity in terms of action potential amplitude (voltage) of three neurons from young (top), middle stage (center) and old (bottom) cultures. (representative data visualization of branch length in **m**). Colour scale represents action potential amplitude. Scale bars: 50 μm (**a**,**e**,**f**), 10 μm (**g**,**h**) and 250 nm (**i**).

proteinopathies, directly linking the loss of TDP-43 nuclear function to neurodegeneration.

The recognition of *STMN2* and *UNC13A* motivated the development of fully human experimental models for TDP-43 proteinopathies to decipher the key targets and downstream pathological mechanisms of TDP-43 misregulation. Induced pluripotent stem (iPS) cell-based systems offer this possibility, and several breakthroughs have been made in recent years with this technology, including the generation and characterization of numerous iPS cell lines from patients with ALS and FTLD (by the Answer ALS project), the recognition of early neuronal phenotypes in human neurons with ALS-linked mutations[21,22] and disease-linked transcriptomic signatures[23,24]. Nonetheless, most studies with iPS cell-derived neurons from patients with TDP-43 proteinopathies have reported low to no TDP-43 pathology[25,26], potentially owing to the early maturation state of human neurons in culture.

## iCoMoNSCs are uniquely homogeneous

We generated a self-renewing human neural stem cell line (iCoMoN-SCs), via manual selection on the basis of their colony morphology[6], from induced pluripotent stem cells (iPS cells), which we derived from normal human skin fibroblasts through episomal reprogramming (Extended Data Fig. 1a). iCoMoNSCs were stable across at least 24 passages, retaining their characteristic radial morphology in cell clusters and apparently normal karyotype, as well as expression of neural stem cell (NSC)-specific markers (Fig. 1a and Extended Data Fig. 1b–f). To determine the homogeneity of the iCoMoNSCs, we performed single-cell RNA sequencing (scRNA-seq) of two replicates at passage 22. Pre-processing, quality control and filtering yielded more than 8,300 cells per replicate with a median number of around 2,000 detected genes and 4,800 unique molecular identifiers (UMIs) (Extended Data Fig. 1g,h), which were separated in tightly associated clusters (Extended Data Fig. 1i), defined mostly by cell cycle stage (Extended Data Fig. 1j) and comprising cells from both replicates (Extended Data Fig. 1k,l), demonstrating that the iCoMoNSCs were extremely homogeneous. The classical NSC marker genes *NES*, *SOX2*, *NR2F1* and *CDH2*, as well as *IRX2* and *SOX1*, were expressed in a subset of cells from all clusters with similar levels (Extended Data Fig. 1m), suggesting that the majority of cells were true, self-renewing NSCs at different cell cycle stages. We then identified cluster marker genes (Supplementary Fig. 1a) and

interrogated the expression of sets of known cell type marker genes (Supplementary Fig. 1b). This showed that most of the cells were in G1/S phase (clusters 0, 1 and 5), followed by approximately 19% of cells (cluster 2) that were marked by classical cell cycle-associated genes (G2/M). For a minority of the cells, markers indicated lower multipotent states with either gliogenic (around 13% of cells, cluster 3) or neurogenic (approximately 8% of cells, cluster 4) nature. Finally, for a very small percentage of cells (approximately 0.3%, cluster 6) the expression of neuron-specific genes and cluster markers demonstrated their committed neuroblast nature (Extended Data Fig. 1i,l and Supplementary Fig. 1a,b). Together, these data showed that up to 79% of cells among the iCoMoNSCs were true, self-renewing NSCs.

We then integrated our data with previously published scRNA-seq datasets from independent human NSCs[7], which were distributed amongst our iCoMoNSCs in all clusters (Fig. 1b–d and Supplementary Fig. 1c). This enabled us to refine our cluster annotation (Fig. 1c). Next, we compared the cluster abundances of all samples individually and found that the iCoMoNSCs and the iPS cell-derived AF22 and Ctrl7 lines had the most similar cell distributions, whereas the SAi2 line that was derived from human fetal hindbrain primary NSCs showed slightly different cell cycle distribution (Fig. 1c,d). Despite the similarities with the independent NSC lines, iCoMoNSCs contained significantly fewer (between 3 and 20 times) committed neuroblasts, represented in cluster 6 (Fig. 1d), indicating that the iCoMoNSCs consisted primarily of self-renewing NSCs.

## iCoMoNSC neurons form functional networks

Upon differentiation (Extended Data Fig. 2a), the iCoMoNSCs consistently generated mixed neuronal and glial multilayer cultures[27,28] (Fig. 1e,f). After 1.5 months of differentiation, these cultures consisted of approximately 30% NEUN+ neurons, later stabilizing at around 35% (Fig. 1e and Extended Data Fig. 2b,c). By contrast, the number of Ki67+ proliferating cells decreased from nearly 100% in iCoMoNSCs (Extended Data Fig. 2d) to around 5% (Extended Data Fig. 2e). To investigate the presence of synaptic markers, we first immunolabelled 3-month-old cultures for SYP (a synaptic vesicle marker; Fig. 1g) and SNAP-25 (a protein of the synaptic vesicle fusion machinery; Fig. 1h) and found a typical punctate pattern. Then, using 3-month-old cultures, we prepared synaptoneurosomes[29] (SNSs)—a subcellular preparation enriched in resealed presynaptic and postsynaptic structures—and analysed them by transmission electron microscopy (TEM), which revealed typical synaptic morphology, consisting of both presynaptic vesicles and postsynaptic densities (Fig. 1i). Immunoblots of total lysates and SNS fractions confirmed the enrichment of synaptic markers (Fig. 1j).

To assess the functionality of these synapses, we first performed in vitro two-photon calcium imaging after bolus loading of 3-month-old cultures with the membrane-permeable ester form of the calcium indicator Oregon Green BAPTA-1. Calcium transients of recorded neuronal somata demonstrated that the cultures indeed displayed sparse spontaneous activity patterns (Extended Data Fig. 2f). To formally confirm neuronal activity in our cultures, we assessed their electrophysiological properties via whole-cell patch-clamp measurements. We observed that 3.5-month-old patched neurons (Extended Data Fig. 2g) had a hyperpolarized resting membrane potential ($-59.7 \pm 4.3$ mV, $n = 7$) and fired single or multiple action potentials upon depolarizing current injection (10 out of 11 neurons; Extended Data Fig. 2h,i). Voltage-clamp recordings showed rapidly inactivating inward currents and slowly inactivating outward currents, typical for Na+ and K+ currents, respectively (peak Na+ current density: $-86.7 \pm 20.5$ pA pF$^{-1}$, peak K+ current density: $149.4 \pm 28.0$ pA pF$^{-1}$, $n = 10$; Extended Data Fig. 2j,k). Collectively, these analyses demonstrated that iCoMoNSC-derived neurons contained voltage-dependent channels and were electrophysiologically active with a hyperpolarized resting membrane potential.

To investigate whether iCoMoNSC-derived neurons were interconnected and displayed coordinated activity, we used high-density microelectrode arrays[30,31] (HD-MEAs; Extended Data Fig. 3b). Neural cultures were plated onto the HD-MEAs after approximately 1, 3 and 6 months of differentiation and were then allowed to reconnect for 1 month before recording to compare cultures denoted here as young (1.5–2 months), middle stage (3–4 months) and old (7–7.5 months) (Extended Data Fig. 3a). Young cultures exhibited a lower burst activity than middle stage and old cultures (Fig. 1k). We then analysed burst metrics[32] (Fig. 1l and Extended Data Fig. 3c,f–j) and found a significant decrease in burst duration between young and middle stage cultures ($P < 0.025$), as well as a decrease in the inter-burst interval coefficient of variation ($P < 0.025$) (Fig. 1l and Extended Data Fig. 3f,h). The higher inter-burst interval coefficient of variation in young cultures compared with middle stage (Fig. 1l and Extended Data Fig. 3h) suggested that early maturation stages were characterized by irregular bursts. Additionally, for 30% of the young cultures a network analysis could not be conducted as bursts were undetectable, which is indicative of still-developing synaptic connections. By contrast, older cultures showed detectable bursts in the majority of HD-MEAs (over 90%). Additionally, the percentage of active electrodes increased 2.4-fold ($P < 0.001$) from young to old cultures (Fig. 1l and Extended Data Fig. 3j).

Single-cell and subcellular-resolution metrics[33] (Fig. 1m and Extended Data Fig. 3d, e, k–t) showed a 1.2-fold decrease ($P < 0.025$) in inter-spike interval coefficient of variation[32] between young and old cultures (Fig. 1m and Extended Data Fig. 3l), indicating more regular firing rates at later developmental stages. In addition, a 0.7-fold lower mean firing rate in young versus old cultures demonstrated an increase in spontaneous activity over time (Fig. 1m and Extended Data Fig. 3m). Longer-lasting action potential recovery times were evidenced by the peak-to-valley ratio metric, which increased from young to middle stage ($P < 0.025$) and from young to old cultures ($P < 0.025$) (Fig. 1m and Extended Data Fig. 3p).

We also analysed the subcellular-resolution features branch length and action potential propagation velocity. A significant increase was found in the neuron branch length between young and middle stage ($P < 0.025$) and between young and old cultures ($P < 0.001$), indicating different functional maturation stages (Fig. 1m,n and Extended Data Fig. 3t). Principal component analysis showed a separation of all three maturation stages based on all 15 analysed parameters (Extended Data Fig. 3u). Together, these functional metrics indicated increased maturation upon ageing of iCoMoNSC-derived neurons in culture with the development of functional neuronal networks. We subsequently refer to these iCoMoNSC-derived cultures as iNets.

## iNets transcriptionally mimic organoids

To further characterize these cultures, we performed scRNA-seq of young, middle stage and old iNets (Extended Data Fig. 3a) in duplicates. After processing, we retained 3,500–8,500 cells per sample with 2,800–4,800 detected genes and 6,000–16,000 UMIs. Using the similarity in their transcriptomes, we grouped cells into 19 clusters or cell identities, which we annotated using their differentially expressed genes as neurons, astrocytes and other glial cells (Fig. 2a). Neuronal and astrocytic clusters showed increasing maturation over time in differentiation, which was evident upon visualization of all experimental (Fig. 2b and Supplementary Fig. 2a) or predicted cell cycle (Fig. 2c) stages. Cluster abundance analysis (Supplementary Fig. 2b,c) and the visualization of individual, cell type-specific markers revealed that our cultures matured over time: NSC-specific markers such as *SOX2* and *NQO1* exhibited high expression in iCoMoNSCs (Supplementary Fig. 2d). Astrocyte maturation was highlighted by *GFAP* (astrocyte marker), which was expressed in middle stage and old neural cultures. *PTPRZ1* (an oligodendrocyte progenitor cell (OPC) marker) was detected in all samples and *DCN* (a pericyte marker) was detected

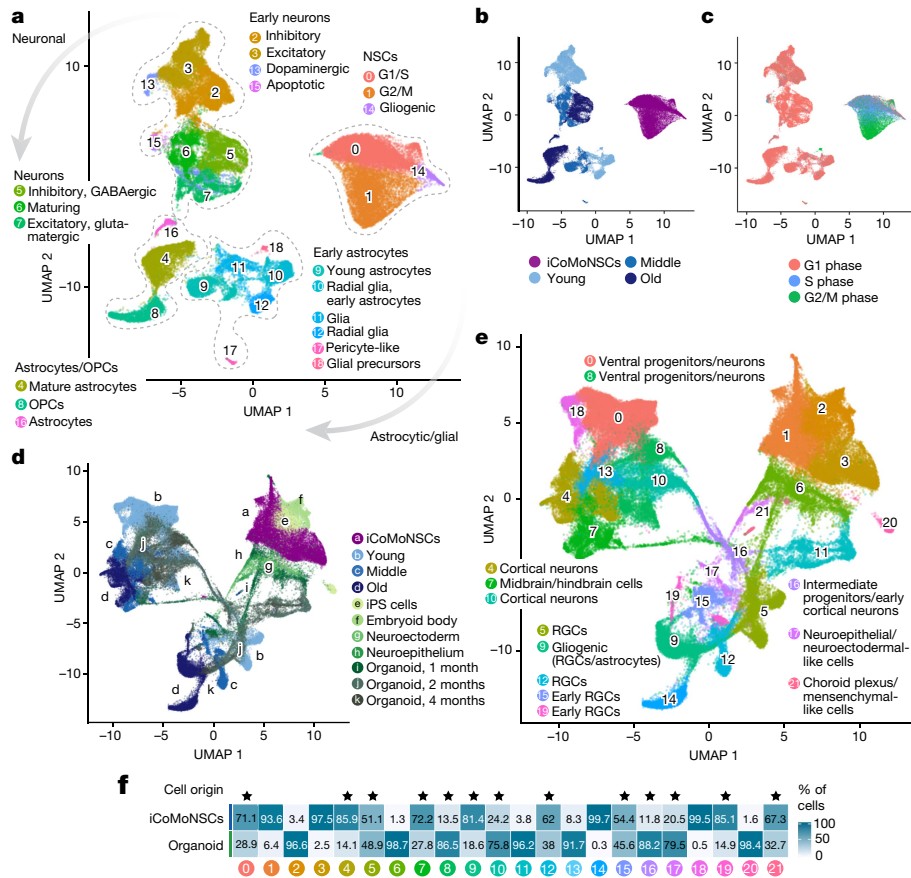

**Fig. 2 | iNets partially resemble brain organoids. a**, UMAP of young, middle stage and old iCoMoNSC-derived iNets. Colours highlight manually annotated clusters. Grey arrows indicate direction of neuronal or astrocytic or glial progressive maturation across all samples. **b,c**, UMAP data from **a**, annotated according to experimental (**b**) or predicted (**c**) cell cycle stage. **d,e**, UMAPs data from **a** integrated with organoid datasets highlighting the experimental stage origin (**d**) or clusters consisting of both iCoMoNSC and organoid cells (**e**). **f**, Cell distribution across all clusters. Stars indicate clusters with cell composition of at least 10% per origin. Cluster names as in **e**. Clusters with less than 10% of cells per origin: (1) iCoMoNSCs, (2) iPS cells, (3) iCoMoNSCs, (6) neuroepithelial-like/retina progenitors, (11) cycling ventral and dorsal neurons, (13) cortical neurons/ventral progenitors, (14) oligodendrocyte progenitor cells (OPCs), (18) maturing neurons and (20) iPS cells.

in young and middle stage cultures (Supplementary Fig. 2e,f). Neuronal maturation was highlighted by *SYP* (a neuronal marker) and *SLC32A1* (a marker of GABAergic (γ-aminobutyric acid-producing) neurons), which were detected in young, middle stage and old cultures, whereas *SLC17A6* (a glutamatergic neuron marker) was mostly detected in old cultures (Supplementary Fig. 2g). Clusters were manually annotated based on analysis of cluster markers (Supplementary Fig. 2b,h), known markers (Supplementary Fig. 3), CoDex (cortical development expression) viewer[34], PanglaoDB database[35] and UCSC Cell Browser. In line with the increasing maturation over time in differentiation, young neurons were annotated as young inhibitory (cluster 2), excitatory (cluster 3) and dopaminergic neurons (cluster 13), as well as apoptotic neurons (cluster 15); at the middle stage, cell clusters were annotated as maturing (cluster 6) and excitatory glutamatergic (cluster 7); and in the old cultures, neuronal subtypes were clearly defined as GABAergic (cluster 5) and glutamatergic (cluster 7). Similarly, gliogenic clusters were annotated as glial precursors (cluster 18), radial glia/early astrocytes (cluster 10), radial glia (cluster 12) and glia (cluster 11) in young cultures; young astrocytes (cluster 9) and glia (cluster 11) at the middle stage; and mature astrocytes (cluster 4), astrocytes (cluster 16) and OPCs (cluster 8) in the old cultures. A small percentage of cells (between 0.16 and 0.7%) was identified as pericytes (cluster 17), regardless of the maturation stage (Fig. 2a–c, Supplementary Fig. 2). The transcriptional maturation of the differentiating iNets was congruent with pseudotime

analysis (Extended Data Fig. 4a–c). Of note, the apparent lack of layer 5 corticospinal motor neurons in iNets indicates that this model does not constitute a complete model of ALS.

To determine the level of maturation of the emerging cell types within iNets, we integrated our data with two scRNA-seq datasets for brain organoids[8] (Fig. 2d and Extended Data Fig. 4d) and identified 22 clusters (Fig. 2e). Cells from the two datasets were mixed in most clusters, but overall the datasets occupied different parts of the two-dimensional space, indicating transcriptomic differences within cell types, potentially depicting differences in their developmental stages, although we note that we cannot exclude the contribution of uncorrected batch effects (Extended Data Fig. 4e,f). With the exception of nine clusters consisting primarily of cells from iNets or from organoids only, all other clusters contained cells originating from both systems (Fig. 2f). In sum, our neural model intersected with human cortical brain organoids within clusters representing intermediate and ventral progenitors, early and late cortical neurons and radial glial cells. Differences were driven by the source cells—that is, NSCs in our model and iPS cells in the brain organoid models. These results suggest that our neural model contained neuronal and glial cells of a transcriptional maturation level similar to that in brain organoids. Notwithstanding the differences in the source cells, these data indicate that, even in the absence of a 3D organ-like cell organization, neuronal and glial gene expression in our iNets bears similarity to that in cortical organoids.

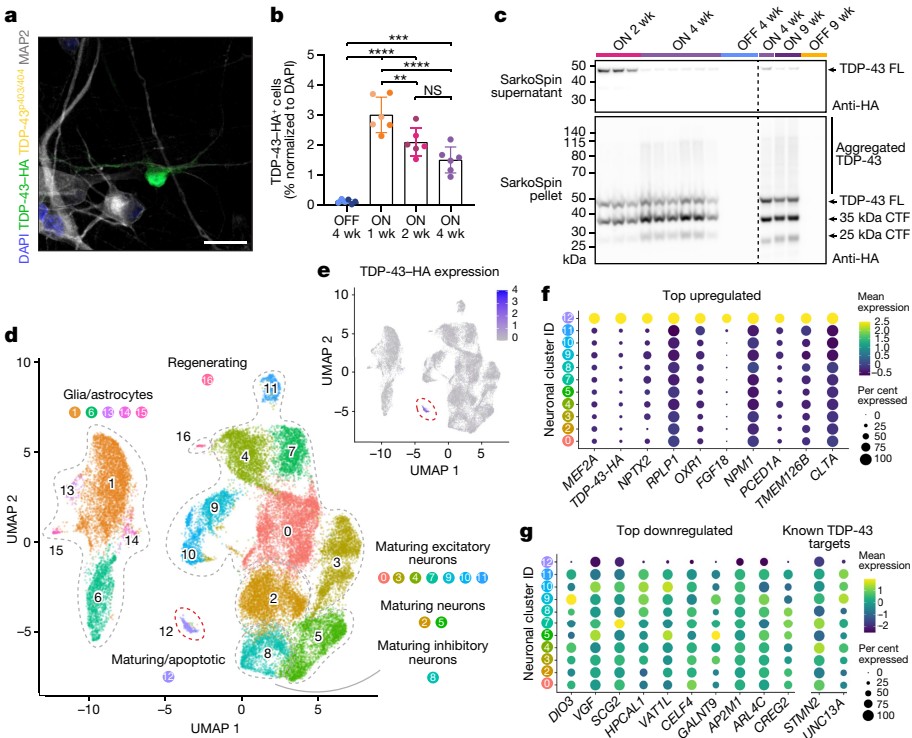

**Fig. 3 | A distinct transcriptional profile in iNet neurons with induced TDP-43 pathology. a**, Immunofluorescence of TDP-43–HA in iNet neurons (representative image from seven repeats). Scale bar, 25 μm. **b**, Quantification of TDP-43–HA-positive neurons over time. Data from 2 representative experiments (out of $n = 7$). Each data point represents a sum of all TDP-43–HA⁺ cells counted from 182 fields of view of an independent well and normalized with DAPI. Each colour represents an independent experiment. Between 3,728 and 8,248 cells were analysed per data point. One-way ANOVA followed by Tukey's multiple comparison (mean of each dataset compared with the mean of every other dataset). Data shown are mean ± s.d. OFF versus ON 1 week, $P < 0.0001$; OFF versus ON 2 weeks, $P < 0.0001$; OFF versus ON 4 weeks,

$P = 0.0001$; ON 1 week versus ON 2 weeks, $P = 0.0081$; ON 1 week versus ON 4 weeks, $P < 0.0001$. Wk, week. **c**, Western blots of SarkoSpin supernatants (top) and pellets (bottom). The dashed line separates independent experiments (the experiment was repeated two times in this setting). CTF, C-terminal fragment; FL, full-length. **d**, UMAP of scRNA-seq TDP-43 overexpression experiment. Colours indicate clusters and dashed lines highlight different cell types. The red dashed line outlines cluster 12, which contains cells that express TDP-43–HA. **e**, UMAP highlighting that TDP-43–HA expression is confined to cluster 12. **f,g**, Dot plot with the scaled average expression of the top 10 upregulated (**f**) and downregulated (**g**) marker genes as well as two known TDP-43 targets (**g**) in cluster 12 compared with other neuronal clusters.

## Sparse TDP-43 pathology in iNets

TDP-43 pathology characterizes affected brain regions of patients with TDP-43 proteinopathies, yet it was recently reported that in FTLD, less than 2% of cortical cells show pathological changes in TDP-43[36]. To simulate this, we transduced young iNets with a lentiviral vector[37] expressing human TDP-43 with a haemagglutinin tag (TDP-43–HA) under the control of the tetracycline responsive element (TRE) promoter[37] (Extended Data Fig. 5a,b). We used a low titre, aiming to transduce around 2% of cells in the network after 1 week of induction (ON induced; OFF not induced) and observed a gradual decrease in the percentage of transgenic neurons over time (Fig. 3a,b), indicating that TDP-43–HA overexpression was toxic to human neurons. We then analysed TDP-43–HA biochemically using SarkoSpin, a method for the specific enrichment of pathological TDP-43 species[14]. We detected progressive accumulation of TDP-43–HA in the SarkoSpin pellet, accompanied by fragmentation and the appearance of high-molecular-weight bands and smears indicative of aggregation (Fig. 3c and Extended Data Fig. 5c,d), reminiscent of preparations from post-mortem brains of patients with TDP-43 proteinopathy[14]. Specifically, in total cell lysates, we detected progressive fragmentation of full-length TDP-43–HA into 35-kDa and 25-kDa C-terminal fragments over time (Extended Data Fig. 5e). Both TDP-43 fragments accumulated in the SarkoSpin pellet, whereas full-length TDP-43–HA was present in the soluble fraction (SarkoSpin supernatant) at two weeks post induction, but was progressively redistributed to the SarkoSpin pellet at later time points (Fig. 3c).

Collectively, our data demonstrate that overexpression of wild-type TDP-43–HA in human neurons resulted in progressive aggregation and fragmentation and the loss of TDP-43–HA-overexpressing cells. Surprisingly, although we found no evidence that TDP-43–HA-overexpressing neurons develop inclusions with a phosphorylated form of TDP-43 (TDP-43p403/404) (Fig. 3a), TDP-43p403/404 emerged and amplified over time in non-transgenic neurons present in the same iNets (Extended Data Fig. 5f,g), showing that pathological TDP-43 changes extend beyond the initially affected cells. At early time points, TDP-43p403/404 signal appeared in the form of small, dot-like pre-inclusions that were largely confined within the soma (Extended Data Fig. 5h), whereas at later time points, these inclusions were larger and additionally present in neuronal processes (Extended Data Fig. 5i). This indicates progressive maturation of TDP-43 inclusions into aggregates in aged iNets.

## TDP-43 pathology alters neuronal RNA profile

To understand the effect of sparse TDP-43–HA overexpression and related pathology in iNets, we induced its expression for two or four weeks, before collecting the cells for scRNA-seq. Samples were analysed at the middle stage (around 3 months), and consisted of both inhibitory and excitatory neurons (Fig. 2a,b) interconnected into functional networks (Fig. 1k). Between 6,000 and 10,000 cells per sample were retained after pre-processing and filtering, with a median number of detected genes of 4,300–5,100 and a median number of UMIs between 15,000 and 20,000. We identified 17 clusters (Fig. 3d),

with a very similar cell type distribution (apoptotic, glial/astrocytic, maturing inhibitory and excitatory neurons) to our non-transduced middle stage samples (Fig. 2a,b), pointing to the reproducibility of cell identities across independent iNets, as shown by the mixing of cells from different experiments when plotted on the same UMAP graph, and by the high pairwise cluster replicability scores for equivalent cell identities across experiments (Extended Data Fig. 6a–c). We quantified the expression of the transgenic TDP-43–HA transcript in all samples and identified marker genes for each cluster. This analysis revealed a single cluster (number 12) that was composed almost exclusively of cells overexpressing TDP-43–HA (only 1.82% of cells in cluster 12 were from the OFF sample and the rest from the ON samples, in particular 42.27% were from the ON sample at 2 weeks; 39.55% of cells were from the ON sample at 4 weeks r1 and 16.36% were from the ON sample at 4 weeks r2) (Fig. 3e and Extended Data Fig. 6a). Cells in cluster 12 showed an increase in total TDP-43–HA expression (log$_2$ fold change of 1.79) compared with all other neuronal clusters, and we were able to detect transgenic TDP-43–HA in 96.82% of cluster 12 cells, but in only 22.21% of all other neuronal cells. Similarly, we detected the construct long terminal repeats or rtTA protein in 96.11% of cluster 12 cells, but in only in 19.01% of all other neuronal cells. Overexpression of TDP-43–HA over a period of 2 or 4 weeks altered the expression of several genes that were identified as cluster 12 markers and were either upregulated (Fig. 3f and Supplementary Table 1) or downregulated (Fig. 3g and Supplementary Table 2) compared to all other neuronal clusters. Within cluster 12, we noticed a significant downregulation of STMN2[3,5] and UNC13A[2,4], which were previously identified as human-specific RNA targets of TDP-43 (Fig. 3g and Supplementary Table 2), validating the relevance of our model for human disease. Of note, several marker genes that were not previously associated with or directly linked to ALS or FTLD neuropathology were found to be upregulated in cluster 12. Among these, neuronal pentraxin 2 (encoded by NPTX2 (also known as NARP)), is a neuron-specific protein that is secreted and involved in long-term neuronal plasticity. To understand whether any of these differentially expressed RNAs were directly bound by TDP-43, we analysed previously published iCLIP datasets from brains of patients with FTLD and controls[13]. We found that all marker genes that we identified to be misregulated in cluster 12 were indeed binding targets of TDP-43, and TDP-43 binding to the vast majority of these mRNA molecules is altered in the brains of patients with FTLD compared with controls (Extended Data Fig. 6d).

## TDP-43 regulates NPTX2 via 3′ UTR binding

We next investigated whether the expression of any of the newly identified cluster 12 marker genes was altered in brain samples from patients with ALS and FTLD. To that end, we re-analysed a RNA-sequencing dataset comparing the transcriptomic profiles of single nuclei from individual neurons with (TDP-43-negative) or without (TDP-43-positive) nuclear clearance derived from brains from patients with FTLD–ALS[38], which is a consequence of pathological TDP-43 accumulation and sequestration of functional protein. Out of the top mRNAs that were upregulated (Fig. 4a) or downregulated (Extended Data Fig. 6e) in cluster 12, six were significantly altered in the same direction in TDP-43-negative nuclei[38] (Supplementary Table 3). The strongest upregulation was for NPTX2 mRNA, which was consistently increased twofold in TDP-43-negative neurons from patients with FTLD–ALS compared with controls (Fig. 4a). Moreover, iCLIP data[13] analysis showed that TDP-43 bound directly to NPTX2 mRNA, primarily at its 3′ untranslated region (UTR) within a highly GU-rich region (Fig. 4b), the sequence specifically identified by the RNA recognition motifs of TDP-43[11,39]. Importantly, this TDP-43–NPTX2 interaction was reduced in brains from patients with FTLD, as shown by the loss of iCLIP crosslinks, marking positions of direct protein–RNA interactions (Fig. 4b).

Notably, many TDP-43 binding sites are not conserved in mice[20], as is the case for STMN2[3,5] and UNC13A[4]. Transcriptomic alterations upon TDP-43 dysregulation are therefore frequently human-specific. We tested whether this is also the case for NPTX2. In vivo iCLIP data[11] showed an absence of TDP-43 crosslinks in the 3′ UTR of Nptx2 in the adult mouse brain, in line with the low sequence conservation scores in this region (Extended Data Fig. 7a). In addition, whereas the rise in NPTX2 mRNA in iNets expressing TDP-43–HA was paralleled at the protein level (as discussed below), this was not the case in primary mouse neurons (Extended Data Fig. 7b). Collectively these data suggest that TDP-43 directly regulates NPTX2 mRNA levels by binding to its 3′ UTR, an event that is human-specific and is disturbed in human neurons with TDP-43 pathology.

Binding of TDP-43 to the 3′ UTR of target genes has been shown to result in reduced mRNA levels of the targets, including GRN and TARDBP[11]. This mechanism mediates the well-established autoregulatory pathway of TDP-43[11,12,40]. To understand whether TDP-43 binding on NPTX2 has a similar effect, we cloned the full 3′ UTR of either TARDBP or NPTX2 downstream of the stop codon of a Renilla luciferase open reading frame (ORF) before transfection in HEK293T cells, along with an unaltered firefly luciferase ORF, which served as a normalization control. Knockdown of TDP-43 (Extended Data Fig. 7c,d) significantly increased the levels of bioluminescence produced from Renilla luciferase fused to the full 3′ UTR of either TARDBP or NPTX2 (Fig. 4c), whereas acute (72 h) overexpression of TDP-43–HA had the opposite effect (Fig. 4d), indicating a bidirectional regulation of NPTX2 mRNA by TDP-43, reminiscent of the regulation on its own mRNA. Next, we tested whether the effect of TDP-43 knockdown could be rescued by the simultaneous overexpression of TDP-43–HA. We generated lentiviral vectors encoding both hU6-driven short hairpin RNA (shRNA) against endogenous TARDBP (or control shRNA) and TRE-driven TDP-43–HA (resistant to shRNA), transduced them into HEK293T cells and induced expression of TDP-43–HA a day later. For the luciferase TARDBP 3′ UTR reporter, we indeed detected a significant rescue of the TDP-43-knockdown phenotype with co-expression of wild-type TDP-43–HA, whereas the effect of TDP-43 knockdown was more severe with overexpression of an RNA-binding-deficient TDP-43–HA mutant (RRMm[41]) (Extended Data Fig. 7e). This demonstrates that it is indeed the RNA-binding function of TDP-43 that mediates the changes in the TARDBP reporter signal. Importantly, similar results were observed with the NPTX2 3′ UTR luciferase reporter in an identical setup (Extended Data Fig. 7e), further confirming the essential regulatory function of the RNA recognition motif-dependent binding of TDP-43 on the 3′ UTR of NPTX2.

## FTLD–ALS RNA profiles mimicked by changes in TDP-43 levels

Expression of TDP-43–HA in iNets causes a loss of solubility of the full-length protein and downregulation of STMN2 and UNC13A, and also culminates in the accumulation of insoluble and pathologically phosphorylated TDP-43 species (Fig. 3c,g and Extended Data Fig. 5c–i). Moreover, luciferase assays suggested that NPTX2 upregulation results from acute loss of function of TDP-43 (Fig. 4c,d). To determine whether the misregulation of NPTX2 induced by TDP-43 is caused by loss or gain of TDP-43 function, we compared the transcriptional profile of iNets expressing TDP-43–HA to iNets in which TDP-43 was knocked down via lentivirus-mediated shRNA, under conditions in which most of the cells were transduced. Bulk RNA sequencing of TDP-43–HA-expressing iNets again identified NPTX2 as a top upregulated gene (Extended Data Fig. 8a and Supplementary Table 4), validating our scRNA-seq data (Fig. 3f and Supplementary Table 1). shRNA targeting TARDBP efficiently reduced TDP-43 expression (log$_2$ fold change of −1.74) (Extended Data Figs. 7c,d and 8b–d and Supplementary Table 5). This reduction in TDP-43 levels resulted in a decrease in STMN2 and UNC13A

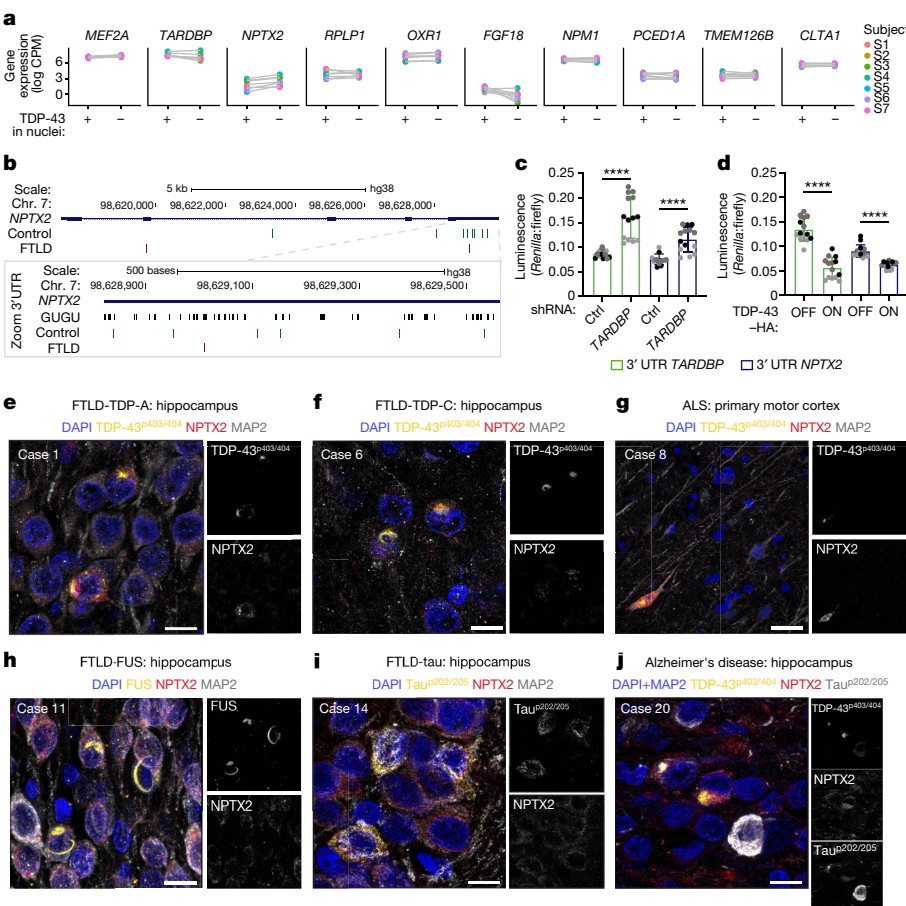

**Fig. 4 | Loss of TDP-43 binding to the 3′ UTR of *NPTX2* leads to its aberrant accumulation in brains of patients with TDP-43 proteinopathy. a**, Gene expression (in CPM (counts per million)) of the top 10 marker genes that are upregulated in cluster 12 in matched TDP-43-negative and TDP-43-positive neuronal nuclei from patients with FTLD–ALS[38] (subject numbers match those in ref. 38). **b**, Top, location of iCLIP crosslink sites in *NPTX2* in control (green) and FTLD patient (red) human brains[13]. Bottom, magnified view of iCLIP crosslinks and GUGU repeats in the *NPTX2* 3′ UTR. **c,d**, Dual luminescence assay showed similar behaviour of *TARDBP* and *NPTX2* 3′ UTR reporters upon TDP-43 knockdown (**c**) and TDP-43–HA overexpression (**d**). Pairs analysed by Mann–Whitney *U* test. Different shades of grey indicate independent experiments. *n* = 14 independently treated wells over 3 experiments. Data shown are mean ± s.d. Two-tailed *P* values. Control (Ctrl) shRNA versus *TARDBP* shRNA for both *TARDBP* 3′ UTR and *NPTX2* 3′ UTR, *P* < 0.0001; TDP-43 OFF versus ON for both *TARDBP* 3′ UTR and *NPTX2* 3′ UTR, *P* < 0.0001. **e–j**, Immunofluorescence of human brain sections: FTLD-TDP-A (**e**) or FTLD-TDP-C (**f**) hippocampus, ALS primary motor cortex (**g**), and FTLD with FUS pathology (FTLD-FUS) (**h**), FTLD with tau pathology (FTLD-tau) (**i**) or Alzheimer's disease (**j**) hippocampus, demonstrating inclusion-like NPTX2 signal only in MAP2-positive neurons containing aggregates composed of TDP-43[p403/404] but not FUS or phosphorylated tau (tau[p202/205]). Immunofluorescence was repeated four (FTLD-TDP-A and FTLD-TDP-C) or two times (FTLD-FUS, FTLD-tau, Alzheimer's disease and ALS). Scale bars: 10 μm (**e,f,h–j**), 30 μm (**g**).

transcripts (Extended Data Fig. 8b and Supplementary Table 5), as well as *PFKP*, *RCAN1*, *SELPLG* and *ELAVL3*, which were previously shown to be downregulated upon TDP-43 knockdown in iPS cell-derived human motor neurons[3] (Supplementary Table 5), confirming that the classical downstream consequences of TDP-43 knockdown also take place in iNets. Notably, TDP-43 knockdown induced a mild but significant increase in *NPTX2* expression (log$_2$F fold change of 3.27 upon overexpression, log$_2$ fold change of 0.402 upon knockdown), which was confirmed on the protein level by western blot (Extended Data Fig. 8c). Furthermore, immunofluorescence of iNets with TDP-43 knockdown revealed an increase in NPTX2 levels in neurons with reduced TDP-43 levels versus non-transduced neighbouring neurons, yet this increase in NPTX2 was minor compared with that in TDP-43–HA-expressing neurons (Extended Data Fig. 8d). In the cases of *STMN2* and *UNC13A*, their downregulation upon TDP-43 knockdown is the result of cryptic exon inclusion, a phenomenon that we replicated in iNets (Extended Data Fig. 8e). By contrast, we did not find evidence for a cryptic exon in *NPTX2* (Extended Data Fig. 8e). Together, these data indicate a probable partial contribution of TDP-43 loss of function to NPTX2 dysregulation.

Gain of function arising because of TDP-43 overexpression may further increase NPTX2 levels. A known consequence of TDP-43 overexpression is the exclusion of constitutive exons[42,43] (skiptic exons). To understand whether the gene dysregulation in our model of TDP-43 overexpression was associated with such skiptic exon events, we performed differential splicing analysis in the bulk RNA-sequencing dataset. This revealed 2,326 genes with differentially spliced exons (Extended Data Fig. 8f and Supplementary Table 6). When we performed the same analysis with our TDP-43-knockdown bulk RNA-sequencing dataset, we found a similar number of genes (2,014) with differentially spliced exons, out of which 740 were shared with the TDP-43–HA overexpression dataset (Extended Data Fig. 8f and Supplementary Table 6), suggesting that these events are not specific to TDP-43 overexpression. Of note, in our datasets, we did not detect a significant change in the reported TDP-43 overexpression-induced skiptic exon-containing genes[43] (*XPNPEP1*, *NUP93*, *MYBBP1A* and *HYOU1*). To better understand the relative contributions of loss-of-function and gain-of-toxicity pathways to boosting NPTX2 levels, we compared the alterations in the transcriptomes of three groups: iNets overexpressing TDP-43–HA, iNets with TDP-43 knockdown and neurons with nuclear loss of TDP-43 in the

brains of patients with FTLD–ALS[38] (Extended Data Fig. 9a). A subset of transcripts was altered both in iNets with TDP-43–HA expression and in iNets with TDP-43 knockdown (17% for downregulated genes, 14% for upregulated genes), including *NPTX2* (Extended Data Fig. 9b). Even though only a small fraction of transcripts is changed in iNets with altered TDP-43 levels and in brains of patients with FTLD–ALS (Extended Data Fig. 9b and Supplementary Tables 7 and 8), our analysis shows that both overexpression and downregulation of TDP-43 recapitulate part of the transcriptomic alterations observed in affected neurons in patients with FTLD–ALS (Extended Data Fig. 9a,b and Supplementary Tables 7 and 8). These results indicate that both overexpression and silencing of TDP-43 levels mimic part of the pathological process occurring in neurons with TDP-43 pathology in patient brains, and suggest that both alterations contribute to the development of NPTX2 pathology in TDP-43 proteinopathies.

## NPTX2 buildup in TDP-43 proteinopathies

To understand whether NPTX2 protein is specifically altered in neurons with TDP-43 pathology, we imaged NPTX2 in iNets using immunofluorescence, and found that this normally secreted and synaptically localized protein[44,45] is aberrantly accumulated within neuronal somata and processes of TDP-43–HA-expressing neurons (Extended Data Figs. 10a–c and 11a). Validating our cellular model and analysis, the affected neurons also exhibited an increase in protein levels of MEF2A and a decrease in STMN2 (Extended Data Fig. 10d,e), as predicted (Fig. 3f,g). Further confirming our findings, we also found NPTX2 accumulation downstream of TDP-43–HA overexpression driven by the less potent constitutive eukaryotic translation elongation factor 1α (EF-1α, gene symbol *EEF1A1*) promoter, instead of the inducible TRE promoter (Extended Data Fig. 11). Moreover, to further reduce the levels of ectopic TDP-43–HA expression, we used constitutive EF-1α-driven TDP-43–HA including its own 3′ UTR, which autoregulates the levels of the transgene—similarly to the endogenous *TARDBP*—and results in reduced NPTX2 accumulation (Extended Data Fig. 11). Of note, expression of HA-tagged FUS, a protein that aggregates in the central nervous system (CNS) of patients with FTLD-FUS and ALS subtypes, induced weak NPTX2 accumulation in only a minority of transgenic iNet neurons (Extended Data Fig. 11a), confirming the specificity of *NPTX2* upregulation to TDP-43 dysfunction.

We then explored whether NPTX2 accumulation also occurred in the brains of patients with neurodegenerative conditions (Supplementary Table 9). To this end, we investigated autopsy brain material from 20 patients with neurodegenerative conditions, including 7 FTLD with TDP-43-immunoreactive pathology (FTLD-TDP), 1 ALS, 3 FTLD-FUS, 4 FTLD-tau and 5 Alzheimer's disease. We used immunofluorescence for NPTX2, TDP-43[p403/404] and MAP2 (a neuronal marker) and found that although NPTX2 levels were very low in neurons of the patients with neurodegenerative conditions, it accumulated into dense, dot-like inclusions in both the somata and processes of neurons with TDP-43 pathology. We observed this characteristic NPTX2 accumulation in the granule cell layer of the dentate gyrus of the hippocampus of patients with FTLD-TDP type A (FTLD-TDP-A; both sporadic and C9ORF72-related) or type C[46] (FTLD-TDP-C), as well as in the frontal cortex of patients with FTLD-TDP-A and in the primary motor cortex of a patient with ALS (Fig. 4e–g and Extended Data Fig. 12a–d). Notably, NPTX2 did not appear to co-aggregate with TDP-43, consistent with RNA misregulation downstream of TDP-43 pathology, rather than direct co-aggregation of the two proteins. To probe the specificity of aberrant accumulation of NPTX2 to TDP-43 pathology, we additionally immunolabelled hippocampal tissue from patients with FTLD-FUS and hippocampal and frontal cortex tissue from patients with FTLD-tau (Supplementary Table 9) using anti-FUS or anti-tau[p202/205] antibodies to label FTLD subtype-specific inclusions. We did not detect any difference in the NPTX2 labelling pattern in FUS or tau inclusion-bearing neurons

compared with their aggregate-free neighbours (Fig. 4h,i and Extended Data Fig. 12e,f), in line with the scarcity of NPTX2 accumulation in HA–FUS-overexpressing iNets (Extended Data Fig. 11a). Notably, whereas aberrant NPTX2 was similarly absent from neurons with tau inclusions in the hippocampus of patients with Alzheimer's disease (Extended Data Fig. 12g), we detected aberrant accumulation of NPTX2 in TDP-43 aggregate-bearing neurons in cases of Alzheimer's disease with TDP-43 co-pathology (Fig. 4j and Extended Data Fig. 12h). Together, this signifies that *NPTX2* misregulation is specific to TDP-43 pathology and not a consequence of protein aggregation in general.

## Lowering NPTX2 rescues TDP-43 toxicity

To investigate the effect of increased NPTX2 levels on neuronal viability, NPTX2–HA was expressed in iNets and the number of surviving transgenic neurons was compared with iNets expressing TDP-43–HA with or without its 3′ UTR and HA–FUS (Fig. 5a). Overexpression of NPTX2–HA resulted in its accumulation in cytoplasm and neurites (Fig. 5b and Extended Data Fig. 13a), mimicking the pattern of NPTX2 accumulation seen in TDP-43–HA-positive neurons (Extended Data Fig. 10a,b and 11a). Of note, direct upregulation of NPTX2–HA, but not of HA–FUS, was neurotoxic (Fig. 5c). Moreover, we observed a significant positive correlation between the level of TDP-43–HA and the percentage of degenerating neurons (Extended Data Fig. 13b). Specifically, we found that TDP-43–HA overexpression driven by the inducible TRE promoter resulted in the highest loss of transgenic neurons over time (Figs. 3b and 5c), whereas constitutive TDP-43–HA overexpression by the less potent EF-1α promoter induced milder neurotoxicity, and further restriction of TDP-43–HA levels by its 3′ UTR was not toxic in this timeframe (Fig. 5c and Extended Data Fig. 11b). Notably, neurotoxicity by TDP-43–HA was more severe than that of NPTX2–HA alone (Fig. 5c), highlighting NPTX2 as one of the downstream executors of TDP-43-driven toxicity.

The neurotoxicity associated with an increase in NPTX2 levels suggests that correcting faulty NPTX2 levels could salvage neurons with TDP-43 proteinopathy. To assess this, we designed shRNAs targeting *NPTX2*, and the two best performing shRNAs were selected for testing in iNets with TDP-43 proteinopathy (Extended Data Fig. 13c–f). We administered *NPTX2* shRNA or control shRNA one day prior to the induction of TDP-43–HA expression, meaning that NPTX2 downregulation was established while TDP-43–HA was triggering its downstream pathological cascades (Fig. 5d). Indeed, treatment with the selected shRNAs markedly lowered NPTX2 levels in iNets expressing TDP-43–HA as probed by western blot (Extended Data Fig. 13e,f) and immunofluorescence analysis compared with control shRNA or no treatment (Fig. 5e and Extended Data Fig. 13g,h). Comparison of the TDP-43–HA-positive cell counts after treatment with the control shRNAs showed either no effect or a slight increase in toxicity, whereas treatment with one of the two *NPTX2* shRNAs showed a mild benefit that did not reach statistical significance (Extended Data Fig. 13i), possibly related to off-target toxicity that is often associated with shRNA treatment. However, treatment with *NPTX2* shRNA D revealed that the correction of NPTX2 levels rescued cells from TDP-43-induced neurotoxicity (Fig. 5f). This NPTX2 silencing-mediated rescue was significant but partial, in line with the multitude of transcripts altered upon TDP-43 dysregulation, of which several are likely to exert toxicity. Overall, our work identified aberrant NPTX2 accumulation as a direct consequence of TDP-43 misregulation in disease, revealed a novel neurotoxic pathway for TDP-43 proteinopathies and provided proof of principle for the correction of NPTX2 levels as a therapeutic strategy (Fig. 5g).

## Discussion

Long-term self-renewing NSC lines derived from embryonic stem cells or iPS cells provide an excellent source of expandable cells[6,7,47]

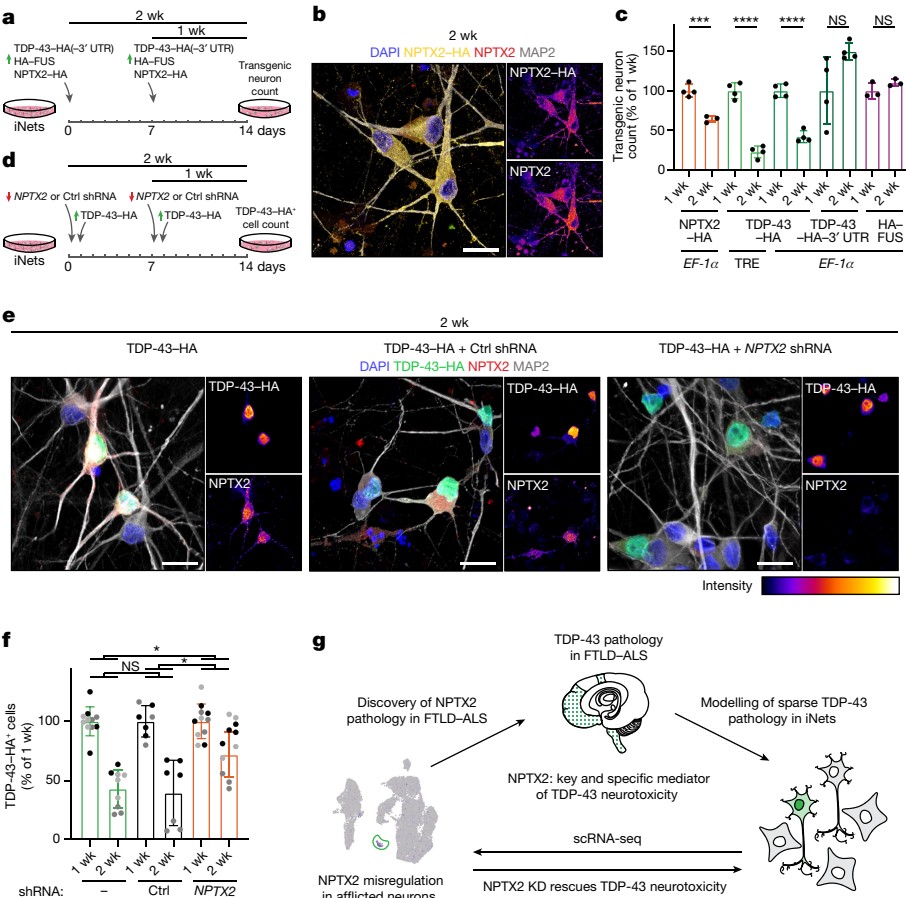

**Fig. 5 | Correcting toxic levels of NPTX2 rescues TDP-43-induced neurodegeneration. a**, Experimental timeline for (**b**,**c**). **b**, HA-tag and NPTX2 immunofluorescence in iNet neurons overexpressing NPTX2–HA at 2 weeks (1 experiment). **c**, Quantification of transgenic neurons. Each data point represents a sum of all HA-tag+ cells counted from 36 fields of view of an independent well and normalized with DAPI and MAP2, shown as a percentage of the of 1-week condition. Between 281 and 2,680 cells were analysed per data point. *n* = 4 independent wells. Unpaired *t*-test for 1 week versus 2 weeks time points for each condition. Data shown are mean ± s.f. Two-tailed *P* values. EF-1α-driven NPTX2–HA 2 week versus 1 week, *P* = 0.0009; TRE-driven TDP-43–HA 2 week versus 1 week, *P* < 0.0001; EF-1α-driven TDP-43–HA 2 week versus 1 week, *P* < 0.0001. **d**, Experimental timeline for **e**,**f**. **e**, Immunofluorescence of iNets expressing TDP-43–HA with or without Ctrl or *NPTX2* shRNA treatment at

2 weeks. **f**, Quantification of TDP-43–HA-transgenic cells in **e**. Each data point represents a sum of all TDP-43–HA+ cells counted from 182 fields of view of an independent well and normalized with DAPI, shown as a percentage of the of 1-week condition. Between 189 and 16,453 cells were analysed per data point. Two-way ANOVA (mixed model) followed by Tukey's multiple comparisons test (mean of each dataset compared with the mean of every other dataset). Different shades of grey indicate independent experiments. Data shown are mean ± s.d. TDP-43–HA only versus TDP-43–HA + *NPTX2* shRNA, *P* = 0.0337; TDP-43–HA + control shRNA versus TDP-43–HA + *NPTX2* shRNA, *P* = 0.0312. **g**, Graphical summary highlighting the NPTX2 misregulation and accumulation in iNets and in patients of the ALS and FTLD spectrum or Alzheimer's disease with TDP-43 proteinopathy, and the alleviation of TDP-43 neurotoxicity by correction of NPTX2 levels. Scale bars, 20 μm.

for creating models of neurogenesis and CNS-related diseases. Here we generated iCoMoNSCs and showed that they represent a uniquely homogenous population of self-renewing NSCs with the potential to give rise to mature and diverse neuronal and glial subtypes. Notably, only a very small percentage (around 0.3%) of iCoMoNSCs was identified as committed neuroblasts. Upon terminal differentiation, iCoMoNSCs consistently generated mixed neuronal and glial cultures with electrophysiologically active neurons that formed interconnected networks with spontaneous coordinated activity, which we call iNets. scRNA-seq analysis confirmed the progressive maturation of neuronal and glial subtypes within iNets, similar to that in human brain organoids[8]. iNets have the advantage of containing a wide variety of cell types, including excitatory and inhibitory neurons and different types of glia, reflecting the heterogeneity of cellular types present in the human brain and brain organoids. Therefore, iNets are more relevant for CNS drug screening purposes than traditional 2D neuronal monocultures or co-cultures with a single glial subtype. Additionally, similar to typical 2D cell culture systems, we show that iNets are perfectly

suitable for tracking axons and studying network connectivity by plating on HD-MEAs, in contrast to 3D systems.

We explored how overexpression of TDP-43 in a minority of neurons would affect iNets over time, aiming to simulate the situation in the diseased brain, which is reported to contain only 2% of cells with TDP-43 pathology[38]. After four weeks of TDP-43–HA overexpression, we observed a decrease in soluble TDP-43–HA levels in parallel with progressive aggregation of insoluble TDP-43 and C-terminal fragmentation[1,14], as well as loss of TDP-43–HA-positive cells. scRNA-seq revealed that cells with TDP-43 overexpression and pathology were characterized by a distinct transcriptional profile, with downregulation of *STMN2*[3,5] and *UNC13A*[2,4], indicating TDP-43 malfunction. In contrast to data from the direct knockdown of TDP-43, however, we could not detect cryptic exons in iNets with expression of TDP-43–HA, possibly owing to their low expression levels and rapid degradation via the nonsense-mediated decay pathway, as shown in other studies[48]. Alternatively, the loss of function as a result of TDP-43–HA overexpression in this timespan may be milder than a global protein loss due to

knockdown. Another possibility is that the overexpression and knockdown may elicit their similar downstream phenotypes (upregulation of NPTX2 and downregulation of STMN2 and UNC13A) via distinct mechanisms. In line with this idea we show that the transcriptional profile of iNets with pure TDP-43 loss of function via knockdown and iNets with TDP-43 pathology overlaps only partially, and that both conditions mimic some of the RNA changes observed in nuclei of neurons with TDP-43 nuclear clearance from patients with FTLD–ALS[38]. Notably, our approach uncovers a novel set of up- and downregulated genes after TDP-43 overexpression. Remarkably, all of the top misregulated RNAs were direct targets of TDP-43, and the binding of the majority of these RNA molecules has previously been found to be altered in the human FTLD brain[13].

Comparing our novel RNA targets with RNA-sequencing data from TDP-43-negative cortical neurons of patients with FTLD–ALS[38] identified *NPTX2* as the most prominently upregulated RNA. TDP-43 binds *NPTX2* mRNA at its 3′ UTR, an event that is disturbed in FTLD[13]. The binding mode and increased *NPTX2* RNA levels indicate a regulatory mechanism that is distinct from cryptic exon suppression[20], which has been demonstrated for *STMN2*[3,5] and *UNC13A*[2,4]. Instead, TDP-43 binding on the 3′ UTR of *NPTX2* might affect the stability or transport of the mRNA, although the precise mechanism will be the focus of future studies. Notably, this misregulation led to aberrant accumulation of NPTX2 protein, which we consistently observed not only in our cellular model, but also in the brains of patients with FTLD–ALS. Indeed, NPTX2 accumulation in patients reliably marked cells with TDP-43 pathology, without co-aggregation of the two proteins. Notably, aberrant accumulation of NPTX2 was not observed in association with other proteinopathies, including FTLD-FUS or FTLD-tau, suggesting a specific link to TDP-43 misregulation. NPTX2 was recently shown to be decreased in the cerebrospinal fluid of patients with symptomatic genetic FTLD, in a manner correlating with clinical severity[49]. We do not know whether the NPTX2 accumulation that we observed in the minority of neurons with TDP-43 pathology is in any way related with the global decrease in NPTX2 levels with age[50] and in dementia[49–51], which is thought to indicate overall synaptic loss. Nonetheless, we cannot exclude the possibility of an interplay between the two events via mechanisms of homeostatic synaptic plasticity, for example.

NPTX2 is a neuronal pentraxin and an immediate early gene that is regulated by neuronal activity[45,52,53]. It is a $Ca^{2+}$-dependent lectin[45] that is secreted from presynaptic terminals[44,45], where it regulates synaptogenesis and glutamate signalling via AMPA receptor clustering[52,54,55]. NPTX2 binding to neuronal pentraxin receptor[54] (NPTXR) mediates synaptic maintenance, plasticity and postsynaptic specialization[52,56] in both excitatory and inhibitory synapses[56]. NPTX2 enhances glutamate receptor 1 function, and its upregulation causes synaptic remodelling by recruiting $Ca^{2+}$-permeable AMPA (α-amino-3-hydroxy-5-methyl-4-isoxazole propionic acid) receptors on the neuronal membrane[57]. Glutamate excitotoxicity has long been proposed to be one of the major pathological pathways selectively killing vulnerable neurons in ALS[58] and other neurodegenerative diseases[59]. Indeed, our data indicate that direct overexpression of NPTX2, which leads to a similar accumulation of the protein in the cytoplasm and neuronal processes as we observed downstream of TDP-43 overexpression, is neurotoxic in iNets. Notably, this direct NPTX2 toxicity was milder compared with TDP-43-triggered toxicity, which we found to correlate with the levels of TDP-43 overexpression, under differential promoter or regulatory control. This finding suggested that NPTX2 is one of several mediators of TDP-43 toxicity, and potentially acts synergistically with STMN2, UNC13A and possibly other TDP-43 targets. Yet, correcting the levels of NPTX2 only with shRNA significantly rescued TDP-43 neurotoxicity in iNets, indicating that NPTX2 is an important modifier of TDP-43 toxicity in human neurons. Given the known limitations of shRNA-mediated gene silencing[60], including off-target toxicity, we find this consistent

and significant rescue of iNets neurons to be particularly encouraging. Future efforts will focus on understanding the molecular mechanism of NPTX2 neurotoxicity, the potential synergistic effect of *NPTX2*, *STMN2* and *UNC13A* misregulation in disease and testing alternative regimens for correcting NPTX2 in models of TDP-43 proteinopathy. We propose that NPTX2 is a key neuronal target of TDP-43, and that its aberrant increase in affected neurons may trigger a pathological cascade leading to neuronal loss in FTLD–ALS. In this context, NPTX2 may represent an important therapeutic target for TDP-43 proteinopathies, along with STMN2 and UNC13A.

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

## Methods

### Generation of iPS cells

iPS cells were generated from control human early neonatal dermal fibroblasts (Gibco C0045C) with an apparently normal karyotype (checked at Cell Guidance Systems, according to CellGS fixed sample protocol) via episomal reprogramming using plasmids coding for *OCT3/4*, *SOX2*, *KLF4* and *TP53* shRNA[61,62]. Plasmids (Addgene #27077, #27078 and #27080; 1 µg each) were electroporated into 1 million fibroblasts using 100 µl Neon tips (Invitrogen MPK10025) on the Neon Transfection System (Pulse Voltage: 1,650 V, Width: 10 ms, Number: 3; Invitrogen MPK5000). After electroporation, cells were grown on gelatin-coated plates (EmbryoMax 0.1% Gelatin Solution (Millipore ES-006-B)) in fibroblast medium (DMEM (Sigma D5671); 20% FBS (Gibco 10270-106); 1× GlutaMAX (Gibco 35050-061); 1× NEAA (Gibco 11140-050); 1× Penicillin/Streptomycin (Sigma P4333-100ML)). On day 3 post-electroporation (and then until day 12), 0.5 µg ml$^{-1}$ of sodium butyrate (Sigma B5887) was added to the culture medium. The fibroblasts were then sub-cultured onto mouse embryonic fibroblasts (MEFs) (Global Stem GSC-6001G or Millipore PMEF-CFX) on the day 5 in iPS cell medium (KnockOut DMEM/F12 (Gibco 12660012); 20% KnockOut Serum Replacement (Gibco 10828028), 1× GlutaMAX (Gibco 35050-061); 1× NEAA (Gibco 11140-050); 80 ng ml$^{-1}$ bFGF (Gibco PHG0261); 1× EmbryoMax 2-Mercaptoethanol (Millipore ES-007-E)). The first iPS cell colonies were observed around day 11. Positive live alkaline phosphatase staining (Invitrogen A14353) was performed on day 14. iPS cell colonies were individually manually picked on day 14–21, resuspended and re-plated onto fresh MEF 24-well plates. iPS cell clones were then expanded and the ones that displayed best proliferation and morphology (NiPS 8, 9 and 10) were banked at passage 10/11 in iPS cell freezing medium (iPS cell medium +10% DMSO (Sigma D2650-100ML)). Upon further expansion to P12-15, cells were prepared for karyotype check at Cell Guidance Systems (according to CellGS fixed sample protocol) and based on their apparently normal karyotyping results, NiPS 10 was selected for further experiments.

### iCoMoNSCs

Similarly to our previous study on human embryonic stem cells[6], iPS cell colonies were manually picked and partially dissociated into smaller clumps and transferred into non-adhesive culture dishes and induced to form embryoid bodies in the presence of EB medium (iPS medium without bFGF). After 5–7 days, embryoid bodies were transferred onto poly-L-ornithine (Sigma P4957-50ML; 20 µg ml$^{-1}$ in sterile water (Gibco 10977035), 1 h at 37 °C followed by 3 washes with PBS (Gibco 10010015) and laminin (Gibco 23017-015); 5 µg ml$^{-1}$ in PBS, 1 h at 37 °C)-coated dishes and left to adhere in NSC medium (DMEM/F12 (Gibco 21331046); 0.5× B27- supplement (Gibco 12587-010), 0.5× N2 supplement (Gibco 17502-048); 1× GlutaMAX (Gibco 35050-061); 25 ng ml$^{-1}$ bFGF (Gibco PHG0261)). Formation of neural rosettes was observed within 4–10 days. Neural rosettes were manually dissected and picked under EVOS XL Core Imaging System (LifeTech AMEX1000), and after dissociation re-plated onto fresh poly-L-ornithine and laminin-coated dishes in NSC medium. After 2–5 days, new and smaller rosettes appeared (R1 rosettes) with the presence of heterogeneous contaminating cells. The R1 rosettes were then manually dissected, picked and dissociated into smaller clumps and re-plated onto fresh poly-L-ornithine and laminin-coated dishes. After further 2–5 days, new rosettes (R2) with minimal contaminating cells appeared. R2 rosettes were then routinely monitored to identify small groups of radially organized cells that were present outside of the neural rosettes and represented an independent 'clone-like population', the iCoMoNSCs. These small patches of iCoMoNSCs were then manually picked and transferred onto freshly poly-L-ornithine and laminin-coated 24-well plates. Clones that showed clear radial and consistent morphology, good attachment, survival and proliferation upon transfer were enzymatically detached using 0.05% Trypsin (Gibco 15400-054 diluted in PBS), which was blocked by 1× Defined Trypsin Inhibitor (Gibco R-007-100) and expanded for numerous passages and banked in NSC freezing medium (NSC medium +10% DMSO (Sigma-Aldrich D2650)). Upon further expansion to P9-13, iCoMoNSCs lines were prepared for karyotype check at Cell Guidance Systems (according to CellGS fixed sample protocol). iCoMoNSCs clone 10/80 with apparently normal karyotype was used in the study. iCoMoNSCs are available upon request.

### Differentiation of iCoMoNSCs into iNets

iCoMoNSCs were plated in NSC medium at 75,000 cells per cm$^2$ onto Matrigel-coated (Corning 354234; ~0.15 mg ml$^{-1}$ diluted in cold DMEM/F12 Gibco 11330032 and incubated at least 1 h at 37 °C) 6-well plates and left to recover and proliferate to reach ~95% confluency. At this point, NSC medium was switched to D3 differentiation medium (DMEM/F12 (Gibco 11330032); 0.5× B27+ supplement (Gibco 17504-044), 1× N2 supplement (Gibco 17502-048); 1× GlutaMAX (Gibco 35050-061); 1× Penicillin/Streptomycin (Sigma P4333-100ML)) supplemented with 5 µM Forskolin (Cayman AG-CN2-0089-M050), 1 µM synthetic retinoid Ec23 (Amsbio AMS.SRP002-2), 500 nM Smoothened agonist SAG (Millipore 5666600) for the first 5 days. On the days 6–10, Ec23 was increased to 2 µM. On days 11–25, Ec23 was decreased to 10 ng ml$^{-1}$, SAG to 50 nM and BDNF (PeproTech 450-02), GDNF (PeproTech 450-10) and CNTF (Alomone labs C-240) were added at 20 ng ml$^{-1}$. At day 26 and onwards, medium was switched to maturation medium (1:1 DMEM/F12:Neurobasal (Gibco 21103049) mix; 1× B27+ supplement, 1× N2 supplement; 1.5× GlutaMAX, 5 µM forskolin, BDNF, GDNF, CNTF, NT-3 (PeproTech 450-03) and IGF-1 (Stem Cell 78022) all at 20 ng ml$^{-1}$ and 10 µM cAMP (Sigma-Aldrich D0260)). Medium was changed daily and almost completely at days 0–10, whereas from this point on only two-thirds of the medium was changed 3 times a week.

For all experiments (except scRNA-seq where the experiment was performed in the original 6-well plate), young to middle stage iNets were dissociated into single-cells suspension using Papain Dissociation System (Worthington LK003150), passed through a 70-µm cell strainer (Falcon 07-201-431), resuspended in maturation medium (for imaging experiments, cells were passed through another 40 µm cell strainer (Falcon 352340)) and re-plated into the corresponding vessel at approximately 0.5 million cells per cm$^2$ as counted by CASY Cell Counter (Innovatis AG). 96-well imaging plates used: Greiner Bio-One 655090 or ibidi µ-Plate 96 Well Black 89626.

### Preparation of human SNSs

iNets were washed with PBS and homogenized as they were scraped off with the SNS buffer containing 0.35 M sucrose pH 7.4, 10 mM 4-(2 hydroxyethyl)-1-piperazineethanesulfonic acid (HEPES; Biosolve 08042359), 1 mM ethylenediaminetetraacetic acid (EDTA; VWR 0105), 0.25 mM dithiothreitol (DTT; ThermoFisher Scientific R0861), 30 U ml$^{-1}$ RNAse inhibitor (Life Technologies N8080119) and cOmplete-mini EDTA free protease inhibitor cocktail (Roche 11836170001; at least 2500 µl volume of buffer per one full 6-well plate of iNets). Homogenates were passed sequentially through three 100-µm Nylon net filters (Millipore NY1H02500), followed by one 5 µm filter (Millipore SMWP013000). The final filtrate was resuspended in 3 volumes of SNS buffer without sucrose and centrifuged at 2,000g, for 15 min at 4 °C to yield a pellet containing SNSs. The pellet was resuspended in 100 µl of SNS buffer, which was used for electron microscopy (60 µl) and western blot analysis (40 µl).

### Cloning of the all-in-one monocistronic TetON cassette, lentiviral vector preparation and transduction

Using NEBuilder HiFi DNA Assembly Cloning Kit (NEB E5520; HiFi kit), human wild-type, full-length TDP-43 with a C-terminal HA tag from a published pcDNA5 plasmid containing the human TDP-43 cDNA sequence[41,63] was inserted into our all-in-one monocistronic TetON

(mTRE; build based on Markusic et. al. [37]), which was previously inserted into a pLVX lentiviral transfer vector (Clontech 632164), while deleting CMV-PGK-Puro, generating mTRE-TDP-43–HA lentiviral transfer vector. mTRE cassette was build using both NEBuilder and Q5 site directed mutagenesis (NEB E0554S; Q5 kit) kits and consists of the Tet-responsive promoter P$_{tight}$, consisting of seven tet operator sequences followed by the minimal CMV promoter (sequence source pCW57.1; Addgene #41393), driving the inducible expression of the downstream TDP-43–HA, followed by downstream IRES2 sequence (sequence source Addgene #60857), which is immediately followed by T7 tag fused to SV40 NLS, which was fused to rtTA-Advanced (sequence source Addgene #41393), which made the rtTA predominantly nuclear, making it readily available for the system, while the T7 tag made rtTA a useful, independent marker of the transgenic cells. See Extended Data Fig. 5a,b for more details. To generate mTRE-GFP-FUS, GFP-FUS sequence from Hock et al. [27] was substituted into mTRE-TDP-43–HA via HiFi Kit.

mTRE-TDP-43–HA and all other lentiviruses were then packaged into lentivirus via Lipofectamine 2000 (Invitrogen 11668019; 8.3 µl per ml final volume) co-transfection with CMV-Gag-Pol (Harvard dR8.91) and pVSV-G (Clontech, part of 631530) plasmids into production HEK293T cells (ATCC CRL-3216; gift from the Greber laboratory) adapted to grow in serum-free conditions (OHN medium; based on Opti-MEM (Thermo-Fisher 11058-021), supplemented with 0.5% B27- (ThermoFisher 12587-010), 0.5% N2 (ThermoFisher 175020-01), 1% GlutaMAX (ThermoFisher 35050038) and bFGF (25 ng ml$^{-1}$; ThermoFisher PHG0261)), which reduces the expression of the gene of interest from the transfer vector (that is, it eliminates traces of tetracyclines in the FBS) as well as eliminates serum-carry over into the lentivirus supernatant. Medium was changed the following morning and supernatants were then collected 48 h post transfection (36 h post-medium change), centrifuged (500$g$, 10 min, 4 °C), filtered through Whatman 0.45-µm CA filter (GE 10462100) and concentrated using Lenti-X Concentrator (Takara 631232) according to the producer instructions (overnight incubation). The resulting lentiviral pellets were then resuspended in complete neuronal maturation medium to achieve 10x concentrated lentivirus preparations, which were titrated using Lenti-X GoStix Plus (Takara 631280). For scRNA-seq and immunofluorescence in Fig. 3b and Extended Data Fig. 5f–i, 10× concentrate of mTRE-TDP-43–HA lentivirus was then used at 1,600 ng (of lentiviral p24 protein as per GoStix Value (GV)) per well of a 6-well plate of differentiated iNets (around 2 months old) along with 3 µg ml$^{-1}$ of polybrene (Sigma-Aldrich TR-1003-G), pipetting the lentivirus concentrate directly onto the culture (drop-wise). Complete neuronal maturation medium was then added to reach 750 µl total. For all other experiments, all lentiviruses were used at 500 ng (GV) ml$^{-1}$ of medium in a total of 1,000 µl in sub-cultured 6-well plates, 350 µl in 12 well plates, 80 (Greiner) or 110 µl (ibidi) for imaging 96-well plates. Medium was exchanged completely the following day. For mTRE vectors, TDP-43–HA or GFP–FUS expression was induced by 1 µg ml$^{-1}$ of doxycycline (DOX; Clontech 631311) when needed.

### Cloning of shRNA and constitutive expression vectors

To generate shRNA-expressing vectors, our previously build pSHE lentivirus transfer vector[64] was modified using Q5 site directed mutagenesis kit (NEB E0554S; Q5 kit) to substitute the shRNA sequence for either *NPTX2* or *TARDBP*-targeting shRNAs (see Supplementary Table 11 for primer sequences) and using NEBuilder HiFi DNA Assembly Cloning Kit (NEB E5520; HiFi kit) to substitute the EGFP sequence with either TDP-43–HA (resulting in pshTDP vectors (used in NPTX2 rescue experiments) carrying hU6-driven *NPTX2* (or NT controls) shRNA and inducible mTRE-TDP-43–HA) or HaloTag (resulting in psHalo vectors (used in TDP-43-knockdown experiments) carrying hU6-driven *TARDBP* shRNA and inducible mTRE-HaloTag). Similarly, pshTDP vectors for dual luminescence experiments were generated so that they carry hU6-driven endo-*TARDBP* shRNA (or NT control) and inducible mTRE-TDP-43–HA (with or without mutations in the RNA-recognition

motif; shRNA-resistant (exploiting silent mutations present in our TDP-43–HA construct)).

To generate lentivirus transfer vectors for constitutive EF-1α-driven expression of TDP-43–HA (with or without 3′ UTR), NPTX2–HA and HA–FUS; using HiFi Kit, first the EF-1α promoter from custom synthetic sequence from GenScript was cloned into mTREAuto-TDP-43–HA vector substituting TRE promoter, generating pLVX-EF-1α-TDP-43–HA vector. Then either NPTX2 ORF (Origene SC122629) or wild-type FUS[27] was cloned into pLVX-EF-1α-TDP-43–HA vector, substituting TDP-43–HA, generating pLVX-EF-1α-NPTX2–HA and pLVX-EF-1α-HA-FUS vectors.

Then the full-length human *TARDBP* 3′ UTR (cloned out from cDNA in-house generated from HEK293T cells) was cloned using HiFi Kit directly downstream to the TDP-43–HA, generating pLVX-EF-1α-TDP-43–HA-3′ UTR vector.

### SH-SY5Y culture and shRNA testing

SH-SY5Y human neuroblastoma cell line (Sigma 94030304) was cultured in DMEM/F12 (Gibco 11330032) supplemented with 1× GlutaMAX (Gibco 35050-061), 1× Penicillin/Streptomycin (Sigma P4333-100ML) and 15 % heat-inactivated FBS (Gibco 10270-106).

Cells (200,000 per well of a 6-well plate) were plated. Cells were transduced the day after and each lentivirus was mixed with spent medium (500 ng (GV) ml$^{-1}$, 1 ml total). Three wells were transduced individually for each shRNA-coding lentivirus (2–4 shRNA sequences per target or 1–2 per nontargeting shRNA were tested; see cloning for details). Medium was changed the day after transduction and cells were then collected 4 days post-transduction in RIPA buffer (500 µl per well) containing 2 tablets of cOmplete EDTA free protease inhibitor cocktail (Roche 11873580001) and 1 tablet of PhosSTOP (Roche 0490684500) per 20 ml of buffer.

### Dual luminescence assay and 3′ UTR reporter plasmids

The in-house generated full-length human *TARDBP* 3′ UTR or human *NPTX2* 3′ UTR (Origene SC213552) were cloned into psiCHECK2-let-7 wild type (Addgene 78260), substituting the let-7 wild-type site (using HiFi Kit) to generate psiCHECK2 3′ UTR reporters used in the dual luminescence assay.

HEK293T cells were plated at 7,500 cells per well in OHN medium (details above) into a white well/bottom Nunc 96-Well plates (Thermo Scientific 136102). The day after, the cells were transduced either with psHalo-shTDP-43 (or NT shRNA) for TDP-43 knockdown or with mTRE-TDP-43–HA (ON or OFF) for TDP-43 overexpression or with pshTDP-sh-endo-*TARDBP* (or NT shRNA) to knock down endogenous TDP-43 and temporarily overexpress wild-type TDP-43–HA or RRMm TDP-43–HA (see vector details above). Twenty-four hours later, 20 ng per well of psiCHECK 3′ UTR pDNA reporters were transfected using Lipofectamine 2000 (Invitrogen 11668019; 0.5 µl per well final) while simultaneously inducing the mTRE-driven TDP-43–HA variants by 1 µg ml$^{-1}$ of DOX (Clontech 631311) or leaving them OFF as a control. Medium was exchanged completely the following day adding 78 µl of fresh OHN medium (with or without DOX). Finally, the following day, all conditions were subjected to a lytic Dual-Glo Luciferase Assay System (Promega E2920) according to the manufacturer instructions, luminescence was read using Tecan Infinite M Plex reader and *Renilla*:firefly luminescence ratios were calculated for each well.

Statistical analysis was performed in Prism (GraphPad San Diego, CA, USA) and unpaired t-test for each lentivirus and 3′ UTR reporter pair was applied on the datasets.

### Transmission electron microscopy

SNS pellet was prepared as mentioned above and submitted to the imaging facility (ZMB) of UZH. In brief, SNS pellets were resuspended in 2× fixative (5% glutaraldehyde in 0.2 M cacodylate buffer) and fixed at room temperature for 30 min. Samples were then washed twice with 0.1 M Cacodylate buffer before embedding into 2% Agar Nobile.

Post-fixation was performed with 1% Osmium for 1 h on ice, washed three times with ddH$_2$O, dehydrated with 70% ethanol for 20 min, followed by 80% ethanol for 20 min, 100% for 30 min, and finally propylene for 30 min. Propylene:epon araldite at 1:1 was added overnight followed by addition of epon araldite for 1 h at room temperature. Sample was then embedded via 28 h incubation at 60 °C. The resulting block was then cut into 60-nm ultrathin sections using an ultramicrotome. Ribbons of sections were then put onto the TEM grid and imaged on the TEM-FEI CM100 electron microscope.

## Fixation, immunofluorescence and imaging of iNets

iNets were fixed with pre-warmed 16% methanol-free formaldehyde (Pierce 28908) pipetted directly into the culture medium, diluted to 4% final formaldehyde concentration, and incubated for 15 min at room temperature. Cells were then washed once with PBS (Gibco 10010015) for 10 min, once with PBS with 0.2% Triton X-100 (Sigma T9284) washing buffer for 10 min and then blocked with 10% normal donkey serum (Sigma-Aldrich S30-M) and 0.2% Triton X-100 in PBS blocking buffer filtered via stericup (Millipore S2GPU02RE) for 30 min at room temperature. Primary antibodies (Supplementary Table 10) were then diluted in blocking buffer and left incubating overnight at 4 °C on an orbital shaker. Cells were then washed 3 × 15 min in washing buffer at room temperature and secondary antibodies (Supplementary Table 10) were then diluted in blocking buffer and incubated for 1.5 h at room temperature. Cells were then again washed 3 × 15 min in washing buffer at room temperature with DAPI (Thermo Scientific 62248) diluted to 1 µg ml$^{-1}$ in the final washing buffer wash. Cells were finally washed 1 × 15 min in PBS at room temperature, and PBS was then added to the wells to store the stained cells at 4 °C.

Stained iNets were imaged using GE InCell Analyzer 2500 HS wide-field microscope (40× air objective; 2D acquisition; 182 fields of view per well; 50 µm separation to avoid counting cells twice) or high content scanner MD ImageXpress Confocal HT.ai (40× Water Apo LambdaS LWD objective; 50 µm spinning disk; 2,048 × 2,048 pixels; 30 z-steps per stack and 0.3 µm step size) for quantification or with Leica SP8 Falcon inverted confocal for high-power, high-resolution microscopy (63× oil objective; 2,096 × 2,096 pixels at 0.059 µm per pixel, approximately 20–30 z-steps per stack and 0.3 µm step size). Laser and detector settings were kept the same for each staining combination and all imaged conditions. Huygens professional (Scientific Volume Imaging) was then used to deconvolute the stacks (from SP5 and SP8 confocals) and the deconvoluted images were further post-processed in Fiji[65] to produce flattened 2D images (z-projection) for data visualization. Immunofluorescence experiments were repeated twice as independent experiments with biologically independent samples.

For quantification of NeuN- and Ki67-positive cells, images were acquired using Leica SP5 inverted confocal with 63× oil objective (1,024 × 1,024 pixels at 1.7× zoom) and cells were counted and normalized to the total number of cells (total number of DAPI-positive nuclei).

Wide-field image quantification was done using trained ilastik[66] algorithms to segment the pixels (positive vs background) of TDP-43–HA, TDP-43$^{p403/404}$ and DAPI staining. Similarly, ImageXpress acquired images (maximum projections of z-stacks) were segmented for TDP-43–HA, NPTX2, MAP2 and DAPI staining. Segmented pictures (182 images per well, per channel or 36 for ImageXpress) were then exported and the total number of DAPI-positive nuclei (and MAP2$^+$ cells for ImageXpress) was quantified in Fiji[65] via batch processing using a custom macro, which is available upon request. TDP-43–HA or TDP-43$^{p403/404}$-positive neurons were counted manually. Nuclear intensity measurements of TDP-43–HA expressed from TRE or EF-1α promoters (with or without 3′ UTR) were performed using a custom Cell Profiler (4.2.5) segmentation pipeline on flattened z-stacks acquired on ImageXpress. Statistical analysis was performed in Prism (GraphPad) and unpaired t-tests, one-way or two-way ANOVA followed by Tukey's multiple-comparison test was applied on the datasets (see figure legends for details).

## Biochemical analysis of protein aggregates using SarkoSpin

A modified SarkoSpin protocol[14,67] was used. iNets were collected in 200 µl of cell lysis buffer (0.5% sarkosyl (Sigma L5125-100G) + 0.5 µl Benzonase (Millipore E1014-5KU)) and 2 mM MgCl$_2$ in 1× HSI buffer (20 ml of 1× HSI: 10 mM TRIS, 150 mM NaCl, 0.5 mM EDTA, 1 mM DTT + 2 tablets of cOmplete EDTA free protease inhibitor cocktail (Roche 11873580001) and 1 tablet of PhosSTOP (Roche 0490684500)). Resuspended samples were transferred into low-protein-binding Eppendorf tubes. From the obtained total cell lysate, 30 µl was used for total blots, 5 µl was used for protein quantification and the remaining sample was further processed and analysed by SarkoSpin. To 170 µl of the remaining cell homogenate, 178 µl 2× HSI with 4% sarkosyl and 52 µl 1× HSI (to obtain final sarkosyl concentration of 2% in total 400 µl volume) was added. The samples were incubated at 37 °C, 600 rpm for 45 min (Thermomixer, Eppendorf), and after increasing the volume by addition of 200 µl 1× HSI, the samples were vortex mixed and centrifuged at 21,200g on a benchtop centrifuge (Eppendorf) for 30 min at room temperature. Supernatants (~450 µl) were transferred to a new tube and the remaining supernatant was carefully removed to leave the pellets completely dry. The latter were then resuspended in 80 µl of 0.5% sarkosyl 1× HSI and analysed by immunoblot as described below. SarkoSpin was independently performed twice using biologically independent samples.

## SDS–PAGE and western blotting

For total cell lysates (both SH-SY5Y and iNets) and lysates of SNS pellets (of iNets), protein concentration was adjusted using Pierce 660 nm Protein Assay Reagent (22660, all reagents from ThermoFisher unless otherwise stated) and 5–10 and 20 µg (total lysates and SNS pellets, respectively) of total protein per lane was used for immunoblots. Samples were resuspended in final 1× LDS loading buffer (NP0007) with 1× final Bolt sample reducing agent (B0009), denatured at 70 °C for 10 min, and loaded on Bolt 12% (SNS) or 4–12% (total and SarkoSpin iNets fractions) Bis-Tris gels (NW04122BOX, NW04127BOX, NW00122BOX, NW00125BOX) using Bolt MES running buffer (B0002) and Bolt Antioxidant (BT0005) and run at 115 V. Gels were transferred onto nitrocellulose membranes using iBlot 2 Transfer NC Stacks (IB23001) with iBlot 2 Dry Blotting System (IB21001) using P0 transfer method. Membranes were then blocked with 5% w/v non-fat skimmed powder milk in 0.05% v/v Tween-20 (Sigma P1379) in PBS (milk PBST) and probed with primary antibodies (Supplementary Table 10) overnight in PBST with 1% w/v milk, washed three times with PBST, followed by incubation with HRP-conjugated secondaries (Supplementary Table 10) in 1% milk PBST. After three washes in PBST, immunoreactivity was visualized by chemiluminescence using SuperSignal West Pico or Femto Chemiluminescent Substrate (PierceNet 34077, 34096) on Amersham Imager 600RGB (GE Healthcare Life Sciences 29083467) or Fusion FX6 EDGE imager. SarkoSpin western blots were independently performed twice using biologically independent samples. All uncropped images of western blots shown in main and extended data figures can be found in Supplementary Figs. 4 and 5.

## Two-photon calcium imaging

iNets were bolus loaded with the AM ester form of Oregon Green BAPTA-1[68] by application of 5 µM solution in HBSS with Ca$^{2+}$/Mg$^{2+}$ prepared from 500 µM stock solution[69] according to manufacturers' instructions. Shortly, iNets that had been in differentiation for 12 weeks were briefly washed twice with pre-heated HBSS and then the 5 µM of Oregon Green BAPTA-1 solution was added and incubated for 1 h at 37 °C. Cells were then washed briefly twice with pre-heated HBSS and subsequently allowed to stabilize at room temperature for 30 min before imaging. Two-photon calcium imaging was performed on a Scientifica Hyperscope with a Ti:sapphire laser (Coherent; ~120 fs laser pulses) tuned to 840 nm. Fluorescence images of 256 × 256 pixels

at 4.25 Hz were collected with a 16× water-immersion objective lens (Nikon, NA 0.8). Data acquisition was performed by ScanImage.

Calcium imaging data were imported and analysed using custom-written routines in ImageJ and MATLAB, which are available upon request. First, fluorescence image time series for a given region were concatenated. The concatenated imaging data was then aligned using TurboReg to correct for small $x$–$y$ drift (alignment on OGB-1 image series, NIH ImageJ; based on Thévenaz et al. [70]). As a next step, the background was subtracted as the bottom first percentile fluorescence signal of the entire image. Average intensity projections of the imaging data were used as reference images to manually annotate regions of interest (ROIs) corresponding to individual neurons. Calcium signals were expressed as the mean pixel value of the relative fluorescence change $\Delta F/F = (F - F_0)/F_0$ in a given ROI. $F_0$ was calculated as the bottom 10th percentile of the fluorescence trace.

## Electrophysiology

Patch-clamp recordings were made on a SliceScope 2000 microscope (Scientifica) using a HEKA EPC10/2 USB amplifier (HEKA Elektronik). Data were low-pass filtered at 10 kHz and sampled at 200 kHz. Patch pipettes were pulled from borosilicate glass (Science Products) using a P-97 Puller (Sutter Instruments). Pipettes had open-tip resistances of 4–8 MΩ when filled with intracellular solution containing (in mM): potassium gluconate 150, NaCl 10, HEPES 10, magnesium ATP 3, sodium GTP 0.3, BAPTA 0.2. A liquid junction potential of +13 mV was corrected for. The extracellular solution contained (in mM): NaCl 135, HEPES 10, glucose 10, KCl 5, CaCl$_2$ 2, MgCl$_2$ 1. Series resistance ranged between 7–22 MΩ and was compensated online by 50–80% with 10 µs lag. All recordings were performed at room temperature (23–25 °C). Resting membrane potential was measured in current-clamp after establishing the whole-cell configuration. To examine action potential firing and membrane properties, current injection protocols with incrementing amplitude (duration, 500 ms; step, ±10 pA) were applied from a holding voltage of approximately −80 mV (achieved by tonic current injection). Input resistance was calculated from the steady-state voltage deflection elicited by hyperpolarizing current injections of −10 pA. Membrane time constant was estimated from exponential fits to the voltage decay.

To record voltage-dependent ionic currents, cells were voltage-clamped at a holding potential of −80 mV. Currents were elicited by 100-ms voltage steps from −80 mV to +40 mV in steps of 10 mV and corrected for leak and capacitance currents using the P/5 method. For analysis of currents, cells were included if remaining series resistance was <10 MΩ. Data were analysed with custom-written routines in Igor Pro (Wavemetrics), which are available upon request.

## HD-MEAs

CMOS-based HD-MEAs[30] were used to record the extracellular action potentials of iCoMoNSC-derived human neural networks. The HD-MEA featured 26,400 electrodes, organized in a 120 × 220 grid within a total sensing area of 3.85 × 2.10 mm². The electrode area was 9.3 × 5.45 µm², and the centre-to-centre electrode distance (pitch) was 17.5 µm, which allowed for recording of cell electrical activity at subcellular resolution. Up to 1,024 electrodes could be simultaneously recorded from in user-selected configurations. The HD-MEA featured noise values of 2.4 µV$_{rms}$ in the action potential band of 0.3–10 kHz and had a programmable gain of up to 78 dB. The sampling frequency was 20 kHz.

**HD-MEA recordings.** The recording setup was placed inside a 5% CO$_2$ cell culture incubator at 37 °C. Recordings were performed using the 'Activity scan assay' and 'Network assay' modules, featured in the Max-Lab Live software (MaxWell Biosystems), as previously described[32]. The spontaneous neuronal activity across the whole HD-MEA was recorded using 6,600 electrodes at a pitch of 35 µm in 7 electrode

configurations for 120 seconds. The most "active" 1,024 electrodes were then used to record network electrical activity for 300 seconds. Active electrodes were identified based on their firing rate, and, among those, the 1,024 electrodes featuring the highest firing rates were selected.

**HD-MEA metrics.** We used metrics similar to those described in Ronchi et al. [32] to characterize and compare the neuronal cultures; we used network, single-cell and subcellular-resolution metrics. As network metrics (Extended Data Fig. 3c) we used the burst duration (BD), inter-burst interval (IBI), inter-burst interval coefficient of variation (IBI cv)[32]. As single-cell metrics (Extended Data Fig. 3d) we used the mean firing rate (MFR), mean spike amplitude (MSA), and the inter-spike interval coefficient of variation (ISI cv)[32]. Additionally, we included the following extracellular waveform metrics (Extended Data Fig. 3e), extracted from SpikeInterface[33], an open-source Python-based framework to enclose all the spike sorting steps:

(1) Half width half maximum (HWHM), half width of trough of the action potential wave at half amplitude.
(2) Peak-to-trough ratio (PTr), ratio of peak amplitude with respect to amplitude of trough.
(3) Peak to valley (PtV), time interval between peak and valley.
(4) Repolarization slope (RepS), slope between trough and return to baseline.
(5) Recovery slope (RecS), slope after peak towards recovery to baseline.

As subcellular-resolution metrics, we extracted the action potential propagation velocity[32] (Vel) and the axon branch length (BL).

The percentage of active electrodes (AE) was also computed to measure the overall number of electrodes that could detect action potentials[32].

**HD-MEA data analysis.** Data analysis was performed using custom-written codes in MATLAB R2021a and Python 3.6.10, which are available upon request.

Spike sorting was performed to identify single units in the extracellular recordings. We used the Kilosort2[71] software within the SpikeInterface[33] framework and the corresponding default parameters. We automatically curated the spike sorting output using the following parameters:

(1) Inter-spike interval violation threshold (ISIt) = 0.5. The ISIt takes into account the refractory period, which follows every action potential. The assumption is that if two action potentials occur within a too short time interval, they most probably come from two different neurons.
(2) Firing rate threshold (FRt) = 0.05. The FRt sets the minimum firing rate of a neuron to be considered as a 'good' unit.
(3) Signal-to-noise ratio threshold (SNRt) = 5. The SNRt takes into account the ratio between the maximum amplitude of the mean action potential waveform and the noise characteristics of the specific channel.
(4) Amplitude cutoff (ACt) = 0.1. The ACt takes into account the false-negative rate—that is, the fraction of spikes per unit with an amplitude below the detection threshold.
(5) Nearest-neighbours hit rate (NNt) = 0.9. After computing the principal component for a unit, the NNt is used to check on the fraction of the nearest neighbours that fall into the same cluster.

**HD-MEA statistical analysis.** Statistical comparisons to compare samples of more than two populations were performed using the Kruskal–Wallis H test. In case the null hypothesis was rejected, we performed a post-hoc Dunn test with Sidák correction for multiple comparisons (Dunn–Sidák multiple-comparison test).

Statistical analysis was performed in MATLAB R2021a.

## HD-MEA plating

HD-MEAs were sterilized for 40 min in 70% ethanol. HD-MEAs were washed three times with sterile deionized water ($dH_2O$) and then dried. After sterilization, HD-MEAs were soaked in 20 µl of poly-D-lysine solution (P6407, Sigma-Aldrich) (diluted to 50 mg ml$^{-1}$ in $dH_2O$) for 1 h at room temperature to render the surface more hydrophilic; each HD-MEA was then washed three times with sterile $dH_2O$ and dried. Thereafter, HD-MEAs were coated with 10 µl Matrigel (354234, Corning), previously diluted in a plating medium (see below) at a 1:10 ratio and incubated at 37 °C for 2 h. iNets dissociated at 1.5, 3 and 7.5 months of age were centrifuged at 188$g$ for 5 min, supernatant was aspirated, and plating medium was added to the cell pellet to reach the desired cell density (200,000 cells per HD-MEA). Matrigel was aspirated from HD-MEAs and cells were plated in a 10 µl drop on the HD-MEAs. Medium (0.9 ml) was added after 2 h of incubation at 37 °C. 50% of plating medium was exchanged one day post plating and subsequently twice a week. Plating and culture medium was almost identical to maturation medium except that it used BrainPhys (05790, Stem Cell Technologies) as a base medium and was further supplemented with 1% penicillin/streptomycin (P4333-100ML, Sigma-Aldrich) to provide protection to cells that had to be periodically moved from culture incubator to recording culture incubator.

## Primary mouse hippocampal neuronal cell culture

Primary neuronal cell cultures were prepared from mouse embryos (E16/17). All animal experimentation, including mouse housing and breeding was done in accordance with the Swiss Animal Welfare Law and in compliance with the regulations of the Cantonal Veterinary Office, Zurich (current approved license ZH169/2022, valid until 16.02.2026). Pregnant C57BL/6 females were delivered from Janvier Labs (France) at day 12 of pregnancy and housed in accordance with the Swiss Animal Welfare Law and in compliance with the regulations of the Cantonal Veterinary Office of Zurich at LASC Irchel, Zurich. In detail, mice were housed in IVC type 2 long (T2L) cages with an enriched environment that consisted of bedding, a red mouse house, tissues and crinklets (sizzled paper). The room temperature was between 21 °C and 24 °C with a humidity level between 35% and 70%. A 12 h light/12 h dark cycle was used. Pregnant females were euthanized in their home cage with $CO_2$ prior to extracting embryos. $CO_2$ euthanasia was chosen over other methods because it is considered rapid, highly successful, safe and easy for the operator and, in our case, compatible with downstream experimental procedures (no physical damage is inflicted to any organs of interest). The fetuses were then extracted and euthanised by decapitation using scissors, which is considered the most rapid and effective method for fetuses. Five embryos per litter were used. Hippocampi were isolated on ice from each embryo's brain, pooled together and digested with trypsin ((Gibco 15400-054) 0,5% w/v supplemented with 4% w/v D-glucose (Sigma G8769)) for 15 min at 37° degrees. Following digestion, trypsin was quenched with horse serum (Sigma H1138) and the tissue was manually dissociated until clumps were not visible anymore. Cells were spun at 120$g$ for 5 min and plated on poly-D-lysine (Sigma P6407) coated glass-bottom 8-well-IBIDI chamber slide (80807; 4 wells per experimental condition) in Neurobasal medium (Gibco 21103049) supplemented with GlutaMAX (ThermoFisher 35050061), 2% B27+ supplement (Gibco 17504-044), 100 U penicillin-streptomycin (Sigma P4333-100ML) and 4% w/v D-glucose (Sigma G8769). To transduce the primary neuronal cultures, TDP-43–HA or GFP–FUS mTRE lentivirus vectors were used at 500 ng (GV) per ml of spent culture medium in a total 100 µl at DIV 11 and the medium was fully exchanged the following day (DIV 12), supplemented with 1 µg ml$^{-1}$ of DOX (Clontech 631311), thereby inducing the expression of transgenes. Cells were fixed at DIV 17.

## Mycoplasma check

All cycling cells used in the study were routinely checked for possible mycoplasma contamination using EZ PCR Mycoplasma Detection Kit (Sartorius 20-700-20). Young iNets (with some residual KI67+ cells) were sporadically checked. No mycoplasma contamination was detected.

## Patient post-mortem brain immunofluorescence

Formalin-fixed, paraffin-embedded hippocampal, frontal or primary motor cortex patient (ALS, FTLD-TDP-A, FTLD-TDP-C, FTLD-FUS, FTLD-tau, Alzheimer's disease) sections (see Supplementary Table 9) were used. All FTLD and Alzheimer's disease tissue samples were donated to Queen Square Brain Bank for Neurological Disorders at UCL Queen Square Institute of Neurology with full, informed consent. Anonymized autopsy ALS sample was collected by the Institute of Neuropathology at UZH. According to Swiss law, anonymized autopsy tissues do not fall within the scope of the Human Research Act and may be used in research. Sections were deparaffinized in three Xylene rounds (5 min each) and rehydrated in decreasing ethanol washes (2× 100% for 10 min; 2× in 95% for 5 min; 80, 70 and 50% for 5 min each) and finally submerged in MilliQ water for 10 min. Antigen retrieval was then performed by microwave heat treatment in sodium citrate buffer (0.01 M, pH 6.0). Sections were then cooled down on ice for 10 min and once quickly washed in PBS, followed by 3x PBS washes for 5 min at room temperature before blocking with blocking buffer (5% normal donkey serum (Sigma-Aldrich S30-M), 3% BSA (Sigma A4503) and 0.25% Triton X-100 (Sigma T9284) in PBS) for 30 min at room temperature. Primary antibodies (Supplementary Table 10) were then diluted in blocking buffer and 300 µl of the antibody mix was evenly put per slide/section. Slides were put into a wet and dark incubation chamber and incubated overnight at room temperature. Slides were rinsed once in PBS and then washed 3× 5 min in PBS on a shaker. Secondary antibodies (Supplementary Table 10) diluted in blocking buffer, centrifuged for 30 min at 15,000$g$ at room temperature and 500 µl of the antibody mix was evenly put per slide/section and put into the incubation chamber to incubate for 2.5 h at room temperature. Slides were then rinsed once in PBS, washed 1× 5 min in PBS on a shaker and to stain the nuclei (DNA), 500 µl of DAPI solution (Thermo Scientific 62248; diluted to 1 µg ml$^{-1}$ in PBS) was evenly added onto the slide/section and incubated for 10 min at room temperature in the incubation chamber. Slides were rinsed once in PBS, washed 2×5 min in PBS on a shaker, followed by Sudan Black (0.2% in 70% ethanol; Sigma 199664) autofluorescence quench for 10 min at room temperature on shaker. Slides were rinsed 6 times in PBS and left washing in PBS until mounted and coverslipped in ProLong Diamond Antifade Mountant (ThermoFisher P36961) and then left to dry in the chemical cabinet at room temperature in the dark. Mounted sections were stored at 4 °C in the dark.

Stained patient brain sections were imaged using Leica SP8 Falcon inverted confocal for high-power, high-resolution microscopy (63× oil objective; 1.7 or 3× zoom; 2,096 × 2,096 pixels at 0.059 µm per pixel or 1,848 × 1,848 pixels at 0.033 µm per pixel, respectively, approximately 20 $z$-steps per stack at 0.3 µm). White laser and HyD detector settings were kept the same for each staining combination and all imaged conditions. Huygens professional (Scientific Volume Imaging) was used to deconvolute the stacks and the deconvoluted images were further post-processed in Fiji[65] to produce a 3D projection for data visualization (the first image of the 3D projection is shown in figures). Post-processing settings were kept the same for all images of all sections from all donors (except for DAPI and MAP2 channels for certain sections when higher brightness settings were used to reach intensity that allowed proper visualization).

## Cell isolation for scRNA-seq

Duplicates (except for TDP-43–HA OFF and TDP-43–HA 2 weeks samples) of iNets (young (1.5 months), middle stage (3 months) and old

(7.5 months)) and TDP-43–HA experiment samples at middle stage, were dissociated into single-cells suspension using Papain Dissociation System (Worthington LK003150), passed through 70-µm and 40-µm cell strainers (Falcon 07-201-431 and 07-201-430), and resuspended in HIB++ medium (Hibernate-E Medium medium (Gibco A1247601) supplemented with EDTA (1 mM final; Invitrogen AM9260G), HEPES (10 mM final; Gibco 15630080), with 1× B27+ supplement (Gibco 17504-044), 1× N2 supplement (Gibco 17502-048); 1× GlutaMAX (Gibco 35050-061), BDNF (PeproTech 450-02), GDNF (Alomone labs G-240), CNTF (Alomone labs C-240), NT-3 (PeproTech 450-03) and IGF-1 (Stem Cell 78022) all at 20 ng ml$^{-1}$) to 1,000 cells per µl using CASY Cell Counter (Innovatis).

## scRNA-seq using 10X Genomics

The quality and concentration of the single-cell preparations were evaluated using an haemocytometer in a Leica DM IL LED microscope and adjusted to 1,000 cells per µl. Ten-thousand cells per sample were loaded in to the 10X Chromium controller and library preparation was performed according to the manufacturer's indications (Chromium Next GEM Single Cell 3′ Reagent Kits v3.1 protocol). The resulting libraries were sequenced in an Illumina NovaSeq sequencer according to 10X Genomics recommendations (paired-end reads, R1 = 26, i7 = 10, R2 = 90) to a depth of around 50,000 reads per cell. The sequencing was performed at Functional Genomics Center Zurich (FGCZ).

## Single-cell data analysis

Data was processed with CellRanger for demultiplexing, read alignment to the human reference genome (GRCh38) and filtering to generate a feature-barcode matrix per sample. Cell doublets were removed with scDblFinder[72] and outlier cells were detected and filtered with the scater R package[73]. In short, cells with more than 3 median-absolute-deviations away from the median number of UMIs, the number of features and the percentage of mitochondrial genes were removed. For the 7.5-month-old samples, we additionally filtered cells with fewer than 2,000 UMIs and fewer than 1,500 detected features. For the TDP-43 overexpression experiment, we additionally filtered cells with fewer than 5,000 UMIs and fewer than 2,500 detected features.

Seurat v3[74] was used for log-normalization and to identify the top 2,000 highly variable genes per sample. Louvain clustering was always performed with resolution 0.4 based on a shared nearest-neighbour graph constructed from the top 20 principal components. The UMAP[75] cell embeddings were computed from the top 20 principal components. Cell cycle scores were estimated with Seurat using a list of G2/M and S phase markers from Kowalczyk et al. [76].

Marker genes that are upregulated in one cluster compared to any other cluster were identified with the findMarkers function from the scran R package[77], which runs pairwise *t*-tests and combines the results into a ranked list of markers for each cluster. Heat maps with marker gene expression show scaled mean log counts of all cells in each cluster. Cluster 12 (TDP-43–HA overexpression experiment) was profiled with the same method against other neuronal clusters (0, 2 to 5, 7 to 11). We note that clustering scRNA-seq data followed by looking for differences between clusters uses the same data twice (that is, 'double dipping')[78]. We provide a ranked list of differential expression statistics to help cluster interpretation, but we are aware that *P* values are difficult to interpret in this setting.

The two iCoMoNSC samples were integrated with Seurat using canonical-correlation analysis[79], the data was scaled and number of UMIs and the percentage of mitochondrial UMIs was regressed out before clustering.

The scRNA-seq data from three different NSC lines[7] were downloaded from ArrayExpress (accession number E-MTAB-8379) and integrated with our iCoMoNSC data using Seurat; the data were scaled and the number of UMIs was integrated out before clustering.

The human organoid (409b2 and H9) scRNA-seq data[8] was downloaded from ArrayExpress (accession number E-MTAB-7552). Data from the iCoMoNSC and iNets (1.5, 3 and 7.5 months old) were integrated with data from cells from human brain organoids using Seurat, and scaled and the number of UMIs and the percentage of mitochondrial UMIs were regressed out before clustering. Batch effects were assessed with the smoothed cms mixing metric from CellMixS[80] using $k = 500$ neighbours and 10 dimensions in principal component analysis space.

In the four samples of the TDP-43 overexpression experiment, we quantified the expression of the TDP-43–HA construct components, including TDP-43–HA, the long terminal repeats, and the rtTA, as well as the endogenous TARDBP using CellRanger. The counts were added to the filtered data from the whole transcriptome CellRanger output. The total TDP-43 log$_2$ fold change between cluster 12 and all other neuronal clusters was computed using the summary.logFC metric from scran's findMarkers.

Pseudotime analysis was carried out with monocle3 v1.0.0[81], commit 004c096[82], rooting the trajectories in our annotated NSC cluster.

To assess the replicability of independent iNets, we reused the TDP-43 overexpression experiment cells and their clustering and cluster annotations (TDP-43–HA OFF only) and similarly the cells from the time series (middle stage), using their canonical genes (that is, excluding TDP-43–HA). We run the same downstream analysis as described above (top 2,000 highly variable genes, 20 principal components, UMAP cell embedding), without integrating the two datasets. Independently, we ran MetaNeighbor v1.8.0[83] to measure cell-type replication across experiments using the annotated clusters (with at least 10 cells) from the aforementioned joint TDP-43–HA OFF and time series middle stage cells. We report the MetaNeighbor area under the receiver operating characteristic (AUROC) values pairwise comparing clusters. These AUROC scores can be read as a probability. An AUROC of 1 depicts a perfect cluster:cluster match (so both clusters are composed by cells from the same cell type), an AUROC of 0.5 is as good as random (unlikely the two clusters belong to the same cell type), and an AUROC of 0 can be read as a strong non-match (so both clusters are clearly not the same cell type).

## Bulk RNA-seq

Middle stage iNets in 12-well plates were transduced with shRNA to either knock down TDP-43 (or control shRNA) or overexpress TDP-43–HA (or non-induced control; see 'Lentivirus preparation') in 4 replicates per condition (individually transduced wells). Two weeks later, RNA was extracted and isolated using RNeasy Plus Mini Kit (Qiagen 74134) according to manufacturer's instructions and an additional DNA digestion step was performed by RNase-Free DNase Set (Qiagen 79254). RNA QC was performed by RNA ScreenTape, revealing that the samples had a RINe value in a range of 9.1–9.8. Novaseq 6000 (Illumina) was used for cluster generation and sequencing according to standard protocol. Sequencing configuration was paired-end 150 bp, 150 million reads depth.

Raw reads were processed with ARMOR[84]. In brief, reads were aligned and counted to the human genome (GRCh38 assembly and Gencode release 43) with salmon v1.4.0 and with STAR 2.7.7a. We modelled the salmon-generated count data with quasi-likelihood (QL) negative binomial generalized log-linear models and ran differential expression analysis with edgeR v3.36.0.

## CLiP-seq

We re-analysed CLiP data from GSE27201 Polymenidou 2011[11] (mouse) including accessions SRR107031 and SRR107032; from E-MTAB-530 Tollervey 2011[13] (human) accessions ERR039847, ERR039848, ERR039843, ERR039842, ERR039844, ERR039845, ERR039846 with the nf-core/clipseq workflow v1.0.0[85] with default parameters and against the GRCh38 or GRCm38 reference genomes, as appropriate. Polymenidou

2011[11] CLiP binding sites were transformed from GRCm38 to GRCh38 coordinates using liftOver from Kent utils v390[86].

## Re-analysis of CLiP and iCLIP data

FASTQ files of TDP-43 human brain iCLIP[13] (three controls and three patients with FTLD) were downloaded from ArrayExpress (E-MTAB-530)[87]. The data were re-analysed using the GRCh38 reference genome with the nf-core/clipseq[85] for pre-processing, mapping and crosslink site identification. Similarly, FASTQ files of TDP-43 CLiP in mouse were retrieved from GSE27201 and processed using nf-core/clipseq and the GRCm38 genome. Source code is available at https://doi.org/10.5281/zenodo.8142336.

## Re-analysis of RNA-seq data from sorted nuclei

We downloaded the RNA-seq gene count table as a supplementary data file (GSE126542_NeuronalNuclei_RNAseq_counts.txt.gz) from GEO accession GSE126542. A differential gene expression analysis between TDP-43 negative and positive nuclei was performed in R version 4.0.5 with edgeR[88] taking into account the paired nature of the data using the quasi-likelihood framework[89] (empirical Bayes quasi-likelihood $F$-test). Reported false discovery rate (FDR) values are adjusted for multiple comparisons using the Benjamini–Hochberg method. Source code is available at https://doi.org/10.5281/zenodo.8142336.

## Splicing analysis to identify skipped exons

The human reference genome (GRCh38) and annotation (version 108) were obtained from ENSEMBL[90]. Illumina Trueseq adapters were removed from RNA-seq reads using cutadapt (version 4.1) with the parameters -q 25 -m 25[91]. The processed reads were mapped to the reference genome using STAR aligner (version 2.7.10b) with default parameters[92]. Subsequently, differential splicing analysis at the event level was performed using rMATS (version 4.1.2) with the parameters -t paired --readLength 100 --variable-read-length[93]. Significant differentially spliced events were defined by FDR < 0.05 and 15% change in absolute value of IncLevelDifference and intersected with a human skipped exon dataset[43] to check for potential skipped exons in our datasets. The results were processed using tidyverse package (v2.0.0) in R[94].

## Cell line identification

Cell lines (SH-SY5Y Sigma 94030304 and HEK293T ATCC CRL-3216) used in the study were checked against cross-contaminated or misidentified cell lines (https://iclac.org/databases/cross-contaminations/). No cross-contaminated or misidentified cell lines were identified. Other cells used in this study were control human early neonatal dermal fibroblasts (Gibco C0045C)—the source cells used to generate iPS cells, iCoMoNSCs and iNets in this paper; and primary mouse hippocampal neurons.

## Reporting summary

Further information on research design is available in the Nature Portfolio Reporting Summary linked to this article.

## Data availability

scRNA-seq and bulk RNA-sequencing data are available via GEO accession number GSE230647. Source data are provided with this paper.

## Code availability

Code has been deposited on Zenodo with https://doi.org/10.5281/zenodo.8142336.

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

**Acknowledgements** We gratefully acknowledge the support of the Swiss National Science Foundation (grants: 310030_192650, PP00P3_176966) and the National Centre for Competence in Research (NCCR) RNA and Disease (51NF40-182880 and 51NF40-205601) and the Swiss Foundation for Research on Muscle Diseases to M.P. M.H.-P. was supported by the Milton Safenowitz Postdoctoral Fellowship from the ALS Association (16-PDF-247), postdoctoral fellowship from the University of Zurich (FK-15-097) and the Promotor-Stiftung from the Georges and Antoine Claraz Foundation. A.H. was supported by the European Research Council (ERC) Advanced Grant 694829 neuroXscales and the corresponding proof-of-concept grant 875609 HD-Neu-Screen. V.I.W. was supported by the FEBS Long-Term Fellowship and is currently supported by the NWO Rubicon Fellowship and D.B. by the Czech Science Foundation (GACR grant no. 18-25429Y). T.L. is supported by an Alzheimer's Research UK Senior Fellowship and Queen Square Brain Bank for Neurological studies is supported by the Reta Lila Weston Institute for Neurological Studies. The funders had no role in the

experimental design, data collection, analysis, and preparation of the manuscript. The authors thank E. Yángüez, A. J. C. de Gouvea and D. Popovis for discussion and full support with scRNA-seq experiments; G. Tan and D. González Rodríguez for bioinformatic support; U. Wagner for advice on statistical analyses; G. Barmettler for technical help with TEM and Urs Greber for providing us with HEK293T cells and lentivirus packaging plasmids. All light imaging as well as TEM was performed with equipment maintained by the ZMB UZH.

**Author contributions** Conceptualization of the study was carried by M.H.-P., V.I.W., K.M.B., S.R. and M.P. M.H.-P. generated iPS cells and iCoMoNSCs and developed the iNets differentiation protocol, designed all vectors, reporters, shRNAs and experiments, carried out majority of cloning and lentiviral vector preparation, luminescence assay, biochemistry and cell and human brain immunofluorescence as well as imaging and data analysis and prepared samples for calcium imaging, patch-clamp, HD-MEA and single-cell and bulk RNA-sequencing experiments. K.M.B. and I.M. analysed the scRNA-seq data and re-analysed iCLIP and single-nuclei RNA-sequencing datasets. I.M. performed all bulk RNA-sequencing data analysis, data interpretation and visualization and provided critical input on the study. S.R. cultured iNets on the HD-MEAs, performed HD-MEA recordings, developed HD-MEA metrics and analysed all recorded HD-MEA data. V.I.W. performed human brain immunofluorescence and imaging, and helped with iNets culture, lentivirus production, single-cell RNA-sequencing sample preparation, data analysis, figure preparation and statistics. Z.M., E.-M.H. and M. Panatta performed cell immunofluorescence and imaging, F.L. contributed to biochemistry experiments and S.S. performed SNS isolation and western blots. V.H. and I.D. performed and analysed whole-cell patch-clamp recordings. E.T. cloned and provided mTRE-GFP-FUS lentivirus, prepared mouse primary hippocampal neurons and helped with RNA isolation for bulk RNA sequencing. M.P.-B. designed, tested and cloned the original TDP-43–HA sequence into the mTRE vector. B.G. helped with iNets culture and designed and cloned pshTDP-endo-TARDBP-mTRE-RRMm-TDP-43–HA vector. A.v.d.B. performed and analysed two-photon calcium imaging. P.S. helped with bulk RNA-sequencing data analysis and visualization. L.D.V. provided the full-length *TARDBP* 3′ UTR cloned from HEK cDNA. K.F., A.A. and T.L. provided human brain sections, neuropathological consultation and critical input on the study. D.B. provided critical feedback on the iCoMoNSC generation and characterization. M.D.R. provided feedback on sequencing data analyses. M.D.R., T.K., M.M., A.H., M.H.-P. and M. Polymenidou provided supervision. M.H.-P., V.I.W., K.M.B., S.R. and M.P. wrote and edited the manuscript and prepared figures. M.P. directed the entire study. All authors read, edited and approved the final manuscript.

**Funding** Open access funding provided by University of Zurich.

**Competing interests** The authors declare no competing interests.

**Additional information**
**Correspondence and requests for materials** should be addressed to Magdalini Polymenidou.

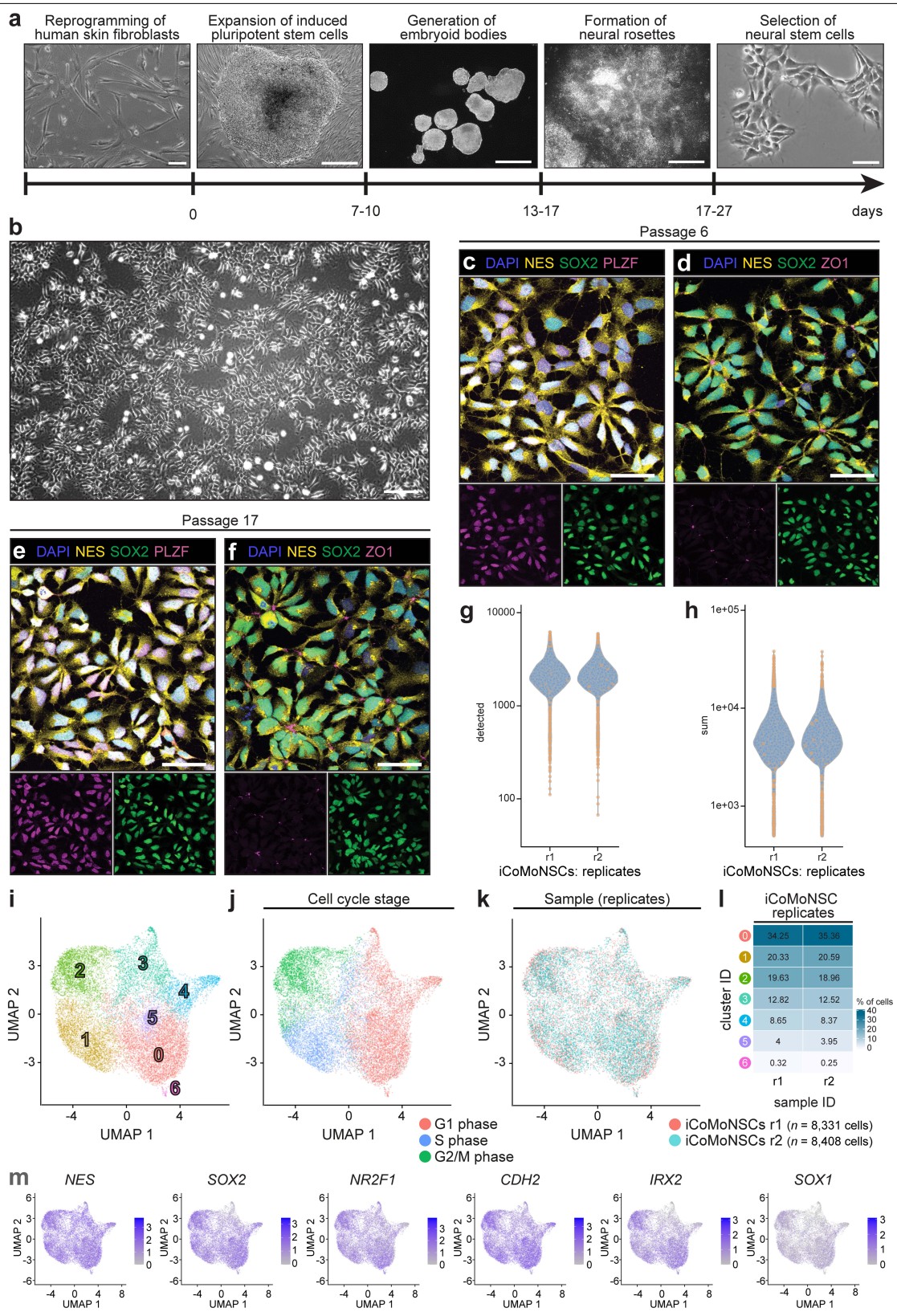

**Extended Data Fig. 1** | See next page for caption.

**Extended Data Fig. 1 | iCoMoNSCs characterization. a**, Phase contrast images from different stages of iCoMoNSC generation. Human fibroblasts (left) were reprogrammed into iPS cell colonies (2nd left), which formed embryoid bodies (middle) and then generated neural rosettes (2nd right). Patches of morphologically distinct colonies migrated out of rosettes and were isolated as clones (right). iCoMoNSC clones were successfully generated from 2 WT iPS cell clones. **b**, Phase contrast image showing the overall homogeneous morphology and pinwheel growth organization of the iCoMoNSCs (representative image from an early passage - passages were imaged regularly). NSC-marker immunofluorescence of iCoMoNSCs at early passage 6 (**c, d**) and late passage 17 (**e, f**) with NES, SOX2 and PLZF (**c, e**) or ZO1 (**d, f**) - early and late passage NSC marker IF was repeated twice. Violin plots showing the number of genes (**g**) or UMIs (**h**) detected in all iCoMoNSCs by scRNA-seq coming from replicates from two independent cell culture dishes. Blue data points are cells retained after quality control, yellow data points are cells that were filtered out. **i**, UMAP of two integrated replicate samples of iCoMoNSCs clustering into 7 clusters, (**j**) same UMAP representing cell cycle stages or (**k**) showing the replicates whose distribution across the clusters is shown in (**l**). **m**, UMAP with normalized expression of selected genes across all iCoMoNSCs. Scale bars, (**a**) from left to right: 100 µm, 500 µm, 500 µm, 500 µm, 50 µm, (**b**) 150 µm, (**c-f**) 50 µm.

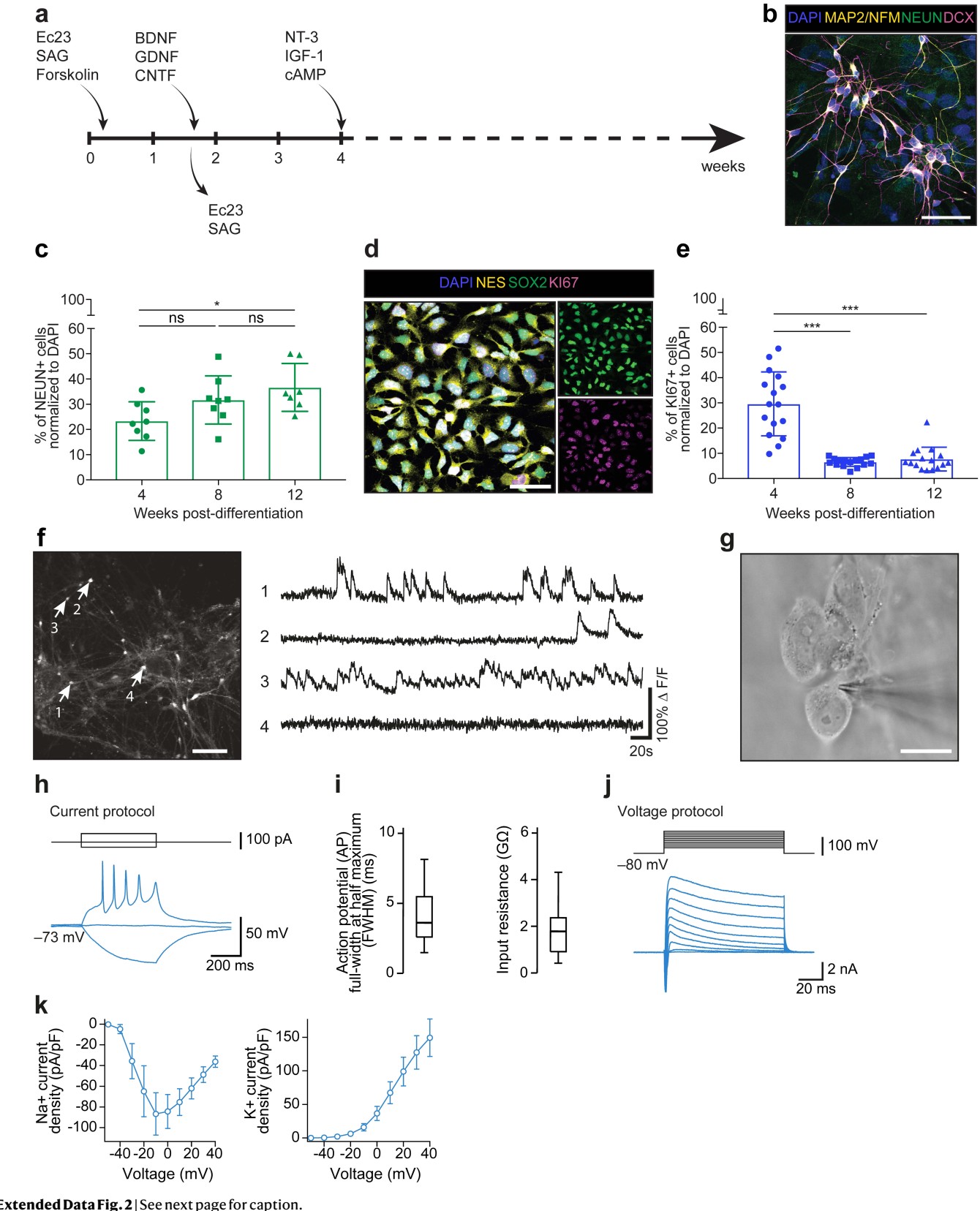

**Extended Data Fig. 2** | See next page for caption.

**Extended Data Fig. 2 | Differentiation of iCoMoNSCs. a**, Schematics of iCoMoNSCs differentiation protocol. **b**, Immunofluorescence of neuronal markers MAP2/NFM, NEUN and DCX at 4 weeks of differentiation. **c**, Quantification of NEUN-positive cells over time in differentiation reaching approx. 40% at 12 weeks. n = 8 independent fields of view normalized to DAPI$^+$ nuclei. Unpaired $t$ tests. Data are from one differentiation and are shown as mean with SEM and $p$ values are two-tailed. 4 weeks vs 12 weeks $p$ = 0.01 **d**, KI67 immunofluorescence of iCoMoNSCs at passage 6. **e**, Quantification of KI67-positive cells over time in differentiation showing a drop to 5% at 12 weeks. n = 13 independent fields of view normalized to DAPI$^+$ nuclei. Unpaired $t$ tests. Data are from one differentiation and are shown as mean with SEM and $p$ values are two-tailed. 4 weeks vs 8 and vs 12 weeks $p$ < 0.0001. **f**, 2-photon calcium imaging of iNets bolus loaded with the AM ester form of Oregon Green BAPTA-1 showed typical firing neurons (neurons 1. and 3.). Calcium imaging was repeated twice; once with 2-photon confocal, once with epifluorescent microscope. **g**, Infrared DIC image of whole-cell patch clamped human neuron at 4 months (representative image from one patch session). **h**, Example voltage responses of a neuron elicited by tonic current injection. **i**, Average action potential and input resistance. Box plots indicate distribution median value and the first and third quartiles, and whiskers extend to minimum/maximum. n = 10 individually patched neurons. Data shown are mean with SEM. **j**, Example current evoked by voltage steps from −60 mV to +40 mV (duration, 100 ms) from a holding voltage of −80 mV. Data recorded from the neuron in (**g, h**). **k**, Peak Na+ and K+ current densities plotted vs. voltage. n = 10 individually patched neurons. Data shown are mean with SEM. Scale bars, (**b,d**) 50 μm, (**f**) 100 μm, (**g**) 20 μm.

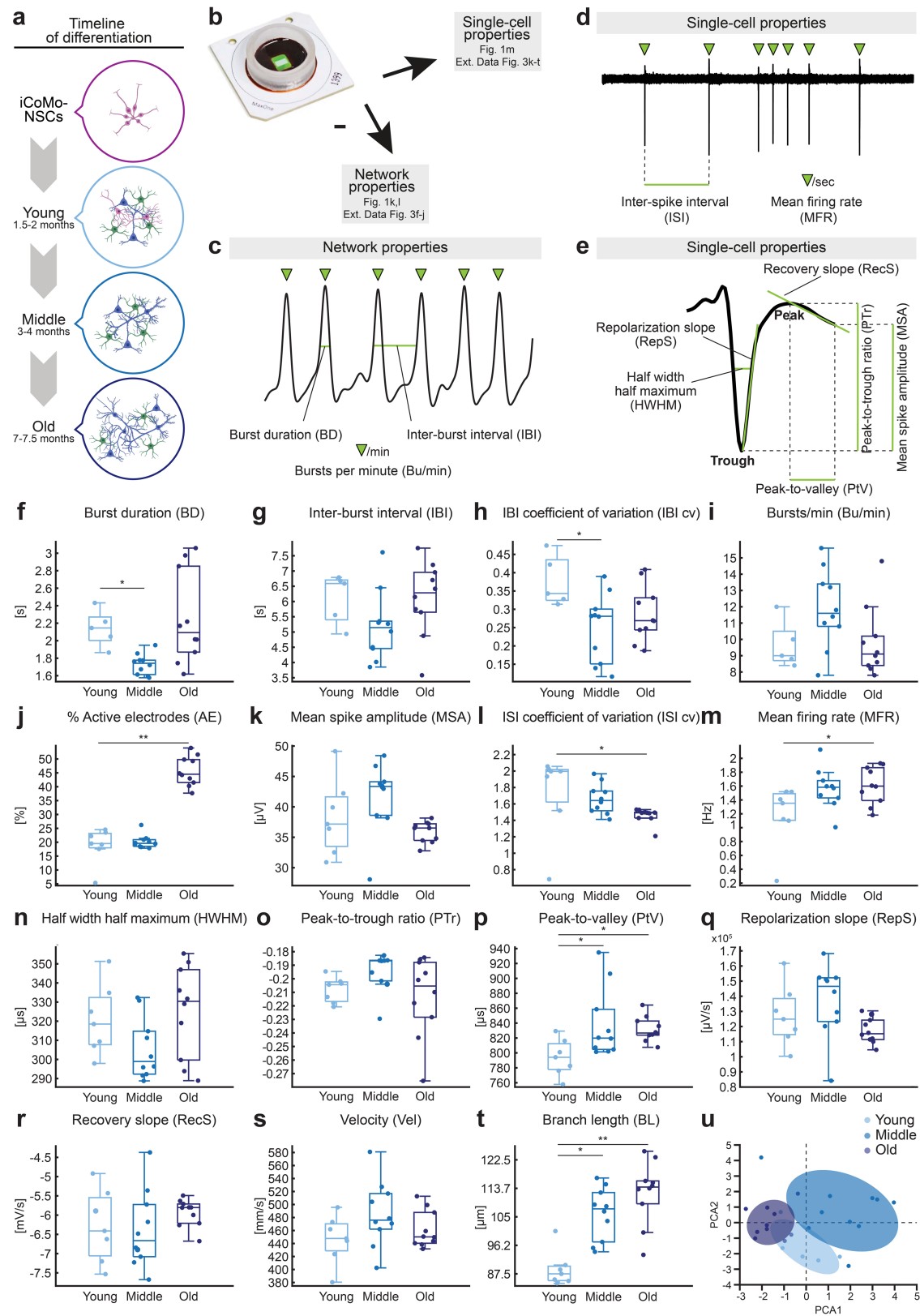

**Extended Data Fig. 3** | See next page for caption.

**Extended Data Fig. 3 | Schematics of network and single-cell properties and individual plots from data shown in Fig. 1l–n. a**, Timeline of the experiment: starting from iCoMoNSCs (0 months), iNets were differentiated, sequenced and plated on MEAs at young (1.5 months), middle (3 months) and old (7 months) stages. Created with BioRender.com. **b**, HD-MEA chip mounted on a printed circuit board featuring a cell culture chamber (ring) and the microelectrode array in the center (green). Image courtesy of MaxWell Biosystems AG, print permitted. Scale bar, 4 mm. **c**, Representative spike time histogram recorded from 1,020 electrodes used to illustrate how network metrics were extracted. **d**, Representative spontaneous APs recorded from one electrode, used to illustrate how single-cell metrics were extracted. **e**, Representative action potential (AP) used to illustrate how features were extracted from the AP shape. Panels **c-e** serve illustrative purposes only. **f-t**, Box plots representing all results illustrated in Fig. 1l, m. Each data point represents the respective metric value from one HD-MEA culture (n = 7 (young), 10 (middle), 10 (old)). Box plots indicate distribution median value and the 25th and 75th percentiles. **f**, BD Young vs Middle $p < 0.007$; **h**, IBIcv Young vs Middle $p = 0.015$; **j**, AE Young vs Old $p = 0.0005$; **l**, ISIcv Young vs Old $p = 0.004$; **m**, MFR Young vs Old $p < 0.025$; **p**, PtV Young vs Middle $p = 0.02$ and Young vs Old $p = 0.003$; **t**, BL Young vs Middle $p = 0.008$ and Young vs Old $p = 0.001$. **u**, Principal component analysis (PCA) demonstrating separation of the three maturation stages.

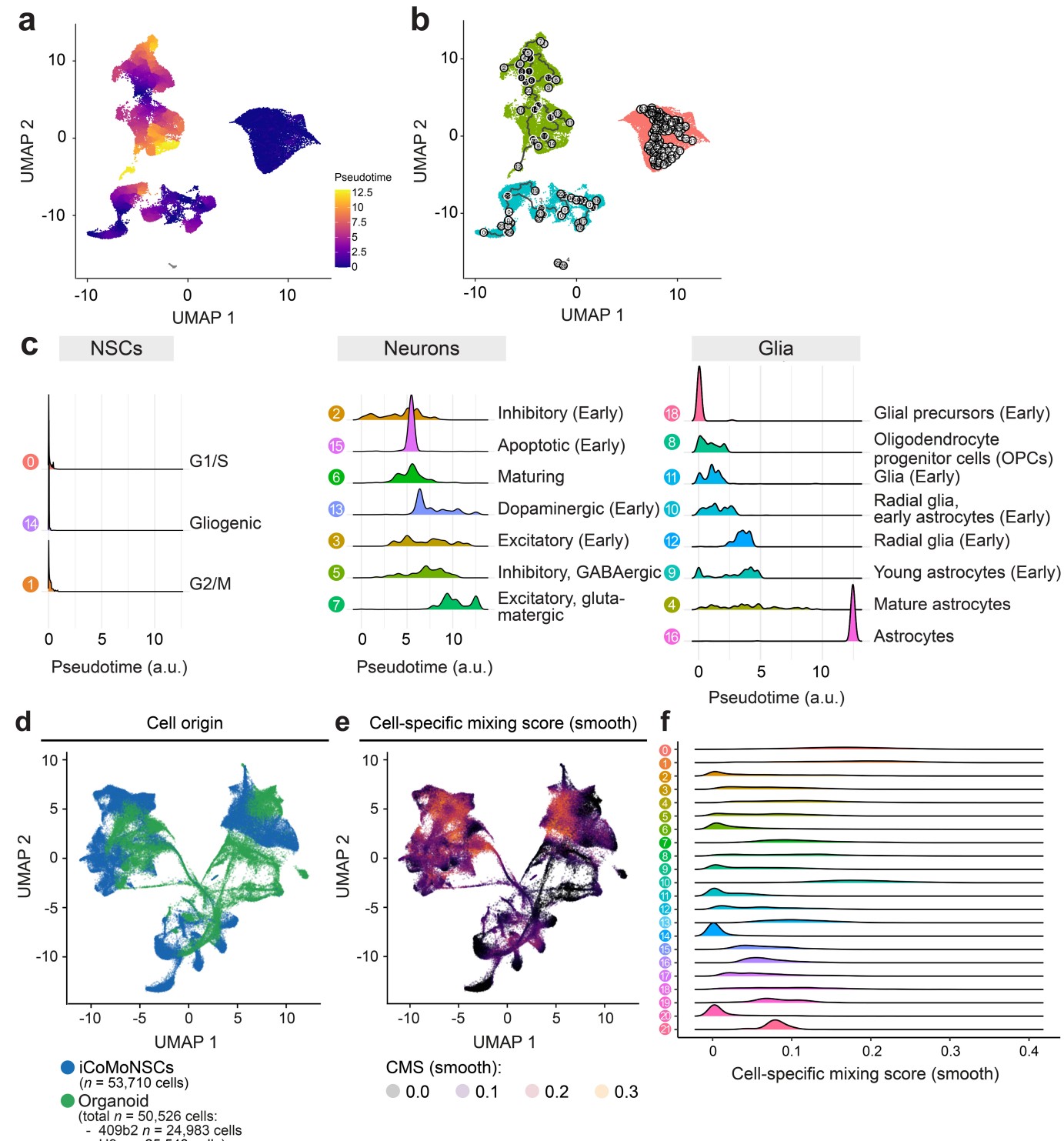

**Extended Data Fig. 4 | iNet maturation: pseudotime analysis and integration metrics with the organoid data. a**, UMAP cell embedding colored by the inferred latent pseudotime. **b**, UMAP depicting predicted pseudotime trajectories, lineage branching points, and lineage outcomes. Black lines depict the structure of the trajectory graph; black circles, branching points where cells can commit to different trajectories; gray circles, outcomes (i.e. cell fates) of the given trajectory; and colors depict cell partitions. **c**, Cluster positioning along a pseudotime rooted on NSCs. Each ridgeline depicts the distribution of pseudotimes across cells belonging to a given cell cluster. **d**, UMAP of young, middle stage and old iNets depicting the data origin after integration of iNets and organoids. **e**, Same UMAP showing the underlying batch effects using the smoothed cell-specific mixing score (cms)[80], a test for batch effects within k-nearest neighboring cells. A high cms score refers to good mixing, while a low score indicates batch-specific bias. The test considers differences in the number of cells from each batch. **f**, Smoothed cms score distribution stratified by cell cluster.

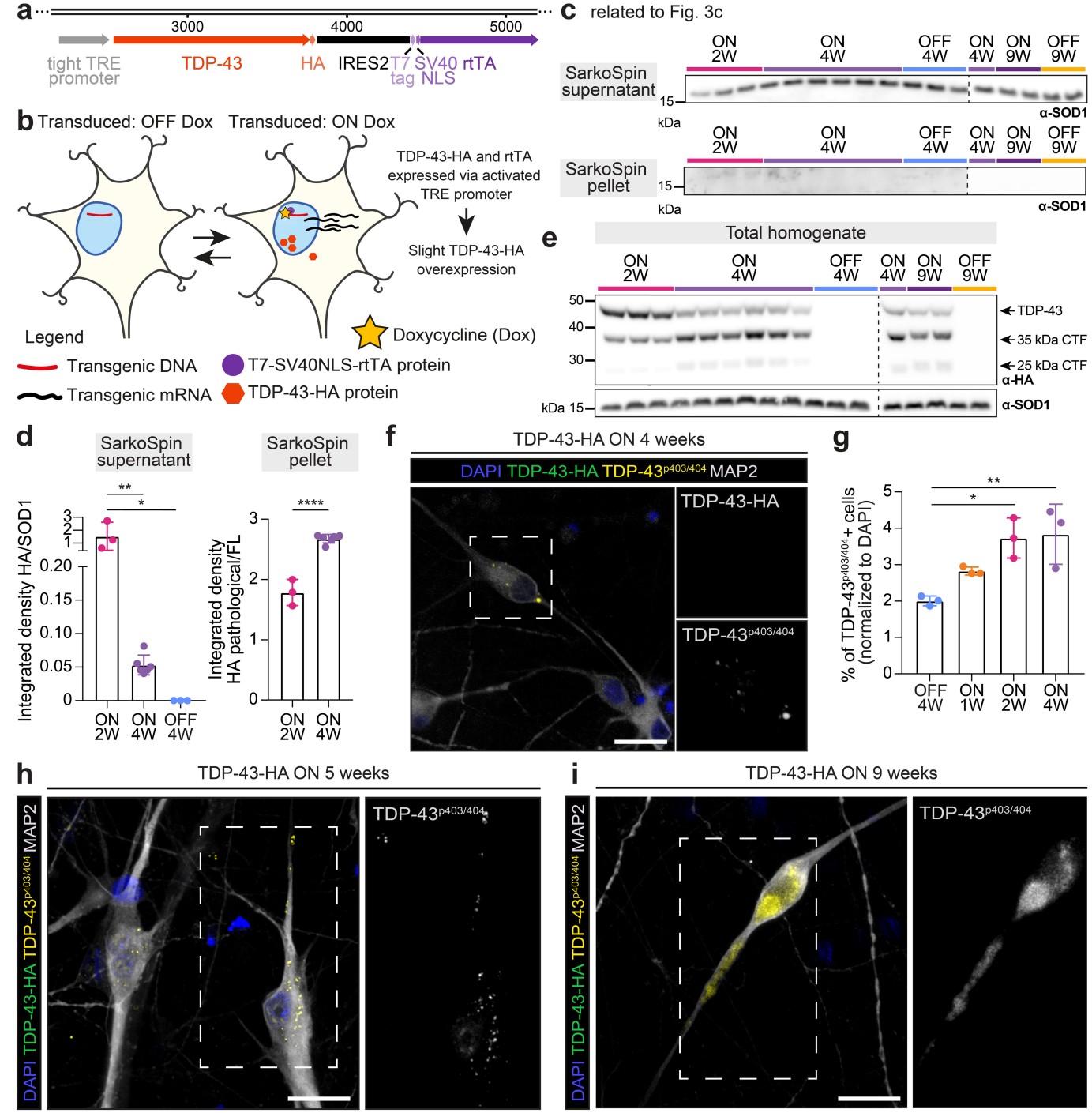

**Extended Data Fig. 5** | See next page for caption.

**Extended Data Fig. 5 | TDP-43-HA-induced pathology in iNets. a**, Schematics of our improved, all-in-one monocistronic TetON cassette, which consists of a single tight TRE (*P*tight) promoter driving inducible expression of TDP-43-HA, which is linked via IRES2 to T7-tagged rtTA with SV40 NLS fused to its N-terminus to increase its nuclear localisation, making it a nuclear marker of transgenic cells. **b**, Schematics showing DOX OFF stage (left) in which no transgenic mRNA or protein could be detected. Upon addition of doxycycline (DOX ON stage; right), the nuclear, T7-tagged rtTA binds the Tet operators in the TRE promoter, which induces overexpression of TDP-43-HA as well as leads to steady expression of rtTA needed for the whole cassette to function. The system is reversible. **c**, SOD1 Western blot (WB) of SarkoSpin fractions of iNets with or without overexpression of TDP-43-HA for 2, 4 or 9 weeks shown in Fig. 3c. **d**, Quantification of Western blots shown in Fig. 3c (unpaired *t* test; *p* values two-tailed) and Extended Data Fig. 5c (one-way ANOVA with Tukey's multiple comparison; mean of each dataset compared with the mean of every other dataset). n = 3 (2 weeks ON and 4 weeks OFF) and 6 (4 weeks ON) independently transduced wells. Data shown are mean with SD. Supernatant 2 W ON vs 4 W ON $p$ = 0.0071 and 2 W ON vs 4 W OFF $p$ = 0.0132, Pellet 2 W ON vs 4 W ON $p$ < 0.0001. Pathological = aggregated and fragmented. **e**, WB on total homogenates (SarkoSpin input). SOD1 was used as a loading control. SarkoSpin was performed from 2 independent experiments. **f**, Immunofluorescence with phospho-specific (S403/404) anti-TDP-43 antibody revealed inclusion-like structures in the soma of TDP-43-HA-negative neurons. **g**, Quantification of TDP-43$^{p403/404}$-positive cells over 4 weeks of TDP-43-HA overexpression. Data from a representative experiment (out of $N$ = 2): each data point represents a sum of all TDP-43$^{p403/404}$-positive cells counted from 182 fields of view of an independent well and normalized with DAPI. Between 3728 - 8248 cells were analyzed per data point. One-way ANOVA with Tukey's multiple comparison (mean of each dataset compared with the mean of every other dataset). Data shown are mean with SD. TDP-43$^{p403/404}$-positive inclusions were found localized to neurites at 5 weeks (**h**) and occasionally grew into-aggregate-like structures at 9 weeks of TDP-43-HA overexpression (**i**). TDP-43$^{p403/404}$ experiment was performed once. Scale bars, 25 µm.

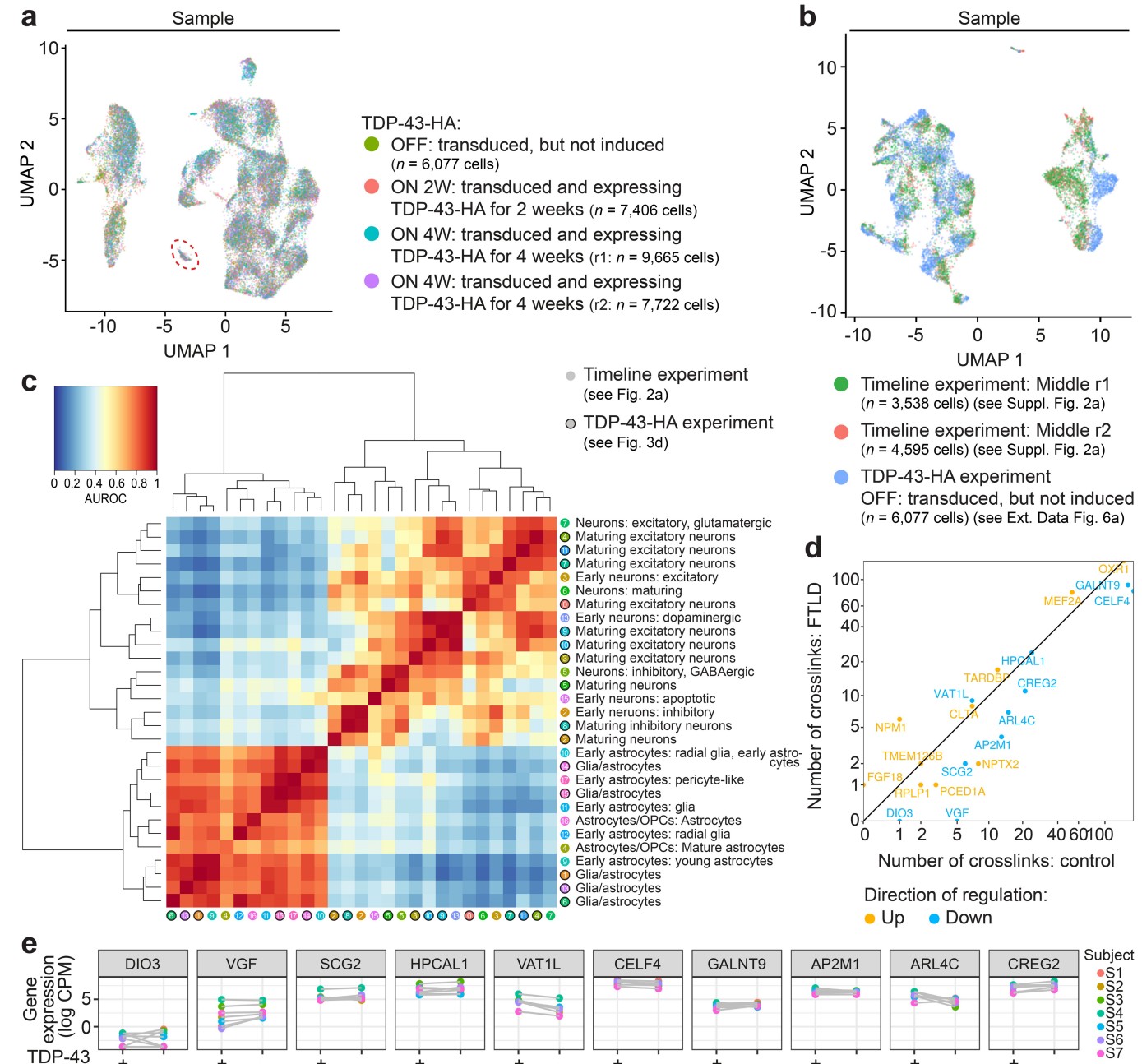

**Extended Data Fig. 6 | iNets exhibit high reproducibility and TDP-43-HA-induced pathology misregulates TDP-43 RNA targets. a**, UMAP highlighting the four samples of the single-cell RNA-seq TDP-43-HA experiment. **b**, UMAP depicting cells from the middle stage iNets of the timeline experiment (two replicates, see Supplementary Fig. 2a) analyzed jointly with TDP-43-HA OFF cells from the TDP-43-HA experiment (see Extended Data Fig. 6a). **c**, Cell type replicability analysis depicting pairwise AUROC values, a measure of similarity, between the annotated single-cell clusters as reported for the timeline experiment (Middle stage iNets, see Fig. 2a) and for the TDP-43-HA experiment (TDP-43-HA OFF, see Fig. 3d); clusters with more than 10 cells only. **d**, Number of iCLIP crosslinks[13] in FTLD patient and control brain samples in cluster 12 up- and -downregulated genes. **e**, Gene expression (CPM, counts per million) of top 10 cluster 12-downregulated markers in matched TDP-43-negative and TDP-43-positive neuronal nuclei from FTLD-ALS patients[38] (subject numbers match[38]).

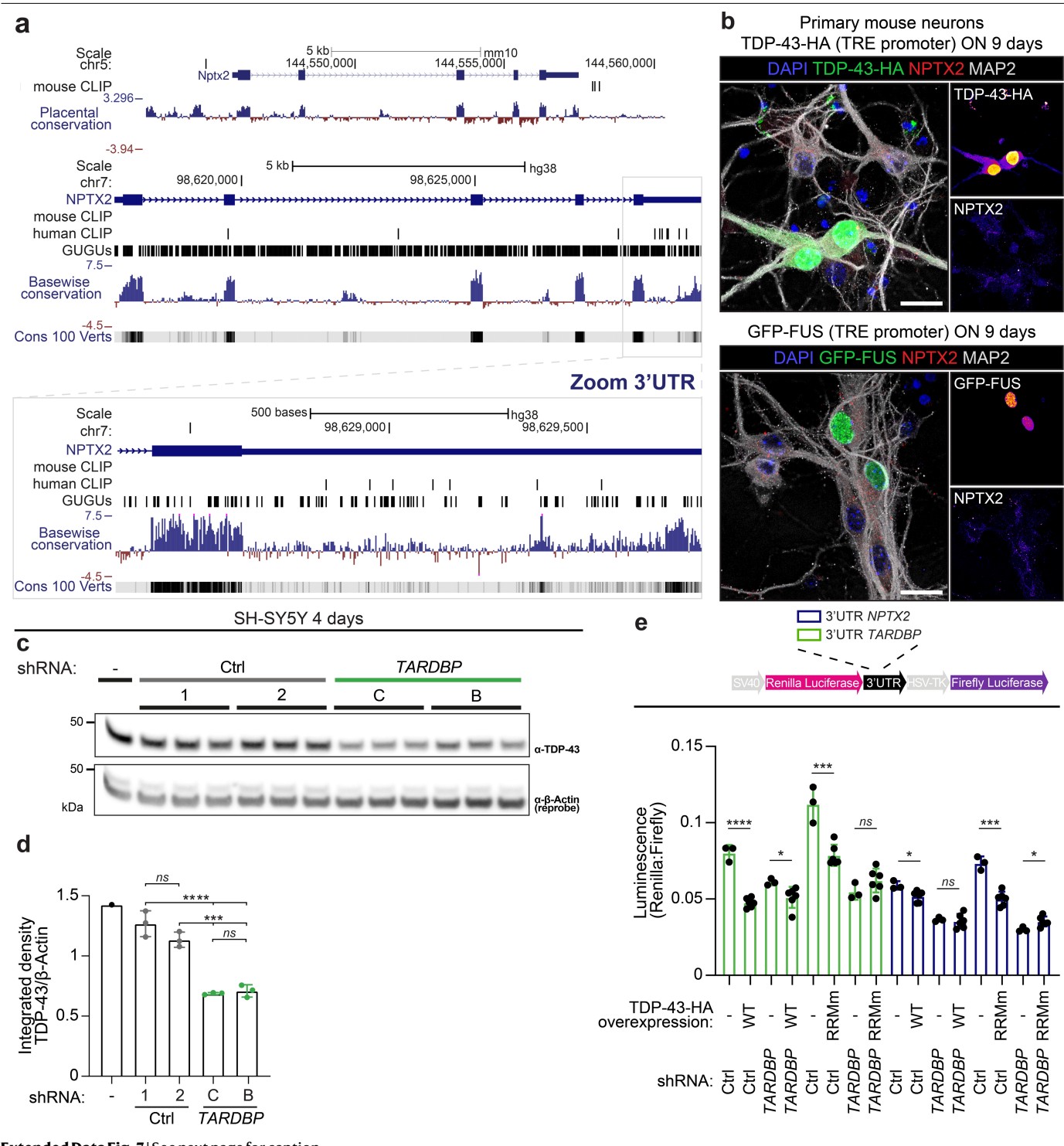

**Extended Data Fig. 7 |** See next page for caption.

**Extended Data Fig. 7 | Human-specific regulation of NPTX2 via TDP-43 binding on its 3'UTR. a**, Genomic track depicting the *Nptx2* gene structure in mouse, the lack of TDP-43 binding in adult mouse brain (iCLiP crosslinks[11]), and the sequence conservation across placental mammals[95] (top). Genomic track depicting the *NPTX2* (human) gene structure, TDP-43 iCLiP crosslinks in mouse brain (after liftover)[11], TDP-43 iCLiP crosslinks in human brain[13], 'GU' repeats, sequence conservation in placental mammals[95], and sequence conservation in vertebrates (middle). Zoom-in on 3'UTR (bottom). **b**, Compared to non-transduced neurons, there is no increase in NPTX2 immunofluorescence labeling in primary neurons overexpressing TDP-43-HA or GFP-FUS (note that HA-FUS also does not induce NPTX2 upregulation in iNets, see Extended Data Fig. 11). Primary neuron experiment was performed once. Scale bars, 20 µm. **c**, SH-SY5Y were transduced with lentivirus coding for *TARDBP* shRNAs or Ctrl shRNAs and the effect on protein levels was checked 4 days later by Western blot. **d**, Quantification of (**c**) showing significant TDP-43 protein lowering. One-way ANOVA followed by Tukey's multiple comparisons test (mean of each dataset compared with the mean of every other dataset, statistics not performed for n < 3). n = 3 individually transduced wells (1 for not treated).

One experiment. Data shown are mean with SD. Ctrl1 shRNA vs *TARDBP* shRNAb *p* < 0.0001; Ctrl1 shRNA vs *TARDBP* shRNAc *p* < 0.0001; Ctrl2 shRNA vs *TARDBP* shRNAb *p* = 0.0003; Ctrl2 shRNA vs *TARDBP* shRNAc *p* = 0.0002. **e**, Dual-Glo Luciferase Assay demonstrating the rescue of *TARDBP* and *NPTX2* 3'UTR reporter signal induced by loss of TDP-43 in KD conditions via temporary overexpression of TDP-43-HA and the opposite effect of RNA Recognition Motif mutant (RRMm) TDP-43-HA. Pairs analyzed by unpaired *t* test. Data shown are mean with SD and *p* values are two-tailed. Ctrl shRNA + TDP-43-HA WT OFF vs Ctrl shRNA + TDP-43-HA WT ON *TARDBP* 3'UTR *p* < 0.0001; *TARDBP* shRNA + TDP-43-HA WT OFF vs *TARDBP* shRNA + TDP-43-HA WT ON *TARDBP* 3'UTR p = 0.0467; Ctrl shRNA + TDP-43-HA RRMm OFF vs Ctrl shRNA + TDP-43-HA RRMm ON *TARDBP* 3'UTR p = 0.0009; Ctrl shRNA + TDP-43-HA OFF vs Ctrl shRNA + TDP-43-HA WT ON *NPTX2* 3'UTR p = 0.0203; Ctrl shRNA + TDP-43-HA RRMm OFF vs Ctrl shRNA + TDP-43-HA RRMm ON *NPTX2* 3'UTR p = 0.0002; *TARDBP* shRNA + TDP-43-HA RRMm OFF vs *TARDBP* shRNA + TDP-43-HA RRMm ON *NPTX2* 3'UTR p = 0.0325 n = 6 independently treated wells (3 for OFF conditions).

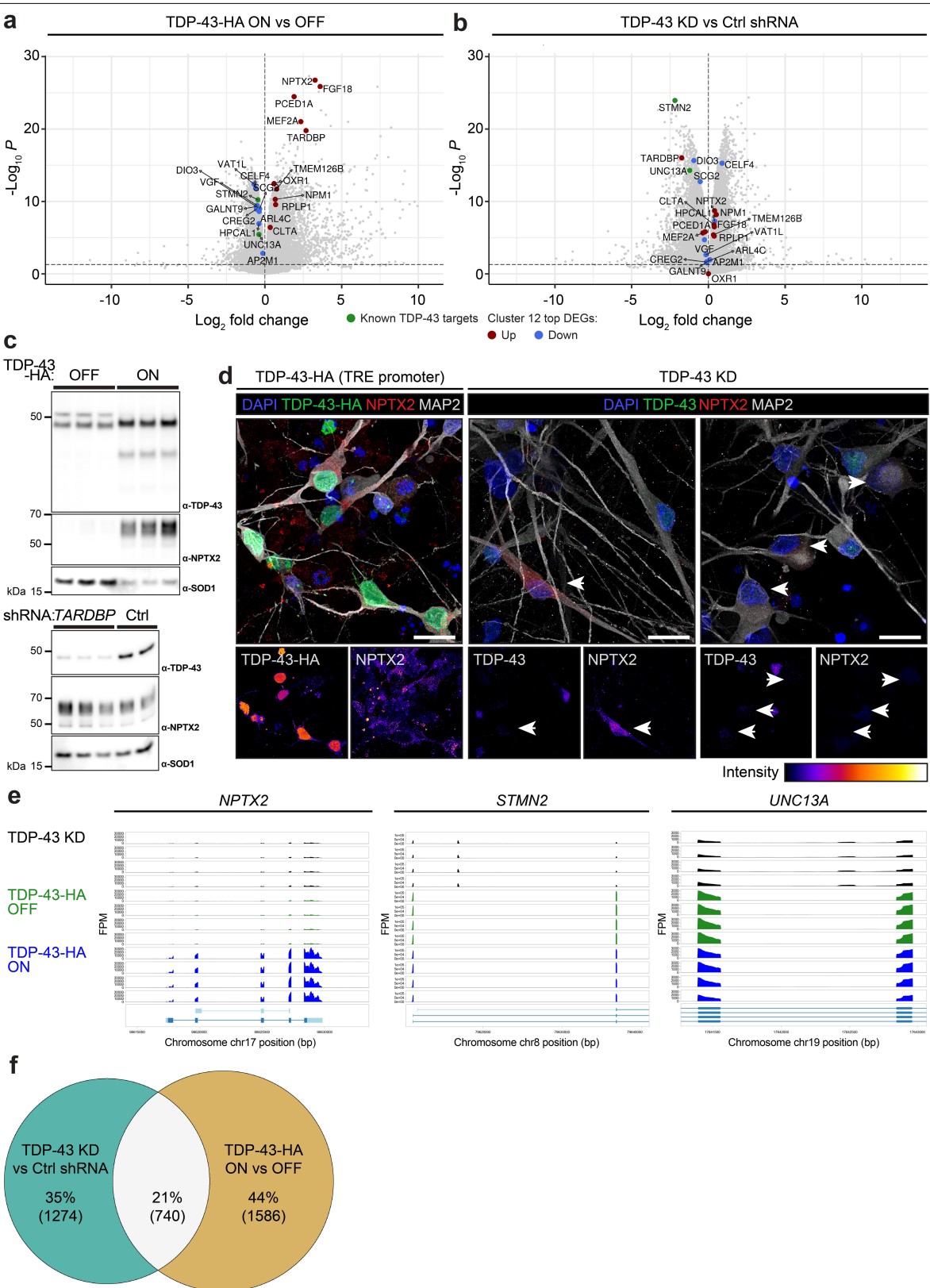

**Extended Data Fig. 8 |** See next page for caption.

**Extended Data Fig. 8 | Mechanism of NPTX2 upregulation in iNets with TDP-43-HA-driven proteinopathy. a**, Volcano plot depicting the differential expression status of the top cluster 12 up- and down-regulated genes (as defined with scRNA-seq data) and the known TDP-43 targets STMN2 and UNC13A in our bulk TDP-43-HA overexpression (OE) RNA-seq. If congruent, down-regulated genes in cluster 12 (blue) should be down-regulated in the bulk TDP-43-HA overexpression (logFC <0), and up-regulated genes in cluster 12 (red) should be up-regulated in the bulk TDP-43-HA overexpression (logFC >= 0). **b**, Volcano plot depicting the top cluster 12 up- and down-regulated genes (as defined with scRNA-seq data) in our bulk TDP-43 KD RNA-seq. **c**, Western blot showing NPTX2 upregulation in iNets with 2 weeks of TDP-43-HA OE (vs OFF, upper panel) or, in lower extent, 2 weeks of TDP-43 KD (vs Ctrl shRNA; lower panel). Each lane represents an individually transduced well. Representative blots from 2 independent experiments. **d**, Immunofluorescence of the same conditions as in (**c**) demonstrating high NPTX2 accumulation in TDP-43-HA overexpressing neurons (left) or moderate (middle) to low (right) in neurons with TDP-43 KD (arrows pointing at low nuclear TDP-43 neurons with elevated NPTX2). Representative images from 2 independent experiments. Scale bars, 20 μm. **e**, Gene expression levels (FPM, fragments per million mapped fragments) along the *NPTX2*, *STMN2* and *UNC13A* gene in iNets with TDP-43 KD, TDP-43-HA ON and OFF; the RefSeq gene model highlighting the canonical exons and introns is shown below. **f**, Venn diagram demonstrating overlap between genes with significant splicing events in TDP-43-HA overexpression (ON vs OFF) and TDP-43 KD (*TARDBP* shRNA vs Ctrl shRNA) in our respective RNA-seq datasets.

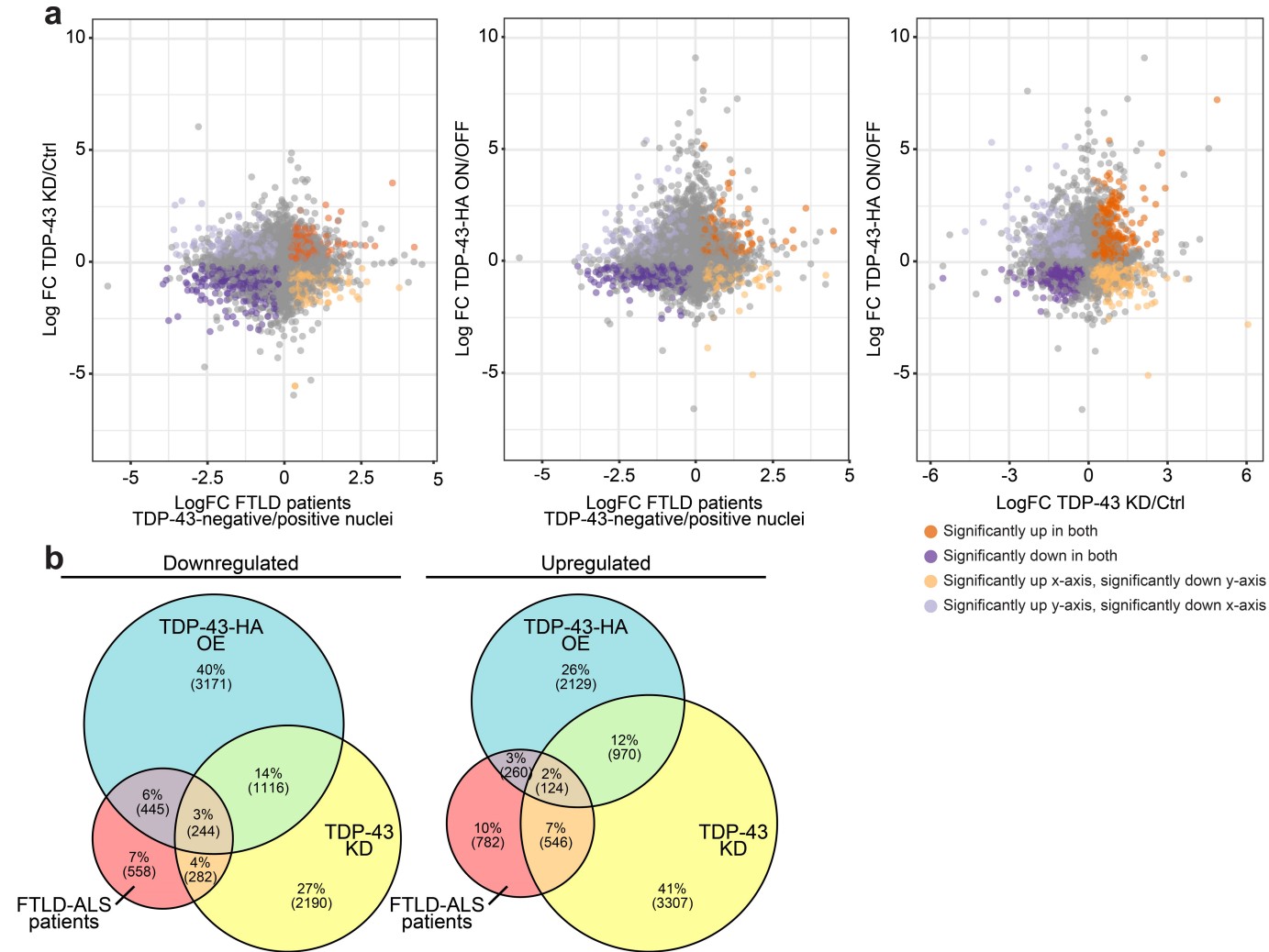

**Extended Data Fig. 9 | Overexpression and KD of TDP-43 partly recapitulate transcriptomic alterations observed in affected neurons in FTLD-ALS patients. a**, 4-way comparisons demonstrating the overlap of genes that are significantly altered in both TDP-43 KD (vs Ctrl shRNA) iNet neurons and in TDP-43-negative FTLD-ALS patient neurons (vs TDP-43-positive)[38] (left),

TDP-43-HA ON (vs OFF) iNet neurons and in TDP-43-negative FTLD-ALS patient neurons (vs TDP-43-positive) (middle), TDP-43-HA ON (vs OFF) and TDP-43 KD (vs Ctrl shRNA) iNet neurons (right). **b**, Venn diagrams visualizing the overlap of significantly downregulated (left) or upregulated (right) genes from all 3 conditions in (**a**). OE = overexpression, KD = knockdown.

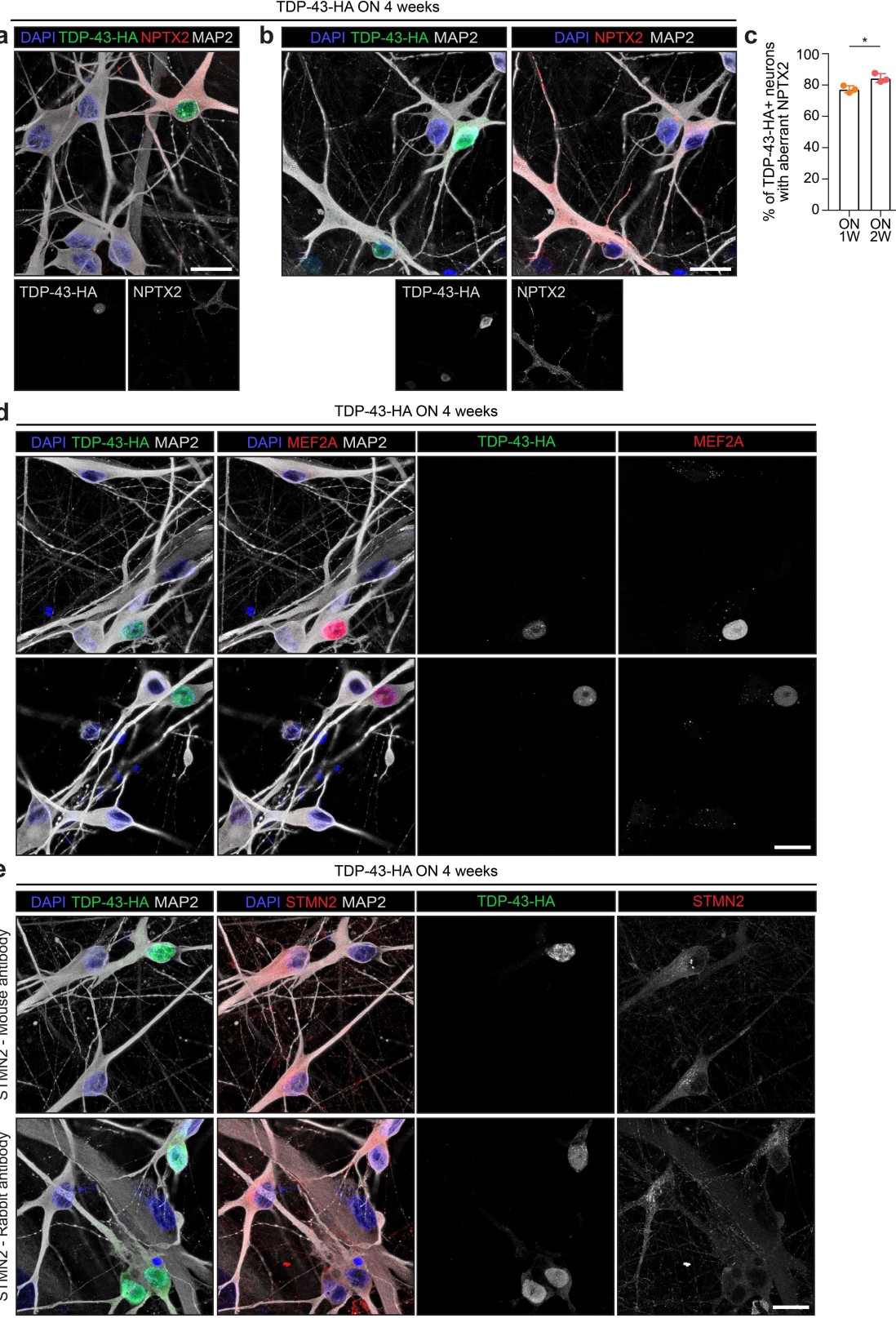

**Extended Data Fig. 10 | Immunofluorescence of cluster 12 misregulated genes in iNets overexpressing TDP-43-HA. a,b** Immunofluorescence of iNets transduced to overexpress TDP-43-HA for 4 weeks demonstrating that only the TDP-43-HA-positive neurons accumulated NPTX2 in the soma and neurites **c**, Quantification of the percentage of TDP-43-HA-expressing neurons with NPTX2 accumulation at 1 and 2 weeks of transgene expression. Unpaired *t* test. Data shown from a representative experiment (out of *n* = 2). Shown are mean with SD and *p* values are two-tailed. 1 W vs 2 W *p* = 0.0271. Each data point represents a sum of all MAP2+ neurons expressing TDP-43-HA that showed NPTX2 upregulation; expressed as a percentage out of total neuron count per well (182 fields of view). Between 855 - 2722 cells were analyzed per data point. Immunofluorescence of iNets with TDP-43-HA expression for 4 weeks showing increased levels of MEF2A in the nucleus (**d**) and lowered levels of STMN2 (demonstrated with 2 different antibodies) in the soma and processes (**e**). MEF2A and STMN2 IF were repeated twice. Scale bars, 20 μm.

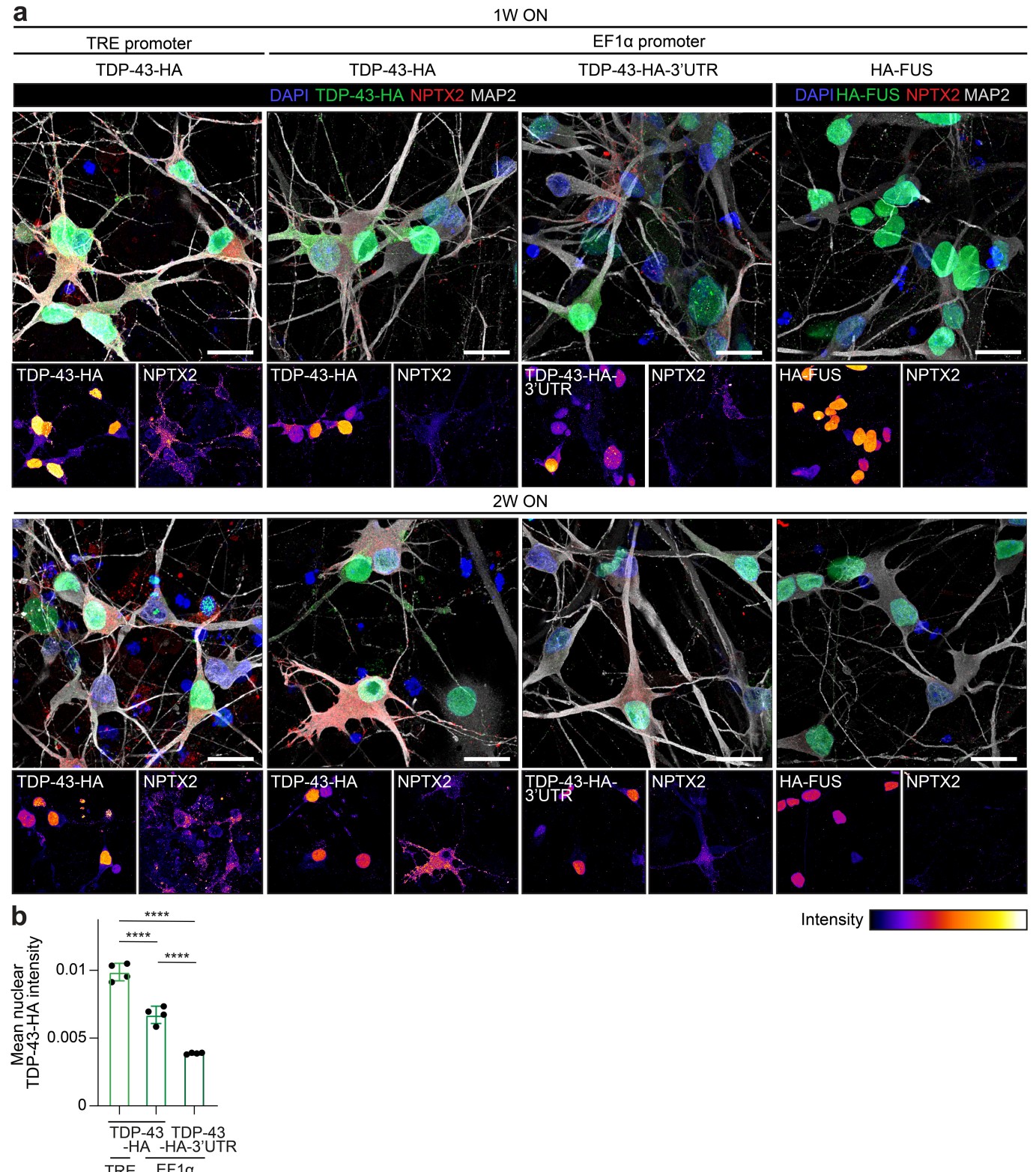

**Extended Data Fig. 11 | TDP-43 dose-dependent toxicity and upregulation of NPTX2. a**, Immunofluorescence of iNets expressing TDP-43-HA without or with its 3'UTR or HA-FUS driven by the inducible TRE or constitutive EF1α promoter for 1 week or 2 weeks revealing specificity of NPTX2 accumulation to TDP-43 dysregulation. Scale bars, 20 μm. **b**, Quantification of averaged mean intensities of nuclear TDP-43-HA signal of TDP-43-HA+ cells in iNets from the 1 week condition in **(a)**. One-way ANOVA followed by Tukey's multiple comparisons test (mean of each dataset compared with the mean of every other dataset). Data shown are mean with SD. TRE-TDP-43-HA vs EF1α-TDP-43-HA; and vs EF1α-TDP-43-HA-3'UTR $p < 0.0001$; EF1α-TDP-43-HA vs EF1α-TDP-43-HA-3'UTR $p < 0.0001$. Each data point represents the average intensity value measured from all TDP-43-HA+ nuclei detected in a well (36 fields of view; 2852-3749 cells measured per condition).

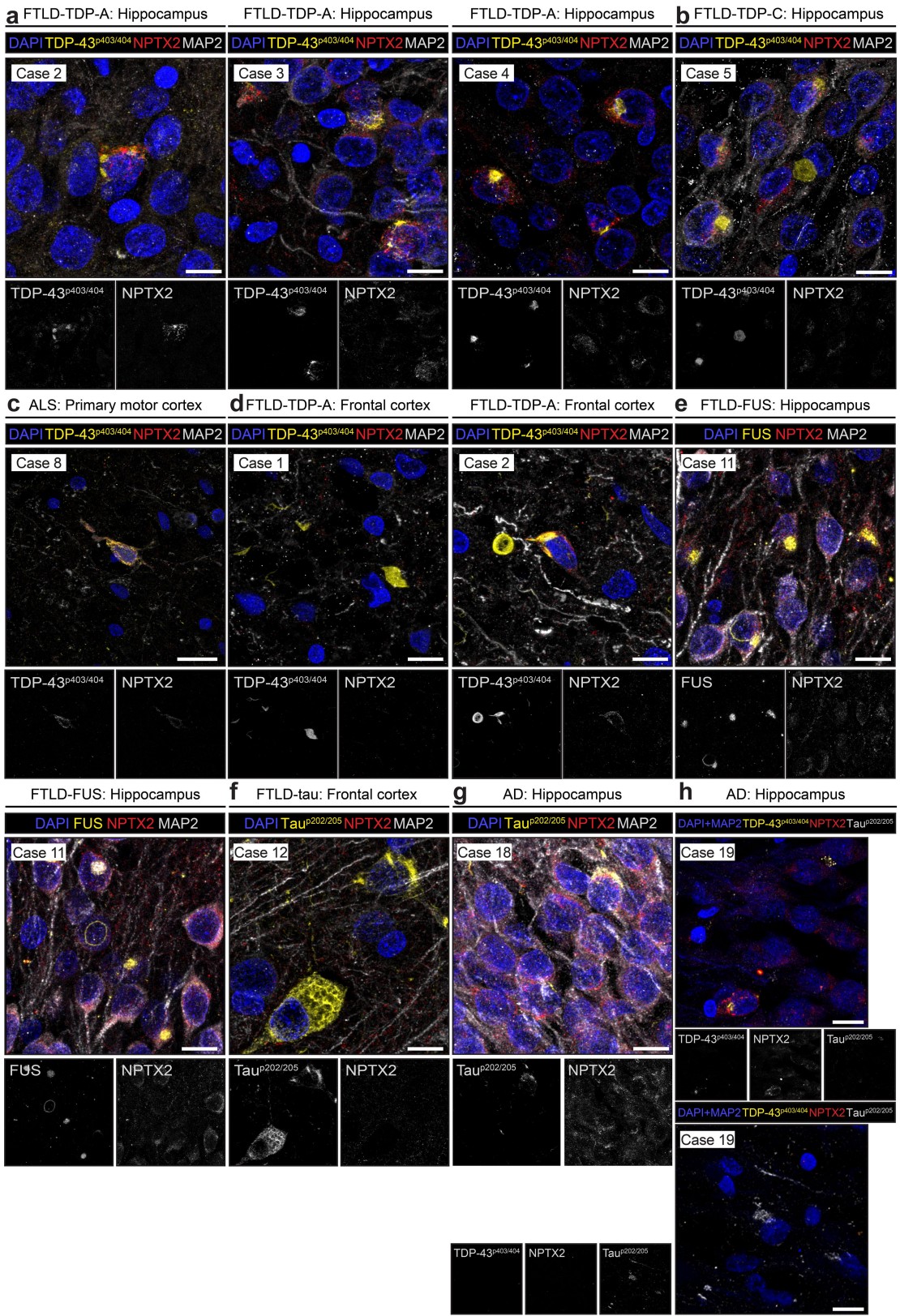

**Extended Data Fig. 12** | See next page for caption.

**Extended Data Fig. 12 | NPTX2 immunofluorescence in brain sections of patients with neurodegenerative proteinopathies.** Additional immunofluorescence gallery of multiple FTLD, ALS and AD cases demonstrating NPTX2 accumulation and inclusions in neurons with TDP-43$^{p403/404}$-positive aggregates in the hippocampus of (**a**) FTLD-TDP-A cases, (**b**) FTLD-TDP-C case and in the primary motor cortex of an ALS case (**c**). **d**, Note that while the NPTX2 pathology could be observed in cortex of both FTLD-TDP and ALS patients, it is only present when the TDP-43$^{p403/404}$-positive aggregate-containing neurons could still be identified by the MAP2 staining (right) as opposed to the disintegrated, "ghost neurons" that left behind only the TDP-43$^{p403/404}$-positive aggregates (left). Absence of NPTX2 accumulation in neurons with FUS or tau$^{p202/205}$ inclusions in FTLD-FUS hippocampus (**e**) or FTLD-tau frontal cortex (**f**), respectively. NPTX2 aberrancy is absent from neurons with tau$^{p202/205}$-positive (**g, h** lower panels) but present in neurons with TDP-43$^{p403/404}$-positive (**h** upper panels) inclusions in the hippocampus of AD cases. Patient IF was repeated 4 (FTLD-TDP-A and -C cases) or 2 times (FTLD-FUS, FTLD-tau, AD, ALS cases). Scale bars, (**a**,**b**,**d-h**) 10 μm, (**c**) 20 μm.

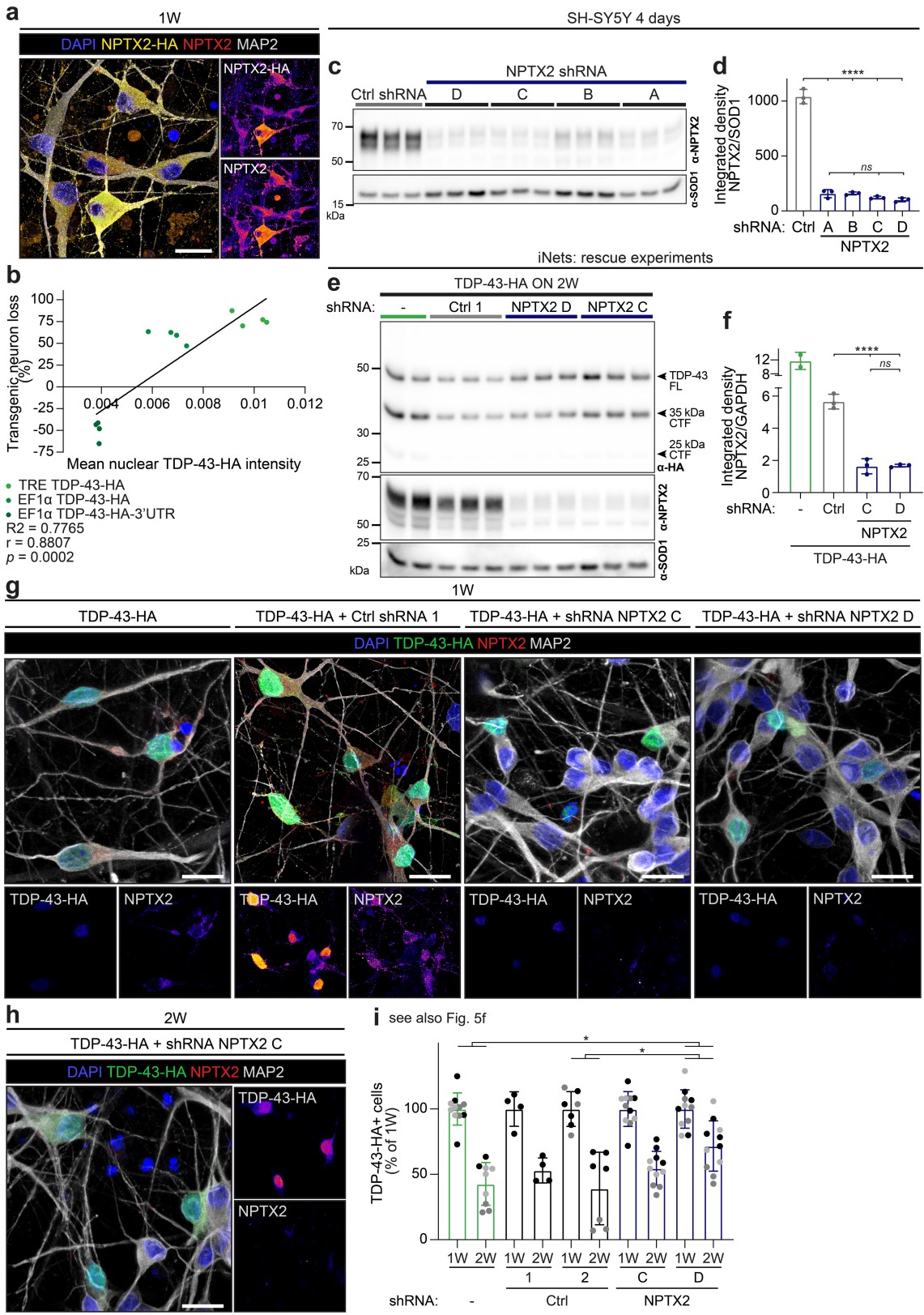

**Extended Data Fig. 13** | See next page for caption.

**Extended Data Fig. 13 | Silencing NPTX2 after onset of TDP-43 pathology alleviates neurotoxicity in iNets. a**, HA and NPTX2 immunofluorescence in EF1α-driven NPTX2-HA overexpressing iNet neurons at 1 week results in an accumulation throughout the soma and processes of transgenic iNet neurons, similar to NPTX2 accumulation induced by TDP-43-HA overexpression. One experiment. **b**, Correlation plot of averaged (per well) mean nuclear TDP-43-HA intensities (1 week) and the % loss of transgenic TDP-43-HA iNet neurons at 2 weeks. TDP-43-HA nuclear intensity was measured in 2852-3749 cells per condition and between 281-2265 cells were analyzed per data point. **c**, Western blot demonstrating the effect of expression of *NPTX2* shRNAs (vs Ctrl shRNA) in transduced SH-SY5Y cells after 4 days. **d**, Signal quantification of (**c**). One-way ANOVA followed by Tukey's multiple comparison test (mean of each dataset compared with the mean of every other dataset). n = 3 individually transduced well. Data shown as mean with SD. Ctrl shRNA vs *NPTX2* shRNAa,b,c and d $p < 0.0001$. **e**, Western blot showing NPTX2 upregulation in iNets upon TDP-43-HA overexpression which is corrected by co-expression of *NPTX2* shRNAs. **f**, Signal quantification of (**e**). One-way ANOVA followed by Tukey's multiple comparisons test (mean of each dataset compared with the mean of every other dataset, statistics not performed for n < 3). n = 3 individually

transduced wells (2 for TDP-43 OE only). Data are shown as mean with SD. TDP-43-HA OE + Ctrl shRNA vs TDP-43-HA OE + NPTX2 shRNAc $p < 0.0001$ or shRNAd $p < 0.0001$. **g**, Representative immunofluorescence of TRE-driven TDP-43-HA-transduced iNets at 1 week, showing clear NPTX2 upregulation and accumulation in both TDP-43-HA only and Ctrl shRNA conditions, which was corrected by *NPTX2* KD. Note that the panel of TDP-43-HA + Ctrl shRNA is derived from a separate experiment. **h**, Representative immunofluorescence (out of 3 experiments) of TRE-driven TDP-43-HA + shRNA *NPTX2* C expressing neurons at 2 weeks. **i**, Quantification of TDP-43-HA-transgenic cell counts in the conditions shown in (**g**,**h**). Each data point represents a sum of all HA+ cells counted from 182 fields of view of an independent well and normalized with DAPI, and shown as % of 1 W week condition. Between 189 - 16453 cells were analyzed per data point. Only significant differences are shown. Two-way ANOVA (mixed model) followed by Tukey's multiple comparison test (compare each dataset mean with every other dataset mean). Differently gray scaled data points are independent experiments. Data shown are mean with SD. TDP-43-HA only vs TDP-43-HA + *NPTX2* shRNAd $p = 0.0356$; TDP-43-HA + Ctrl shRNA2 vs TDP-43-HA + *NPTX2* shRNAd $p = 0.0316$. Scale bars, (**a**, **g**, **h**) 20 μm.

# Reporting Summary

## Statistics

For all statistical analyses, confirm that the following items are present in the figure legend, table legend, main text, or Methods section.

| n/a | Confirmed | |
|---|---|---|
| ☐ | ☒ | The exact sample size (*n*) for each experimental group/condition, given as a discrete number and unit of measurement |
| ☐ | ☒ | A statement on whether measurements were taken from distinct samples or whether the same sample was measured repeatedly |
| ☐ | ☒ | The statistical test(s) used AND whether they are one- or two-sided<br>*Only common tests should be described solely by name; describe more complex techniques in the Methods section.* |
| ☐ | ☒ | A description of all covariates tested |
| ☐ | ☒ | A description of any assumptions or corrections, such as tests of normality and adjustment for multiple comparisons |
| ☐ | ☒ | A full description of the statistical parameters including central tendency (e.g. means) or other basic estimates (e.g. regression coefficient) AND variation (e.g. standard deviation) or associated estimates of uncertainty (e.g. confidence intervals) |
| ☐ | ☒ | For null hypothesis testing, the test statistic (e.g. *F*, *t*, *r*) with confidence intervals, effect sizes, degrees of freedom and *P* value noted<br>*Give P values as exact values whenever suitable.* |
| ☒ | ☐ | For Bayesian analysis, information on the choice of priors and Markov chain Monte Carlo settings |
| ☒ | ☐ | For hierarchical and complex designs, identification of the appropriate level for tests and full reporting of outcomes |
| ☒ | ☐ | Estimates of effect sizes (e.g. Cohen's *d*, Pearson's *r*), indicating how they were calculated |

*Our web collection on statistics for biologists contains articles on many of the points above.*

## Software and code

Policy information about availability of computer code

| Data collection | Leica Application Suite X for microscopy. EVOS M7000 Imaging System Software for microscopy. MetaXpress (version 6.7.1) for microscopy. InCell Analyzer 2500 (version 7.1).  ScanImage for microscopy. Amersham Imager 600RGB integrated software and Fusion FX6 EDGE imager integrated software for western blot imaging. HD-MEA data were recorded using MaxLab Live. |
|---|---|
| Data analysis | Images acquired on confocal microscopes were deconvoluted using SVI Huygens (version 22100p3) and processed for visualization in Fiji (20230118). Widefield and MetaXpress images were pixel segmented in Ilastik (version 1.3.3post3) and quantified in Fiji. Nuclei signal intensity data was analyzed in CellProfiler (version 4.2.5). Western blots were processed and analyzed in Fiji. Lentiviral vector titers were checked using Takara GoStix Plus app (version 3.0.2). ImageJ (release  1.54g) and Matlab (version 2016a) were used to analyze 2-photon calcium imaging data. Patch clamp data were analyzed in Igor Pro. Data analysis of HD-MEA data was performed using custom-written codes in MATLAB (R2021a) and (Python 3.6.10) and Kilosort2 software within the SpikeInterface framework for spike sorting. Statistical analysis was performed in Prism (version 9.5.0) unless otherwise mentioned.<br><br>Data was processed with CellRanger for demultiplexing, read alignment to the human reference genome (GRCh38) and filtering to generate a feature-barcode matrix per sample. Cell doublets were removed with scDblFinder64 and outlier cells were detected and filtered  with the "scater" R package. Seurat v3 was used for log-normalization and to identify the top 2000 highly variable genes per sample. Marker genes that are upregulated in one cluster compared to any other cluster were identified with the findMarkers function from the scran R package. scRNAseq samples were integrated in Seurat v3. Batch effects were assessed with the smoothed `cms` mixing metric from CellMixS. CellRanger was used for quantification of the expression of the TDP-43-HA construct components. The total TDP-43 log2FC between cluster 12 and all other neuronal clusters was computed using the `summary.logFC` metric from scran's findMarkers. Pseudotime analysis was carried out with monocle3 v1.0.0, commit 004c096, rooting the trajectories in our annotated NSC cluster. MetaNeighbor v1.8.0 to measure cell-type replication across experiments using the annotated clusters. Raw reads from bulk RNA seq were processed with ARMOR76. Briefly, reads were |

aligned and counted to the human genome (GRCh38 assembly and Gencode release 43) with salmon v1.4.0 and with STAR 2.7.7a. We modeled the salmon-generated count data with quasi-likelihood (QL) negative binomial (NB) generalized log-linear models and ran differential expression analysis with edgeR v3.36.0. We reanalyzed CLiP data from GSE27201 Polymenidou 2011 (mouse) and Tollervey 2011 (human) with the nf-core/clipseq workflow v1.0.077 with default parameters and against the GRCh38 or GRCm38 reference genomes, as appropriate. Polymenidou 2011 CLiP binding sites were transformed from GRCm38 to GRCh38 coordinates using liftOver from Kent utils v39078. CLiP and iCLiP data was re-analysed using the GRCh38 reference genome with the nf-core/clipseq77 for preprocessing, mapping and crosslink site identification. Re-analysis of RNA-seq data from sorted nuclei was performed via differential gene expression analysis between TDP-43 negative and positive nuclei was performed in R version 4.0.5 with edgeR taking into account the paired nature of the data using the quasi-likelihood framework.

Splicing analysis to identify skipped exons was performed using the human reference genome (GRCh38) and annotation (version 108) were obtained from ENSEMBL91. Illumina Trueseq adaptors were removed from RNA-seq reads using cutadapt (version 4.1) with the parameters -q 25 -m 25 . The processed reads were mapped to the reference genome using STAR aligner (version 2.7.10b) with default parameters. Subsequently, differential splicing analysis at the event level was performed using rMATS (version 4.1.2) with the parameters -t paired --readLength 100 --variable-read-length. Significant skipped exons events were defined by FDR < 0.05 and 15 % change in absolute value of IncLevelDifference and intersected with a human skipped exon dataset43. The results were processed using tidyverse package (v2.0.0) in R. Upset plot was prepared using Complexheatmap package (v2.17.0) in R.

Code is deposited on Zenodo with doi https://doi.org/10.5281/zenodo.8142336.

For manuscripts utilizing custom algorithms or software that are central to the research but not yet described in published literature, software must be made available to editors and reviewers. We strongly encourage code deposition in a community repository (e.g. GitHub). See the Nature Portfolio guidelines for submitting code & software for further information.

# Data

Policy information about availability of data

All manuscripts must include a data availability statement. This statement should provide the following information, where applicable:
- Accession codes, unique identifiers, or web links for publicly available datasets
- A description of any restrictions on data availability
- For clinical datasets or third party data, please ensure that the statement adheres to our policy

Data availability
scRNA-seq and bulk RNA-seq data are available via GEO accession number GSE230647.

Other databases used in this study:
- The scRNA-seq data from Lam 2019 were downloaded from ArrayExpress (accession number E-MTAB-8379).
- Reanalyzed CLiP data from GSE27201 Polymenidou 2011 (mouse) including accessions SRR107031 and SRR107032; from E-MTAB-530 Tollervey 2011 (human) accessions ERR039847, ERR039848, ERR039843, ERR039842, ERR039844, ERR039845, ERR039846.
- FASTQ files of TDP-43 human brain iCLIP13 (three controls and three FTLD patients) were downloaded from ArrayExpress (E-MTAB-530).
- FASTQ files of TDP-43 CLiP in mouse were retrieved from GSE27201.
- We downloaded the RNA-seq gene count table as a supplementary data file (GSE126542_NeuronalNuclei_RNAseq_counts.txt.gz) from GEO accession GSE126542.

Code availability
Code is deposited on Zenodo with doi https://doi.org/10.5281/zenodo.8142336.

# Research involving human participants, their data, or biological material

Policy information about studies with human participants or human data. See also policy information about sex, gender (identity/presentation), and sexual orientation and race, ethnicity and racism.

| Reporting on sex and gender | Used sex to identify patient sex. |
|---|---|
| Reporting on race, ethnicity, or other socially relevant groupings | We do not report on race, ethnicity, or other socially relevant groups. |
| Population characteristics | Basic demographic information was provided with the human brain tissue, which was selected based on the underlying pathological diagnosis and the known mutations that were determined in the clinical setting. Other demographic data included age at onset, disease duration, age at death and sex. No other information was provided or was relevant for the present study. |
| Recruitment | The participants were recruited through the brain donation programme at Queen Square Brain Bank, therefore there is no self-selection biases. Cases identified for this project were selected based on their underlying pathological diagnosis. |
| Ethics oversight | Formalin-fixed, paraffin-embedded hippocampal, frontal or primary motor cortex patient (ALS, FTLD-TDP Type A, FTLD-TDP Type C, FTLD-FUS, FTLD-Tau, AD) sections (see Supplementary Table 9) were used. All FTLD and AD tissue samples were donated to Queen Square Brain Bank for Neurological Disorders at UCL Queen Square Institute of Neurology with full, informed consent and the material was supplied with an approved material transfer agreement (EXT MTA 07-2017). Anonymized autopsy ALS sample was collected by the Institute of Neuropathology at UZH. According to Swiss law, anonymized autopsy tissues do not fall within the scope of the Human Research Act and may be used in research. |

# Field-specific reporting

Please select the one below that is the best fit for your research. If you are not sure, read the appropriate sections before making your selection.

☒ Life sciences          ☐ Behavioural & social sciences          ☐ Ecological, evolutionary & environmental sciences

For a reference copy of the document with all sections, see nature.com/documents/nr-reporting-summary-flat.pdf

# Life sciences study design

All studies must disclose on these points even when the disclosure is negative.

| | |
|---|---|
| Sample size | For experiments with iNets, minimum of 3 individually treated wells (i.e. individually transduced) of imaging multi well plates (or a larger vessels for bulk RNA seq or biochemistry) were used per condition per experiment as this was previously determined to be sufficient sample size (Emmenegger et al. 2021 EMBO Molecular Medicine).<br><br>Luminescence experiments with HEK293T cells were also performed with at least 3 replicates (i.e. individually transfected and transduced wells). No size estimation was performed as the sufficient sample size was determined by trial experiments.<br><br>Minimum of 3 wells of 6 well plates per LV (shRNA) were used for SH-SY5Y LV transduction to determine shRNA efficacies. No size estimation was performed as the sufficient sample size was determined by previous experiments independent from this paper (Avar et al. 2022 The EMBO Journal).<br><br>4 wells of an ibidi chamber slide per LV were used in the primary mouse neuron experiment. No size estimation was performed as the sufficient sample size was determined by previous experiments independent from this paper .<br><br>Formalin-fixed, paraffin-embedded hippocampal, frontal or primary motor cortex patient (ALS, FTLD-TDP Type A, FTLD-TDP Type C, FTLD-FUS, FTLD-Tau, AD) sections (see Supplementary Table 9) were used based on availability, and at least 2 slides per patient per staining were used. |
| Data exclusions | No data were excluded. |
| Replication | All repeated experiments were successful (i.e. significant or same trend; did not fail; did not result in an opposite effect) and this is mentioned in the text or visualized in the figure legend. |
| Randomization | For all experiments with iNets and HEK293T cells, wells for different conditions were randomly selected (but kept within the rows or columns). SH-SY5Y cells for shRNA testing were randomly transduced in 6 well plates (but 2 lentiviral vectors were used per plate). Patient brain cases identified for this project were selected based on their underlying pathological diagnosis and availability. |
| Blinding | All iNets imaging quantification experiments were imaged with high-content microscopes and data were analyzed in one batch, i.e. the information about the sample type was only revealed and used when plotting the results of quantification and therefore blinding was not relevant.<br>For all other experiments, blinding was not possible due to nature of the experiments. |

# Reporting for specific materials, systems and methods

We require information from authors about some types of materials, experimental systems and methods used in many studies. Here, indicate whether each material, system or method listed is relevant to your study. If you are not sure if a list item applies to your research, read the appropriate section before selecting a response.

### Materials & experimental systems

| n/a | Involved in the study |
|---|---|
| ☐ | ☒ Antibodies |
| ☐ | ☒ Eukaryotic cell lines |
| ☒ | ☐ Palaeontology and archaeology |
| ☐ | ☒ Animals and other organisms |
| ☒ | ☐ Clinical data |
| ☒ | ☐ Dual use research of concern |
| ☒ | ☐ Plants |

### Methods

| n/a | Involved in the study |
|---|---|
| ☒ | ☐ ChIP-seq |
| ☒ | ☐ Flow cytometry |
| ☒ | ☐ MRI-based neuroimaging |

## Antibodies

| | |
|---|---|
| Antibodies used | Precise information about antibodies can be found in the Supplementary Table 10 |

Primary antibodies:
AQP4, Rb, Novus Biologicals #NBP1-87679
DCX, Gt, Santa Cruz Biotechnology #sc-8066 LOT E0115, clone C-18
FUS, Ms, ProteinTech #60160-1-Ig, LOT 10003284, clone 3A10B5
GFAP, Gt, Abcam #ab53554
HA, Rb, Cell Signaling Technology #3724, LOT 11, clone C29F4
HA, Ms, Biolegend #901516, LOT B324182, clone 16B12
HA, Ms, ThermoFisher #26183, LOT YB362804, clone 2-2.2.14
KI67, Rb, Abcam #ab16667, clone SP6
MAP2, Ms, Sigma #M1406, clone AP-20
MAP2, Ch, Abcam #ab5392, LOTs GR3426750-3, GR3450786-1
MEF2A, Rb, Santa Cruz Biotechnology #sc-17785, LOT D29121, clone B-4
NEFM, Ms, Thermo Scientific #13-0700, LOT RL246769, clone RMO-270
Nestin, Ch, Online antibodies #ABIN187958
NEUN, Ch, Millipore #ABN91, LOT 2695293
NPTX2, Rb, Proteintech #10889-1-AP, LOTs 00052993, 00053480
NUMA, Rb, Bethyl #A301-510A, LOT 1
PLZF, Rb, Santa Cruz Biotechnology #sc22839, LOT G1414, clone H300
PSD-95, Ms, Abcam #ab2723-100, LOT XE341719, clone 6G6-1C9
SNAP-25, Ms, #SMI81, clone SMI-81
SOD1, Rb, Enzo #ADI-SOD-100, LOTs 09082051, 03122119
SOX2, Gt, Santa Cruz Biotechnology #sc17320, LOT E0715, clone Y-17
STMN2, Ms, Proteintech #67204-1-Ig, LOT 10011368, clone 1F6C4
STMN2, Rb, Proteintech #10586-1-AP, LOT 00058414
SYP, Rb, Santa Cruz Biotechnology #sc-9116, clone H93
Tau p202/205, Ms, ThermoFisher #MN1020, LOT VL3113305, clone AT8
TDP-43p403/404, Custom made, Murinised human monoclonal
VIM, Ch, Millipore #AB5733, LOT 3822323
ZO1, Rb, Millipore #AB2272, LOT 2549491
β-ACTIN, Ms, Sigma #A5441, LOT 000126949, clone AC-15
TDP-43 FL, Rb, Proteintech #18280-1-AP, LOTs 00072915, 00025213
TDP-43 3H8, Ms, Novus #NBP1-92695, LOTs 022221, 082119, clone 3H8

Secondary antibodies:
Donkey anti-Ch 488, Jackson Immuno Research #JAC703-546-155, LOT 2420700
Donkey anti-Ch 568, Jackson Immuno Research #JAC703-586-155, LOT 141724
Donkey anti-Ch 647, Jackson Immuno Research #JAC703-606-155, LOT 157044
Donkey anti-Gt 488, ThermoFisher #A11055, LOT 1771339
Donkey anti-Gt 594, ThermoFisher #A11058, LOT 1445994
Donkey anti-Gt 647, ThermoFisher #A21447, LOT 1739289
Donkey anti-Ms 488, ThermoFisher #A21202, LOT 2428531
Goat anti-Ms 555 PLUS, ThermoFisher #A4287, LOT WH334377
Donkey anti-Ms 568, ThermoFisher #A10037, LOT 2300930
Donkey anti-Ms 647, ThermoFisher #A31571, LOT 2555690
Donkey anti-Rb 488, ThermoFisher #A21206, LOT 2289872
Donkey anti-Rb 488 PLUS, ThermoFisher #A32790, LOT VJ314112
Donkey anti-Rb 568, ThermoFisher #A10042, LOT 2540901
Donkey anti-Rb 647, ThermoFisher #A31573, LOT 2359136
Goat anti-Ch 647, ThermoFisher #A21449, LOT 1744743
Goat anti-Ch 647 PLUS, ThermoFisher #A32933, LOT XD344361
Goat anti-Ms-HRP, Jackson Immuno Research #115-035-146, LOT 165273
Goat anti-Rb-HRP, Jackson Immuno Research #115-035-144 LOT 163676

Validation

Precise information about antibodies can be found in the Supplementary Table 10

Primary antibodies:
AQP4, PMID:36803838, AB_11006038
DCX, PMID:26337870; PMID:2770949, AB_2088494
FUS, PMID:36171642, AB_10666169
GFAP, PMDI:36384142; PMID:35431801, AB_880202
HA, PMID:25380328, AB_1549585
HA, PMID:32515353, AB_2820200
HA, PMID:12119359, AB_10978021
KI67, PMID:24424056, AB_302459
MAP2, PMID:19058188, AB_477171
MAP2, PMID: 28521134; PMID: 28637841, AB_2138153
MEF2A, PMID: 29518074; PMID: 36263180, AB_627921
NEFM, PMID:28472866, AB_2532998
Nestin, PMID:32778115
NEUN, PMID:28555077, AB_11205760
NPTX2, PMID:29844474; PMID:34565284, AB_2153875
NUMA, AB_999641
PLZF, PMID:30759202, AB_2304760
PSD-95, PMID:30638745, AB_303248
SNAP-25, PMID:18041776, AB_2315336
SOD1, PMID:30044993, AB_10616253

SOX2, PMID:31883789, AB_2286684
STMN2, AB_2882497
STMN2, AB_2197283
SYP, PMID:29798891, AB_2199007
Tau p202/205, PMID:31325178, AB_223647
TDP-43p403/404, PMID:34806807
VIM, PMID:31461644, AB_11212377
ZO1, AB_10807434
β-ACTIN, PMID:31549732, AB_476744
TDP-43 FL, PMID:36303452, AB_2240312
TDP-43 3H8, PMID:37182104, AB_11005586

# Eukaryotic cell lines

Policy information about cell lines and Sex and Gender in Research

| Cell line source(s) | HEK293T (ATCC #CRL-3216) - collaborator Urs Greber (UZH). SH-SY5Y - Sigma (#94030304). Human early neonatal dermal fibroblasts - Gibco #C0045C (male). |
|---|---|
| Authentication | HEK293T cells used for luminescence assay and lentiviral production were provided by a collaborator (Urs Greber) and were authenticated by the vendor (ATCC #CRL-3216). SH-SY5Y were obtained directly from Sigma (#94030304) and thus authenticated by the vendor. iCoMoNSCs were generated in this manuscript from in-house generated iPSCs that were generated from human early neonatal dermal fibroblasts purchased directly from Invitrogen (Gibco #C0045C). Fibroblasts, iPSCs and iCoMoNSCs were checked for karyotype. |
| Mycoplasma contamination | All cycling cells used in the study were routinely checked for possible mycoplasma contamination using EZ PCR Mycoplasma Detection Kit (Sartorius #20-700-20). Young iNets (with some residual KI67+ cells) were sporadically checked, too. No mycoplasma contamination was detected. |
| Commonly misidentified lines (See ICLAC register) | Cell lines (SH-SY5Y Sigma #94030304 and HEK293T ATCC #CRL-3216) were checked against cross-contaminated or misidentified cell lines (https://iclac.org/databases/cross-contaminations/). No cross-contaminated or misidentified cell lines were identified. |

# Animals and other research organisms

Policy information about studies involving animals; ARRIVE guidelines recommended for reporting animal research, and Sex and Gender in Research

| Laboratory animals | Mus musculus C57BL/6 strain. Mouse embryos (E16/17) were used. |
|---|---|
| Wild animals | No wild animals were used in the study. |
| Reporting on sex | Primary neurons were derived from hippocampi of all fetuses available - i.e. the resulting culture was made of mixed sex. |
| Field-collected samples | No field-collected samples were used. |
| Ethics oversight | Primary neuronal cell cultures were prepared from mouse embryos (E16/17). Pregnant C57BL/6 females were delivered from Janvier Labs (France) at day 12 of pregnancy. All animal experimentation, including mouse housing and breeding was done in accordance with the Swiss Animal Welfare Law and in compliance with the regulations of the Cantonal Veterinary Office, Zurich (current approved license ZH169/2022, valid until 16.02.2026). |

Note that full information on the approval of the study protocol must also be provided in the manuscript.

# Plants

| Seed stocks | N/A |
|---|---|
| Novel plant genotypes | N/A |
| Authentication | N/A |

