## [Peer Review File · Nature]

Manuscript Title: iNets with TDP-43 pathology reveal toxic NPTX2 misregulation in ALS/FTLD

Reviewer Comments & Author Rebuttals

Reviewer Reports on the Initial Version:

Referees' comments:

Referee #1 (Remarks to the Author):

In this manuscript, Magda Polymenidou and colleagues present a new method to induce self-renewing, homogeneous neural stem cells (iCoMoNSCs) from iPSCs. The iCoMoNSCs contain fewer postmitotic cells compared to existing human iPSCs-derived neural stem cells (NSCs) such as fetal-derived neural stem cell line SAI2, iPSCs-derived NSC line AF22 and Ctrl-7. Also, iCoMoNSCs can be induced into a cortical organoid-like cell population by differentiation. The degree of maturation of the induced neurons was well assessed not only by gene expression analysis but also by electrophysiological assays such as patch-clamp recording and microelectrode arrays, indicating that it is possible to generate a sufficiently mature cell population without creating 3D organoids. Furthermore, by overexpressing the FTD/ALS protein TDP-43 using lentiviruses in this model, they confirmed a decrease in expression in genes such as STMN2 and UNC13A, whose dysregulation have been recently connected to TDP-43 pathology in FTD/ALS, and also discover an upregulation in NPTX2, which the authors propose could be developed into a novel therapeutic target.

There are two main parts to this manuscript: 1) the establishment of a novel protocol/cell line of NSCs from iPSCs, which permits long-term culture and the ability to monitor neuronal activity and neuronal maturation state and 2) the discovery of a novel pathological candidate associated with TDP-43 pathology in FTD and ALS. Both components are exciting and, in my opinion, of great interest (I cannot imagine a single ALS laboratory that will not want to immediately start using this model system for their studies). And it comes at an exciting time in the ALS field, with the discovery of TDP-43 targets, many of which are human-specific, demanding human model systems to study them. Especially ones that have robust synaptic physiology and neuronal activity.

I have several comments and suggestions for the authors to consider. Below I offer some comments on the stem cell experiments but since my expertise is in FTD/ALS mechanisms and TDP-43 I focus most of my suggestions on that aspect and trust the other referees to rigorously evaluate the model development parts.

1. It would be helpful if the authors could directly test the reproducibility and advantages of iCoMoNSCs over other systems. The authors carefully designed iCoMoNSCs and present compelling data using various cellular, molecular, and electrophysiological assays. The iCoMoNSCs system is of great interest because it has the potential to be induced into functionally mature neurons with more reproducibility and uniformity compared to some recently developed organoid model systems. However, it seems that the authors have only established a single cell line for their analyses. It would be of interest to consider generating multiple independent lines from existing iPSCs and/or patient-derived cells. The future utility of this model system depends on the ability of many different researchers to easily generate these cells from different starting material and to compare results across independent lines and perturbations.

2. The authors nicely present multi-electrode array (MEA) data on their cells (Extended Fig. 4), and show differences based on the maturity of the neurons. There are now reports of MEA studies using organoids (Passaro AP, et al. *Front Neurosci.* 2021. PMID: 33510616). It would be useful for the authors to compare their results with these results and to clarify the advantages of using

iCoMoNSCs. Furthermore, in the organoid model, the burst frequency seems to increase with longer incubation periods (Cleber A Trujillo, et al. Cell Stem Cell. 2019, PMID: 31474560). The authors' results seem to be similar for young and old compared to middle for some measurements, including bursts/min (Extended Fig 4f,g,h,I,k,u). Do the authors have an explanation for these potential differences?

3. The authors perform experiments to use lentiviruses to sparsely deliver HA-tagged TDP-43 into their iCoMoNSCs. They observe a decrease in expression of some known TDP-43 targets (e.g., STMN2 and UNC13A) and upregulation of a novel target – NPTX2. Because there is currently interest in comparing both TDP-43 loss-of-function and gain-of-function effects, in my opinion, it would be enormously helpful if the authors could perform the same type of experiment but using lentiviruses to deliver RNAi to knockdown TDP-43 instead of overexpression. Does this lead to the same or different effects on target genes? Because TDP-43 is aggregation-prone, its overexpression may cause aggregation and result in loss-of-function. It is important for the authors to determine if the NPTX2 upregulation is due to gain or loss of TDP-43 function.

4. The authors show that STMN2 and UNC13A were downregulated in their model. These two genes have recently been identified as cryptic splicing targets of TDP-43 (STMN2: Melamed et al, 2019; Klim et al., 2019, UNC13A: Ma et al., 2021; Brown et al., 2021). Is the downregulation of these genes caused by TDP-43-HA expression in the authors' model associated with cryptic exon inclusion or another mechanism? This should be a straightforward experiment to test with available material and the RT-PCR protocols that are associated with the manuscripts listed above.

5. The authors' results showing that TDP-43 pathology strongly correlates with NPTX2 upregulation is very compelling and exciting. I know space is always limited but I think that the images in Extended Data Fig. 10 are very beautiful, and the authors should include even more of these examples in the main text. One question is about specificity of NPTX2 upregulation? Is this just in situations with TDP-43 pathology or is it a general response to neuronal stress or degeneration? The authors could test this by performing the same immunostaining for NPTX2 but in other neurodegenerative disease samples w/o TDP-43 pathology (e.g., FTLT-tau; SOD1-ALS; FTLT-FUS, AD, HD, etc.). In other words, is the upregulation of NPTX2 a specific response to TDP-43 pathology?

6. An important question that this work raises is: does upregulation of NPTX2 in neurons cause neuronal dysfunction? Or is it a protective response? The authors show that in neurons with TDP-43 accumulation in their model, NPTX2 is upregulated, whereas it is at low levels elsewhere. They show similar results by analysis of human postmortem materials – upregulation of NPTX2 in the neurons with TDP-43 pathology and low level NPTX2 in neurons w/o TDP-43 pathology. Does NPTX2 make the neurons sicker or instead protected from degeneration? The authors speculate that NPTX2 upregulation could sensitize neurons to glutamate excitotoxicity (by enhancing glutamate receptor function). There are a couple ways the authors could consider testing this hypothesis:

a. Can knockdown of NPTX2 rescue any neuronal defects caused by TDP-43 up- or down-regulation in their system or in other cultured neurons in which TDP-43 accumulation causes NPTX2 to become upregulated?

b. A difficult but potentially impactful experiment would be to see if their model system has neuronal activity defects in the specific neurons with TDP-43-HA expressed, and if so, can these be rescued by targeting NPTX2? This would allow the authors to test if NPTX2 upregulation contributes to any TDP-43-dependent phenotypes in their system.

I note that both experiments are complicated because in addition to NPTX2 upregulation, there is also STMN2 and UNC13A downregulation, and potentially many other TDP-43 targets, which could all contribute to phenotypes. In some ways, the authors' iCoMoNSCs system might be the best

way to test the relative contributions of these targets.

7. Have the authors generated iCoMoNSCs from FTL or ALS patient cells? Do these have any molecular phenotypes (changes in gene expression) or physiological phenotypes (neuronal activity)? Do they exhibit TDP-43 pathology? I am sure that there will be great interest from the community in using this protocol to generate these cells from many different patient samples and any hints at if this is a fruitful direction would be informative.

8. What is the mechanism by which TDP-43 regulates NPTX2 mRNA levels? The iCLIP data are clean and convincing, providing evidence that TDP-43 can bind to NPTX2's 3'UTR and this function presumably becomes lost when it aggregates. Does TDP-43 binding have any effects on NPTX2 accumulation and potential secretion? Can the authors attempt to study this in an in vitro system? For example, would mutating the TDP-43 binding sites in NPTX2's 3'UTR or using a heterologous 3'UTR prevent up-regulation?

9. How do the authors interpret the discrepancy between their results of NPTX2 being upregulated in neurons with TDP-43 pathology (Fig. 4) and the recently published results of decreased NPTX2 in the CSF of symptomatic FTL patients (van der Ende et al., 2020)? Presumably, these patients have TDP-43 pathology as well.

10. Nptx2 knockout mice have been reported to have enhanced anxiety. Overexpression of Nptx2 in the hippocampus is sufficient to suppress stress-induced anxiety behaviors in mice (Simon Chang et al, Neuropsychopharmacology. 2018, PMID: 29844474). The authors should consider the implications of these results for interpreting their data, albeit with the caveat that mechanisms might be different between mouse and human. Since the authors have not performed functional assays to test if NPTX2 upregulation in iCoMoNSCs causes neuronal dysfunction or vulnerability, it might be too soon to conclude that NPTX2 contributes to pathogenicity. At the minimum, the authors should consider that NPTX2 expression might be a protective response to TDP-43 pathology.

Minor points

1. Although 7-8 months is a long time in culture, compared to the onset of neurodegenerative diseases in mid- to late-life, it is still young. The authors might want to re-consider whether "old" is an appropriate term.
2. Slight Typos; Line 915 (P24, L12) TDP-43p403 to TDP-43p403, Extended figure 5h; 404b2 to 409b2)
3. Is it possible to perform pseudo-time analysis to confirm the maturation of iCoMoNSCs in culture?
4. Please add quantification to Fig. 3c and Figure 4d.

Referee #2 (Remarks to the Author):

Towards the goal of studying TDP-43 pathology in a human context Hruska-Plochan and colleagues have developed a human in vitro system based on neural stem cells (termed iCoMoNSCs) derived from human iPSCs. These cells were differentiated and subsequently formed neuronal networks. The authors show a beautiful culture system with mature neurons and glial cells with synaptic connectivity and increased maturity over time – up to 7.5 months in vitro, which is very impressive. The authors compare expression data on their cultures to that of published data on organoids to show that there are resemblances. To study TDP-43 pathology, the authors overexpress wild-type TDP-43 in a minority of cells and identify potential cell loss, neuropathology and key molecules including NPTX2 which may be highly relevant for disease as also demonstrated by findings in post mortem tissues.

The findings in this study are original and the quality of the data and the presentation of the data is very high. The clarity of the writing is very good and the referencing appropriate.

Major comments:

1. While the cultures clearly are of very good quality it is not directly evident how this system is an improvement compared to organoids for example. Furthermore, for the study of ALS, it seems pertinent to generate layer 5 corticospinal motor neurons, which are the cells showing particular vulnerability in the brain in this disease, and these are apparently not being generated in either of the systems (organoids or iCoMoNSC system).

2. The authors mention that it is warranted to conduct long-term culture studies for TDP-43 pathology in a human disease context, yet the TDP-43 experiments are conducted on a 2-4 week basis. Based on the ability of the authors to generate long-term in vitro cultures I would have wished that the authors had used patient lines with TARDBP mutations and isogenic corrected controls and investigated pathology long-term (over 6 months) rather than using overexpression of wild-type TDP-43 and analyzing effects short term.

3. The authors show that after a slight overexpression of TDP-43-HA (wild-type) in a small subset of cells (about 2%), there was a continuous decrease in the number of neurons expressing the TDP-43 protein after 2 and 4 weeks (Figure 2d). The authors describe this as an indication that these neurons degenerated. While this may be the case, it would have been informative to see colocalization experiments of cells with TDP-43-HA and apoptosis markers to more conclusively show that this is what happened and exclude other possible scenarios, for example that the transgenic expression was silenced. This could also have been addressed in the single cell RNA sequencing experiments.

4. The observation that TDP-43^{p403/404}-positive inclusions emerge and amplify over time (in neurons lacking TDP-43-HA) is a compelling finding (see extended Data figure 8c,d) but also somewhat puzzling. Did this occur in cells other than neurons? Did it occur also in cultures where no TDP-43-HA overexpression was done or is it somehow related to the transgenic overexpression and spread of pathology? It would be nice if the authors could follow up and clarify this matter which is potentially very important in a disease pathology (spread) scenario.

5. In the single cell RNAseq experiment after TDP-43-HA overexpression (2 or 4 weeks) it would be nice to clarify which neurons were lost with TDP-43 overexpression (as shown in Figure 2d), that is, the specific identity of these cells that were particularly sensitive and the possible pathology which caused death. Currently in Figure (3d-e) it is not evident what cells are missing in the disease-context.

6. The finding that NPTX2 is upregulated and accumulating in neurons that express TDP-43-HA (Figure 4b,d) is a compelling finding. However, it is not yet clear if this upregulation is related to detrimental processes in the cells or compensatory protective mechanisms. As NPTX2 is accumulating in cells that can withstand a higher level of TDP-43 than normal without degenerating for 4 weeks (as shown by the authors in Figure 2b, half the cells that harbour TDP-43-HA overexpression are lost at 4 weeks) this may in fact be beneficial.

7. The authors speculate that NPTX2 malfunction in TDP-43 pathology may increase neuron vulnerability to glutamate toxicity in ALS and FTL and that the NPTX2 increase may trigger a pathological cascade leading to ALS. While this is an interesting thought it is still highly speculative. As the authors have a dynamic in vitro system where this could be studied, they should investigate longitudinally if TDP-43-HA harboring cells have increased levels of NPTX2 prior to degenerating and if this is the cause of the 50% cell loss in their system, and they should also treat the cells with ASOs against NPTX2 to see if neuronal loss can be prevented. Furthermore, they should misexpress NPTX2 alone to see if this is sufficient to induce ALS pathology in neurons without TDP-43 pathology.

Minor comment:

Text on top of cluster in Figure Extended Data Figure 1m is cut a bit (NR2F1)

Author Rebuttals to Initial Comments:

Referee #1 (Remarks to the Author):

In this manuscript, Magda Polymenidou and colleagues present a new method to induce self-renewing, homogeneous neural stem cells (iCoMoNSCs) from iPSCs. The iCoMoNSCs contain fewer postmitotic cells compared to existing human iPSCs-derived neural stem cells (NSCs) such as fetal-derived neural stem cell line SAI2, iPSCs-derived NSC line AF22 and Ctrl-7. Also, iCoMoNSCs can be induced into a cortical organoid-like cell population by differentiation. The degree of maturation of the induced neurons was well assessed not only by gene expression analysis but also by electrophysiological assays such as patch-clamp recording and microelectrode arrays, indicating that it is possible to generate a sufficiently mature cell population without creating 3D organoids. Furthermore, by overexpressing the FTD/ALS protein TDP-43 using lentiviruses in this model, they confirmed a decrease in expression in genes such as STMN2 and UNC13A, whose dysregulation have been recently connected to TDP-43 pathology in FTD/ALS, and also discover an upregulation in NPTX2, which the authors propose could be developed into a novel therapeutic target.

There are two main parts to this manuscript: 1) the establishment of a novel protocol/cell line of NSCs from iPSCs, which permits long-term culture and the ability to monitor neuronal activity and neuronal maturation state and 2) the discovery of a novel pathological candidate associated with TDP-43 pathology in FTD and ALS. Both components are exciting and, in my opinion, of great interest (I cannot imagine a single ALS laboratory that will not want to immediately start using this model system for their studies). And it comes at an exciting time in the ALS field, with the discovery of TDP-43 targets, many of which are human-specific, demanding human model systems to study them. Especially ones that have robust synaptic physiology and neuronal activity.

Our response: We thank the referee for accurately summarizing our data and for the positive assessment of our work.

I have several comments and suggestions for the authors to consider. Below I offer some comments on the stem cell experiments but since my expertise is in FTD/ALS mechanisms and TDP-43 I focus most of my suggestions on that aspect and trust the other referees to rigorously evaluate the model development parts.

1. It would be helpful if the authors could directly test the reproducibility and advantages of iCoMoNSCs over other systems. The authors carefully designed iCoMoNSCs and present compelling data using various cellular, molecular, and electrophysiological assays. The iCoMoNSCs system is of great interest because it has the potential to be induced into functionally mature neurons with more reproducibility and uniformity compared to some recently developed organoid model systems. However, it seems that the authors have only established a single cell line for their analyses. It would be of interest to consider generating multiple independent lines from existing iPSCs and/or patient-derived cells. The future utility of this model system depends on the ability of many different researchers to easily generate these cells from different starting material and to compare results across independent lines and perturbations.

Our response: We thank the referee for this important suggestion, which we have followed. Indeed we have acquired 21 iPSC lines from ALS patients and controls from the AnswerALS initiative¹, as well as 11

edited iPSC lines from the NIH/JAX initiative^{2,3} and an additional iPSC line from a collaborator. We have already successfully generated two novel iCoMoNSC lines, of which the strong morphological resemblance to the control iCoMoNSCs used in the current manuscript at the iCoMoNSC stage and during differentiation reflects the reproducibility of the generation protocol (**Rebuttal Figure 1**). We are planning to continue with the generation and characterization of more lines to further compare data and long-term phenotypes, as the referee recommends. However, these studies are time-consuming and will be the focus of another follow-up study, independent from this manuscript. We feel that the work presented in this manuscript demonstrates the utility of the iCoMoNSCs and we do not want to further delay publication of the study. Most importantly, we think that there is urgency in reporting the role of NPTX2 in human disease.

Rebuttal Figure 1. Generation and differentiation of iCoMoNSC derived from independent iPSC lines. Phase contrast and brightfield images of iCoMoNSCs were generated from a FUS R495X iPSC line (obtained through the NIH/JAX initiative^{2,3}, right panel) and generated and differentiated from a *PRNP* KO iPSC line (obtained through a collaborator, middle panels). The similarities between these novel iCoMoNSCs and the control iCoMoNSCs used in this manuscript both during proliferation and differentiation are striking. Passages: control iCoMoNSCs P6 (top), P24 (bottom); *PRNP* KO P11, FUS R495X P10. Scale bars, 150 μ m (upper left panel, see Extended Data Fig. 1b), 275 μ m (all other panels).

2. The authors nicely present multi-electrode array (MEA) data on their cells (Extended Fig. 4), and show differences based on the maturity of the neurons. There are now reports of MEA studies using organoids (Passaro AP, et al. Front Neurosci. 2021. PMID: 33510616). It would be useful for the authors to compare their results with these results and to clarify the advantages of using iCoMoNSCs. Furthermore, in the organoid model, the burst frequency seems to increase with longer incubation periods (Cleber A Trujillo, et al. Cell Stem Cell. 2019, PMID: 31474560). The authors' results seem to be similar for young and old

compared to middle for some measurements, including bursts/min (Extended Fig 4f,g,h,i,k,u). Do the authors have an explanation for these potential differences?

Our response: We thank the reviewer for the valuable comments and pointing us towards the existing organoid models. While 2D models are well established systems for cell line characterization, compound testing and developmental studies, evidenced by rich electrophysiological data, 3D models are more representative of the real cellular and tissue environment, as they include volume effects and better recapitulate the nature of human organs for drug testing. iNets offer the advantage of including most cell types that are present in brain organoids, so that the relevance for drug screening is most likely higher than that of 2D monocultures or co-cultures with a single glial subtype, e.g., astrocytes. Additionally, similar to typical 2D cell culture systems, iNets are suitable for tracking axons and studying network connectivity upon plating on HD-MEAs. In contrast, when using 3D models, one would only have access to neurons at the surface of the spherical organoid that are in contact with the electrodes, which would limit access to neuronal networks and axonal branches, all of which are arranged in 3D. Of course, it would be possible to slice organoids, but one would have, again, only access to signals and structures in 2D, so that a more extensive analysis of networks would not be possible, as large fractions of the 3D neuronal ensembles cannot be recorded. Considering advantages and disadvantages of both systems, we think that both 2D and 3D systems are useful and relevant models to study disease development and perform compound testing *in vitro*. Our iNets may combine advantages from 2D and 3D systems. We have now added the following sentences in the discussion of our revised manuscript to clarify these points: *“iNets have the advantage of containing a wide variety of cell types, including excitatory and inhibitory neurons and different types of glia, reflecting the heterogeneity of cellular types present in the human brain and brain organoids. Therefore, iNets are more relevant for CNS drug screening purposes than traditional 2D neural monocultures or co-cultures with a single glial subtype. Additionally, similar to typical 2D cell culture systems, we show that iNets are perfectly suitable for tracking axons and studying network connectivity upon plating on HD-MEAs, in contrast to 3D systems.*

Regarding the second point, we agree that longer incubation time on MEAs often entails changes in electrophysiological parameters, such as the burst frequency or firing rate. The nature of such changes, be it an increase or decrease in bursting or firing rates, is dependent on the studied cell types, as has been previously demonstrated on the same MEA platform by Ronchi *et al.*⁴. Here, it was not our main goal to compare electrophysiological metrics across development on MEAs. We wanted, instead, to investigate if different NSC maturation times could result in different degrees of connectivity, once cells had been transferred from the growth substrate and re-plated on the HD-MEAs. For 30% of HD-MEAs with young cultures, a network analysis could not be conducted, as bursts could not yet be detected, which is indicative of still-developing synaptic connections (similar to Fig. 2B of Trujillo, *et al.*⁵). More mature cultures (middle and old), instead, reliably showed network bursts. A plating of the cells directly on HD-MEAs from the beginning and culturing for up to 10 months, as has been performed in the cited paper, would most likely result in similar increases in the burst-frequency as observed by Trujillo *et al.*⁵ However, the exact cell composition would significantly influence the observed network bursting patterns.

3. The authors perform experiments to use lentiviruses to sparsely deliver HA-tagged TDP-43 into their iCoMoNSCs. They observe a decrease in expression of some known TDP-43 targets (e.g., STMN2 and UNC13A) and upregulation of a novel target – NPTX2. Because there is currently interest in comparing

both TDP-43 loss-of-function and gain-of-function effects, in my opinion, it would be enormously helpful if the authors could perform the same type of experiment but using lentiviruses to deliver RNAi to knockdown TDP-43 instead of overexpression. Does this lead to the same or different effects on target genes? Because TDP-43 is aggregation-prone, its overexpression may cause aggregation and result in loss-of-function. It is important for the authors to determine if the NPTX2 upregulation is due to gain or loss of TDP-43 function.

Our response: We thank the referee for raising this very important point that we have addressed with extensive additional experimentation and analyses in our revised manuscript (please see also response to points #4 and #8 of referee #1). Specifically, we performed the following two sets of experiments to address this point:

1. To directly address the effect of TDP-43 binding on the *NPTX2* 3'UTR, we used luminescence assays, in which the renilla luciferase gene was fused to the full 3'UTR of either *TARDBP* or *NPTX2*. *TARDBP* served as a control in this assay, since we have previously shown that TDP-43 protein binds this region and triggers a splicing event that targets the transcript to nonsense mediated decay thereby decreasing its levels⁶. Knockdown (KD) of TDP-43 significantly increased the levels of bioluminescence produced from renilla luciferase, fused to the full 3'UTR of either *TARDBP* or *NPTX2* (new **Fig. 4c**), while acute (72 hrs) overexpression (OE) of TDP-43-HA had the opposite effect (new **Fig. 4d**), indicating a bidirectional regulation of *NPTX2* mRNA by TDP-43, reminiscent of the regulation on its own mRNA. These experiments with acute KD or overexpression of TDP-43 in HEK293T cells indicated that the effect we described in our human cultures after several weeks of TDP-43-HA overexpression (i.e. *NPTX2* upregulation) mimics the effect of TDP-43 loss-of-function. This is in line with our interpretation that TDP-43 overexpression in iNet neurons progressively leads to insolubility, fragmentation and aggregation (see **Fig. 3c**), thereby interfering with its functionality.
2. To further understand the contribution of gain and loss-of-function mechanisms, we tested the impact of TDP-43 KD or OE on gene expression. First, we tested the efficacy of our new shRNAs against *TARDBP* and found that they efficiently reduced TDP-43 expression (See new **Extended Data Fig. 9c,d**, new **Extended Data Fig. 10b-d**, new **Supplementary Table 5**). Then, following the referee's recommendation, we generated new samples for both treatments (KD and OE), yet this time we targeted the majority of cells using a higher lentiviral titer and sequenced 150 million paired end reads of 150 base pairs each, with bulk RNAseq. Then, we estimated the differential gene expression in both conditions (log fold changes (logFC) of each treatment against their matching control, i.e. TDP-43-HA OE induction (ON) vs no induction (OFF); and TDP-43 KD vs a control shRNA, and compared them to each other, as well as to a previously published dataset that reported the transcriptome of neurons with nuclear clearance of TDP-43 in the brains of ALS-FTLD patients⁷ (new **Extended Data Fig. 11a**). Our analyses showed that a subset of transcripts was altered in both iNets with TDP-43-HA-expression and TDP-43 KD (17% for downregulated genes, 14% for upregulated genes), including *NPTX2* (new **Extended Data Fig. 11b**). Even though only a small fraction of transcripts is changed in both iNets with altered TDP-43 levels and ALS-FTLD patient brains (new **Supplementary Tables 7,8**), our analysis shows that both overexpression and downregulation of TDP-43 recapitulate part of the transcriptomic alterations

observed in affected neurons in ALS-FTLD patients (new **Extended Data Fig. 11**, new **Supplementary Table 7,8**). Importantly, our Western blot and immunofluorescence analyses show that TDP-43 KD leads to reduced NPTX2 protein levels (**Extended Data Fig. 10c-d**). We want to thank the referee for his/her suggestion that inspired this new section in our study. We will make all these datasets publicly available upon publication and hope that the scientific community will profit by cross-examining and reanalyzing them.

4. The authors show that *STMN2* and *UNC13A* were downregulated in their model. These two genes have recently been identified as cryptic splicing targets of TDP-43 (*STMN2*: Melamed et al, 2019; Klim et al., 2019, *UNC13A*: Ma et al., 2021; Brown et al., 2021). Is the downregulation of these genes caused by TDP-43-HA expression in the authors' model associated with cryptic exon inclusion or another mechanism? This should be a straightforward experiment to test with available material and the RT-PCR protocols that are associated with the manuscripts listed above.

Our response: The referee raises an important point that we have extensively addressed experimentally. We first checked the *STMN2* and *UNC13A* transcripts in our original TDP-43-HA-expressing cell cluster (scRNA-seq), but we were unable to detect cryptic exons. Since the shallow sequencing depth of scRNA-seq reduces our sensitivity (even though we broadened our search for cryptic exon events for all cell barcodes, including “unhealthy” - failing the QC - cells), we then explored cryptic exon inclusion events in our deeply sequenced bulk RNA-seq (150 million paired end reads of 150 bp) from iNets with TDP-43 OE and KD datasets (please see also response to points #3 and #8 of referee #1). We found detectable *UNC13A* and *STMN2* cryptic exons in our TDP-43 KD samples (new **Extended Data Fig. 10e**), as well as significant downregulation of *PFKP*, *RCAN1*, *SELPLG* and *ELAVL3* (new **Supplementary Table 5**), which were previously shown to be downregulated upon TDP-43 KD in iPSC-derived human motor neurons⁸, confirming that the classical downstream consequences of TDP-43 KD also take place in iNets. In contrast to the direct KD of TDP-43, however, we could not detect the previously reported cryptic exons in iNets with TDP-43-HA OE, possibly due to their low levels and rapid degradation via the nonsense mediated decay pathway, as recently shown in other studies⁹.

A recent study, published after we received the referees' comments, showed that a consequence of TDP-43 overexpression is the exclusion of constitutive exons¹⁰, in line with an independent study¹¹. The authors of Carmen-Orozco *et al*¹⁰ argued that excessive levels of nuclear TDP-43 triggers constitutive exon skipping (“skiptic exons”) that is detected in some human brains, but is not correlated with disease, unlike the incorporation of cryptic exons that occurs after loss of TDP-43. To understand if our iNet model merely reports the consequences of high nuclear TDP-43 levels, we performed splicing analysis of our TDP-43-HA overexpression bulk RNA-seq datasets. We found that there is no significant overlap (31 out of 82 with a moderate/low cut-off) with the reported transcripts carrying skiptic exons (new **Extended Data Fig. 10f**, new **Supplementary Table 6**), suggesting a distinct transcriptional profile in our iNets with TDP-43 accumulation.

5. The authors' results showing that TDP-43 pathology strongly correlates with NPTX2 upregulation is very compelling and exciting. I know space is always limited but I think that the images in Extended Data Fig. 10 are very beautiful, and the authors should include even more of these examples in the main text. One question is about specificity of NPTX2 upregulation? Is this just in situations with TDP-43 pathology

or is it a general response to neuronal stress or degeneration? The authors could test this by performing the same immunostaining for NPTX2 but in other neurodegenerative disease samples w/o TDP-43 pathology (e.g., FTLN-tau; SOD1-ALS; FTLN-FUS, AD, HD, etc.). In other words, is the upregulation of NPTX2 a specific response to TDP-43 pathology?

Our response: We appreciate the positive assessment of the image gallery shown in Extended Data Fig. 10 and at the reviewer's request we have added an extra panel to the main Figure 4 (FTLN-C: Hippocampus, see **Fig. 4g**). Following the reviewer's suggestion to examine the specificity of NPTX2 upregulation to TDP-43 pathology, we have expanded our neuropathological analysis to include a total of 20 cases of neurodegeneration. In addition, we utilized our iNet model to probe the specificity of NPTX2 accumulation to TDP-43 pathology. Specifically, we performed the following two sets of experiments:

1. We immunolabeled post-mortem human brain sections of 12 additional patients with other neurodegenerative proteinopathies, namely FTLN-FUS ($N=4$), FTLN-Tau ($N=4$) and Alzheimer's disease (AD, $N=3$), for NPTX2 and disease-specific inclusions (FUS and phosphorylated tau, respectively). In contrast to the clear accumulation of NPTX2 in neurons with TDP-43 inclusions that we had reported in FTLN-TDP and ALS patients in the original manuscript (**Fig. 4f-h**, **Extended Data Fig. 14a-d**), the labeling pattern of NPTX2 did not differ between neurons with FUS or tau inclusions and their unaffected neighboring neurons (new **Fig. 4i-k**, new **Extended Data Fig. 14e,f**) in FTLN-FUS, FTLN-Tau and AD cases. Strikingly, in AD patients with concomitant TDP-43 aggregation ($N=2$), a co-pathology present in a substantial subset of AD patients¹²⁻¹⁶, we observed NPTX2 accumulation in neurons with TDP-43 inclusions, but not neurons with tau inclusions or those free of aggregates (new **Fig. 4l**, new **Extended Data Fig. 14g**). Taken together, NPTX2 accumulation in patients with neurodegenerative proteinopathies is specific to TDP-43 pathology and does not occur as a result of other protein aggregations, including FUS or tau.
2. To further probe the specificity of NPTX2 upregulation, we went back to our iNet model and overexpressed either TDP-43-HA or HA-FUS (in both cases, the HA tag is fused to the prion-like domain) for 1 or 2 weeks using the same lentiviral backbone for transgene expression. Immunofluorescent labeling of NPTX2 revealed a stark difference between the conditions at both timepoints: low levels of NPTX2 were detected in the HA-FUS- compared to TDP-43-HA-expressing iNet neurons (**Extended Data Fig. 13**). Therefore, we conclude that NPTX2 accumulation is specific to TDP-43 misregulation in both patient brains and iNet cultures.

6. An important question that this work raises is: does upregulation of NPTX2 in neurons cause neuronal dysfunction? Or is it a protective response? The authors show that in neurons with TDP-43 accumulation in their model, NPTX2 is upregulated, whereas it is at low levels elsewhere. They show similar results by analysis of human postmortem materials – upregulation of NPTX2 in the neurons with TDP-43 pathology and low level NPTX2 in neurons w/o TDP-43 pathology. Does NPTX2 make the neurons sicker or instead protected from degeneration? The authors speculate that NPTX2 upregulation could sensitize neurons to glutamate excitotoxicity (by enhancing glutamate receptor function). There are a couple ways the authors could consider testing this hypothesis:

a. Can knockdown of NPTX2 rescue any neuronal defects caused by TDP-43 up- or down-regulation in their system or in other cultured neurons in which TDP-43 accumulation causes NPTX2 to become upregulated?

b. A difficult but potentially impactful experiment would be to see if their model system has neuronal activity defects in the specific neurons with TDP-43-HA expressed, and if so, can these be rescued by targeting NPTX2? This would allow the authors to test if NPTX2 upregulation contributes to any TDP-43-dependent phenotypes in their system.

I note that both experiments are complicated because in addition to NPTX2 upregulation, there is also STMN2 and UNC13A downregulation, and potentially many other TDP-43 targets, which could all contribute to phenotypes. In some ways, the authors' iCoMoNSCs system might be the best way to test the relative contributions of these targets.

Our response: We thank the referee for these thoughtful suggestions, which inspired a whole new set of experimentation that we present in our new **Figure 5** and new **Extended Data Fig. 15** (please see also our response to point #10, reviewer #1 and points #6 and #7, reviewer #2). First, to understand if *NPTX2* upregulation is toxic to neurons, we directly overexpressed NPTX2-HA in iNets and compared its effect on neuronal counts to TDP-43-HA overexpression, the latter with variable levels, achieved with differential promoter or regulatory (3'UTR) control. We also used iNets to overexpress HA-FUS, which we found inert under the same experimental conditions and timeframe. Our data indicate that direct overexpression of NPTX2, which leads to a similar accumulation of the protein in the cytoplasm and neuronal processes as we observed downstream of TDP-43 overexpression, is neurotoxic in iNets. Importantly, and in line with this referee's predictions, this direct NPTX2 toxicity was milder compared to TDP-43-triggered toxicity, which we found to correlate with the levels of TDP-43 overexpression. This finding suggested that NPTX2 is one of several mediators of TDP-43 toxicity, and is potentially acting synergistically with STMN2, UNC13A (and likely other TDP-43 targets) in disease. However, we were excited to see that, when we performed the rescue experiments that the referee suggested, correcting the levels of just NPTX2 with shRNA significantly rescued TDP-43 neurotoxicity in iNets. This result indicates that, even though other TDP-43 targets including STMN2 and UNC13A undoubtedly contribute to neurotoxicity, correcting just NPTX2 levels is beneficial for neurons. These data identify NPTX2 as an important modifier of TDP-43 toxicity in human neurons. Given the known limitations of shRNA-mediated gene silencing¹⁷, including off-target toxicity, we find this consistent and significant rescue of iNets neurons particularly encouraging. Future efforts will focus on testing alternative regimens for correcting NPTX2 in models of TDP-43 proteinopathy, including antisense oligonucleotides.

We have not yet addressed the potential neuronal activity defects triggered by elevated NPTX2 levels (or TDP-43 accumulation). This is a crucial next step that we are now starting to address in collaboration with the team of Martin Mueller, an expert electrophysiologist at the University of Zurich. We also plan to test the potential synergistic effect of *NPTX2*, *STMN2* and *UNC13A* misregulation on neurophysiology. However, these studies are time-consuming and extensive and we hope that the referee will recognize that this work is the next chapter of understanding the role of NPTX2 in disease and will be the topic of a follow-up study. Given the lengthy revision of the current work, we do not want to further delay publication of our current findings.

7. Have the authors generated iCoMoNSCs from FTLD or ALS patient cells? Do these have any molecular phenotypes (changes in gene expression) or physiological phenotypes (neuronal activity)? Do they exhibit TDP-43 pathology? I am sure that there will be great interest from the community in using this protocol to generate these cells from many different patient samples and any hints at if this is a fruitful direction would be informative.

Our response: We agree with the referee that generating iNets from FTLD or ALS patient cells will be very interesting and we hope that indeed this protocol will be adapted by other labs in the field. To this end, we have already acquired 21 iPSC lines from ALS patients and controls from the AnswerALS initiative¹, as well as 11 edited iPSC lines from the NIH/JAX initiative^{2,3} and an additional iPSC line from a collaborator (please see point #1 of referee #1, and point #2 of referee #2). We have successfully generated two novel iCoMoNSC lines, of which the strong morphological resemblance to the control iCoMoNSCs used in the current manuscript at the iCoMoNSC stage and during differentiation reflects the reproducibility of the generation protocol (**Rebuttal Figure 1**). However, the full phenotypic characterization of these lines and the respective iNets is time-consuming and beyond the scope of the current manuscript.

8. What is the mechanism by which TDP-43 regulates NPTX2 mRNA levels? The iCLIP data are clean and convincing, providing evidence that TDP-43 can bind to NPTX2's 3'UTR and this function presumably becomes lost when it aggregates. Does TDP-43 binding have any effects on NPTX2 accumulation and potential secretion? Can the authors attempt to study this in an in vitro system? For example, would mutating the TDP-43 binding sites in NPTX2's 3'UTR or using a heterologous 3'UTR prevent up-regulation?

Our response: Following the reviewer's suggestion, we further investigated the mechanistic link between TDP-43 dysregulation and *NPTX2* upregulation and misaccumulation. As pointed out by the reviewer, the loss of iCLIP cross-links (dataset from Tollervey et al¹⁸) in the *NPTX2* 3'UTR of FTLD patients compared to control brains (**Fig. 4b** and new **Extended Data Fig. 8k**) suggested a 3'UTR-dependent regulation of *NPTX2* levels by TDP-43. To confirm this experimentally, we set up a novel 3'UTR dual luciferase reporter assay for *NPTX2* in the lab and compared it to the 3'UTR of *TARDBP* in an identical setup (see also point #3 of referee #1). In this setup, either of the 3'UTR sequences were cloned downstream of renilla luciferase. The renilla-derived bioluminescence signal is therefore modulated by 3'UTR-mediated alterations in expression levels. Firefly luciferase served as a normalization control. We performed four sets of experiments using transfection and/or transduction in HEK293 cells:

1. We modified endogenous TDP-43 levels by shRNA-mediated KD (new **Extended Data Fig. 9c,d**), which resulted in an increased *NPTX2* 3'UTR reporter signal (new **Fig. 4c**). This proved that TDP-43 controls *NPTX2* expression levels via its 3'UTR. This accumulation of NPTX2 was then confirmed at the protein level by immunofluorescence and Western blot (new **Extended Data Fig. 10c,d**). Importantly, in another set of experiments (please see also response to points #3 and #4 of referee #1) TDP-43 KD was analyzed by bulk RNA-seq which also confirmed significantly elevated RNA levels of endogenous *NPTX2* (new **Extended Data Fig. 10b**).

2. Next, we performed acute overexpression experiments, overexpressing TDP-43-HA and found a drop in *NPTX2* 3'UTR reporter signal (new **Fig. 4d**), further confirming that TDP-43 regulates *NPTX2* levels by means of its 3'UTR.
3. Following the reviewer's suggestion, we additionally mutated the 3'UTR of *NPTX2* by swapping all GUs to AAs, aiming to provide another line of evidence showcasing the direct regulation of *NPTX2* levels by TDP-43 binding to its 3'UTR. However, this *NPTX2* AA mutant had overall strongly reduced bioluminescence levels from the renilla open-reading frame (**Rebuttal Figure 2**), indicating loss of stability of the 3'UTR. While we did not continue with this construct, we do note that we did not observe a consistent change in this construct upon TDP-43 OE or KD.
4. We then wondered which TDP-43 domain was responsible for the regulating effect observed in setup 1 and 2 above. Typically, TDP-43 binds RNA via its RNA Recognition Motifs (RRM1 and RRM2, amino acids 106-259¹⁹). Therefore, we silenced endogenous TDP-43 with shRNA, which leads to increased *NPTX2* 3'UTR reporter signal (new **Fig. 4c**). Then we attempted to rescue this effect in the same cells by introducing either wild-type (WT) or TDP-43 with carrying mutations in its two RRMs rendering them unable to bind RNA and therefore nonfunctional (RRMm, F147A, F149A, F194A, F229A, F231A)²⁰. Whereas WT TDP-43 rescued the effect of TDP-43 KD on the reporter signal (no net effect), the RRMm TDP-43 variant did not (new **Extended Data Fig. 9e**). This indicates that the regulation of *NPTX2* is mediated by direct binding of TDP-43 via its RRMs on the 3'UTR of *NPTX2*.

We note that for setup 1, 2 and 4, the *TARDBP* 3'UTR reporter was examined in parallel to the *NPTX2* 3'UTR reporter and showed the regulation in the same direction, but stronger potency upon manipulation of TDP-43 levels, confirming the specificity of our luciferase assays. However, the molecular mechanism of the regulation on *TARDBP* and *NPTX2* 3'UTR is likely distinct. We are currently investigating the mechanisms of this regulation, which is going to be a part of another study.

Taken together, these experiments show that TDP-43 binds the 3'UTR of *NPTX2* via its RRM and that this binding regulates the levels of *NPTX2*. Notably, this conclusion is further corroborated by the absence of iCLIP TDP-43 cross-links in the 3'UTR of *Nptx2* and the experimentally confirmed lack of *Nptx2* accumulation in transgenic TDP-43-HA-expressing primary mouse neurons (new **Extended Data Fig. 9a,b**).

Rebuttal Figure 2. NPTX2 3'UTR AA mutant loses stability. Dual luciferase assay showing the behavior of *TARDBP*, *NPTX2* and *NPTX2* AA 3'UTR mutant upon TDP-43 KD (a) or TDP-43-HA overexpression (b). One-way ANOVA followed by Tuckey's multiple comparisons test. Significance is shown only between the pairs. Note that the *NPTX2* AA 3'UTR reporter conditions had significantly lower luminescence compared to all other conditions in both setups (significance stars not shown in figure), reflecting loss of stability. Differently gray scaled data points are independent experiments. The first 4 conditions for each setup are shown in the new Fig. 4c,d in the revised manuscript.

9. How do the authors interpret the discrepancy between their results of NPTX2 being upregulated in neurons with TDP-43 pathology (Fig. 4) and the recently published results of decreased NPTX2 in the CSF of symptomatic FTLD patients (van der Ende et al., 2020)? Presumably, these patients have TDP-43 pathology as well.

Our response: This is an important point that initially perplexed us as well. As the reviewer correctly points out, recent studies reported decreased NPTX2 levels in the CSF of FTLD patients and found that they are particularly decreased at the phenoconversion point^{21,22}, suggesting that NPTX2 CSF levels might represent a disease biomarker. This was interpreted by the investigators as a molecular signature of synaptic loss occurring at this critical point in disease progression. In line with this interpretation, the studies show similar findings not only in FTLD-TDP patients, but also FTLD-Tau, which we now show does not lead to NPTX2 accumulation in affected neurons (new Fig. 4i,j-l and new Extended Fig. 14e,f). An important aspect that explains the apparent discrepancy between our results and the CSF studies is that we describe NPTX2 upregulation *downstream* of TDP-43 pathology and therefore accumulation within *the specific cells that are affected in disease*, which represent a small minority, as previously reported (less than 2%)⁷. In contrast, CSF levels report secreted NPTX2 protein from the entire CNS tissue, in which the effect of the cells with TDP-43 pathology and NPTX2 accumulation is likely diluted.

To test this hypothesis, we performed Western blot analysis of total frontal cortex brain homogenates from 14 FTLN patients and 2 control patients without neurodegeneration. Our results clearly demonstrate that the overall levels of NPTX2 in FTLN are significantly decreased (**Rebuttal Fig. 3**), in line with the notion that the decrease in NPTX2 protein levels detected in CSF^{21,22} is a consequence of a global loss of synapses, occurring as the disease progresses. Therefore, we conclude that while TDP-43 malfunction triggers NPTX2 upregulation and accumulation in the minority of neurons with TDP-43 pathology (see **Fig. 4** and **Extended Data Fig. 14**), the overall synaptic loss that occurs in neurodegeneration results in the total decrease in NPTX2 protein in total brain homogenate and CSF samples. We do not know at this time if the two events are in any way related (through synaptic homeostatic mechanisms for example), and this will be the focus of our future studies on the mechanism of neuronal dysfunction triggered by NPTX2 misregulation (please see point #6 of referee #1, above).

We have now added the following sentence in our discussion to address this point: *“We do not know at this time if the NPTX2 accumulation that we observed in the minority of neurons with TDP-43 pathology is in any way related with the global decrease in NPTX2 levels with age⁵⁹ and in dementia⁵⁸⁻⁶¹, which is thought to indicate overall synaptic loss. Yet, we cannot exclude the possibility of an interplay between the two events via mechanisms of homeostatic synaptic plasticity, for example.”*

Interestingly, our Western blot analysis revealed a striking change in the NPTX2 band pattern in the total brain homogenates from FTLN patients. Specifically the higher 65kDa fuzzy band likely representing a glycosylated form of NPTX2, which is a N-glycosylated secreted protein²³, is partially or fully lost in FTLN patients, accompanied by appearance of several lower molecular bands, potentially resulting from proteolytic cleavage (**Rebuttal Figure 3a, lower panel**). We plan to further investigate this aspect in a follow-up study.

Rebuttal Figure 3. Total NPTX2 levels are significantly reduced in brain homogenates of FTLD patients. **a**, Western blot (WB) analysis of 14 total brain homogenates from FTLD Type A and Type C patients and 2 non-neurodegenerative controls using a phospho-specific (pS409/410) anti-TDP-43 antibody to demonstrate pTDP-43 pathology in all patient samples but not in the 2 controls (quantification shown in **b**). Interestingly, NPTX2 in FTLD patient brain homogenates showed changes in the NPTX2 band pattern with the higher 65kDa band disappearing (likely a glycosylated form) accompanied by appearance of multiple lower molecular weights (likely proteolytic cleavage fragments). Importantly, the total NPTX2 signal was significantly downregulated in total brain homogenates of FTLD patients (**c**). SOD1 was used as a control for all Western blots (each marker has its own SOD1 re-probe). Unpaired *t* tests were used to determine statistical significance between the FTLD and control brain homogenate Western blots.

10. Nptx2 knockout mice have been reported to have enhanced anxiety. Overexpression of Nptx2 in the hippocampus is sufficient to suppress stress-induced anxiety behaviors in mice (Simon Chang et al, Neuropsychopharmacology. 2018, PMID: 29844474). The authors should consider the implications of these results for interpreting their data, albeit with the caveat that mechanisms might be different between mouse and human. Since the authors have not performed functional assays to test if NPTX2 upregulation in iCoMoNSCs causes neuronal dysfunction or vulnerability, it might be too soon to conclude that NPTX2 contributes to pathogenicity. At the minimum, the authors should consider that NPTX2 expression might be a protective response to TDP-43 pathology.

Our response: We thank the reviewer for raising this important point, which we explored further in our revised manuscript, both bioinformatically and experimentally. There are two important aspects in this comment which we discuss below:

1. **Is the TDP-43/NPTX2 regulation conserved in mouse?** Our new data shown in **Extended Fig. 9a-b** suggests that TDP-43 regulation of NPTX2 is human-specific. Given our reanalysis of Tollervey 2011's iCLIP-seq data¹⁸ pointing to TDP-43 binding to the *NPTX2* 3'UTR, and the known sequence specificity of TDP-43 binding to RNA, we reasoned that the potentially different mouse-human regulatory features could be explored at two levels: either in terms of TDP-43 species-specific binding, or sequence conservation. To explore the potential differential binding of TDP-43 to *Nptx2*/*NPTX2* 3'UTR in mouse and human, we re-analyzed our mouse TDP-43 CLIP-seq data⁶, and observed sparse binding, mostly downstream to the *Nptx2* 3'UTR. To compare the mouse TDP-43 binding sites in *Nptx2* to the human binding sites in *NPTX2*, we translated the mouse binding sites to human coordinates using liftOver²⁵, and compared those to the (human) Tollervey *et al* 2011 data¹⁸. We found that the mouse TDP-43 binding sites do not overlap to those in humans, suggesting either a weak sequence conservation or distinct regulatory mechanisms (new **Extended Data Fig. 9a**). We evaluated the sequence conservation of the *NPTX2* 3'UTR within mouse and human, and also across placentals²⁶. Across placental mammals, the *NPTX2* 3' UTR PhyloP scores are higher than in *NPTX2* introns and lower than in *NPTX2* exons, suggesting the *NPTX2* 3' UTR is conserved, meaning less conserved than *NPTX2* exons, but more than *NPTX2* introns. As for a direct mouse vs human pairwise conservation comparison, we noticed local spots of low sequence conservation within the 3'UTR in close proximity to human (GUGU)_n motifs and TDP-43 binding sites from Tollervey *et al* 2011¹⁸.

In agreement with the predictions from our bioinformatic analyses, we have directly tested the effect of TDP-43-HA overexpression – under the same inducible promoter that we have used in the human iNets – on primary mouse neurons. We show that, in contrast to human neurons, TDP-43-HA accumulation does not trigger *Nptx2* upregulation and accumulation in mouse neurons (new **Extended Data Fig. 9b**). This result is similar to what has been reported for other key TDP-43 RNA targets, *STMN2* and *UNC13A*.

2. **Is NPTX2 upregulation pathogenic?** This is a crucial question that we experimentally addressed in our revised manuscript. As shown in new **Figure 5** and **Extended Data Figure 15**, we show that direct overexpression of *NPTX2* triggers neurotoxicity in iNets, whereas reducing the levels of *NPTX2* by shRNA while simultaneously overexpressing TDP-43-HA partially rescues neurons. Please see also our response to point #6, reviewer #1 and points #6 and #7, reviewer #2 for more details on these new experiments.

Minor points

1. Although 7-8 months is a long time in culture, compared to the onset of neurodegenerative diseases in mid- to late-life, it is still young. The authors might want to re-consider whether "old" is an appropriate term.

Our response: We thank the referee for this comment and we agree that the culture age is not directly comparable to the age of a human brain. We have thought long about the proper nomenclature for these

cultures and we think that “young”, “middle” and “old” along with the actual time in differentiation, as we have explained in our manuscript, best represent the different conditions analyzed. We would, therefore, prefer to maintain these terms.

2. Slight Typos; Line 915 (P24, L12) TDP-43p403 to TDP-43p403, Extended figure 5h; 404b2 to 409b2)

Our response: We thank the referee for noticing these mistakes, which we have corrected in our revised manuscript.

3. Is it possible to perform pseudo-time analysis to confirm the maturation of iCoMoNSCs in culture?

Our response: We thank the referee for her/his suggestion which we have followed. We agree that a trajectory inference method might shed light into the relative positioning of our clusters along a pseudotime. We inferred the underlying trajectory, rooted them in our neuronal stem cells, and explored the pseudotime distribution across clusters (new **Extended Data Fig. 7**). As depicted in our new **Extended Data Figure 7c**, we found a noticeable concordance between the median estimated pseudotime within clusters, and our original cluster annotation. Specifically, we ran a single-cell trajectory analysis with monocle version 3_1.0.0²⁷ and found four independent trajectories associated to cell identities: NSCs, neurons, astrocytes, and pericyte-like. After learning the trajectories graph, we ordered cells on a pseudotime rooting the trajectory on NSCs (**Extended Data Fig. 7b**), and explored the positioning of our manually annotated, unsupervised clusters along the pseudotime. The transcriptional maturation of the differentiating iNets was congruent with pseudotime analysis.

4. Please add quantification to Fig. 3c and Figure 4d.

Our response: At the reviewer’s request, we have added quantification of Fig. 3c and 4d, which can be found in the revised manuscript in our new **Extended Data Fig. 8d** and **12b**, respectively. Both quantifications fortify our, previously descriptive, conclusions, being that there is a progressive loss of solubility and concurrent aggregation and fragmentation at the expense of full-length protein levels in our TDP-43-HA-expressing cultures (**Fig. 3c, Extended Data Fig. 8c,e**) and that the vast majority (around 80%) of TDP-43-HA expressing neurons is also hallmarked by aberrant NPTX2 accumulation (**Fig. 4e, Extended Data Fig. 12a**).

Referee #2 (Remarks to the Author):

Towards the goal of studying TDP-43 pathology in a human context Hruska-Plochan and colleagues have developed a human in vitro system based on neural stem cells (termed iCoMoNSCs) derived from human iPSCs. These cells were differentiated and subsequently formed neuronal networks. The authors show a beautiful culture system with mature neurons and glial cells with synaptic connectivity and increased maturity over time – up to 7.5 months in vitro, which is very impressive. The authors compare expression data on their cultures to that of published data on organoids to show that there are resemblances. To study TDP-43 pathology, the authors overexpress wild-type TDP-43 in a minority of cells and identify

potential cell loss, neuropathology and key molecules including NPTX2 which may be highly relevant for disease as also demonstrated by findings in post mortem tissues.

The findings in this study are original and the quality of the data and the presentation of the data is very high. The clarity of the writing is very good and the referencing appropriate.

Our response: We gratefully acknowledge the referee's positive assessment of our work.

Major comments:

1. While the cultures clearly are of very good quality it is not directly evident how this system is an improvement compared to organoids for example. Furthermore, for the study of ALS, it seems pertinent to generate layer 5 corticospinal motor neurons, which are the cells showing particular vulnerability in the brain in this disease, and these are apparently not being generated in either of the systems (organoids or iCoMoNSC system).

Our response: The comparison between iNets versus organoids and also versus faster methods for neuronal differentiation from iPSC, such as the broadly used i3 Neurons, is an important message of our study. We think that each of these methodologies have advantages and disadvantages and will be used in a complementary fashion to address specific scientific questions. In particular, iNets offer the possibility of producing highly mature and functional neurons with unique longevity in a highly reproducible manner, as our scRNA-seq indicates. Moreover, since there is no spatial organization (unlike organoids), iNets can easily be subcultured and then analyzed by several downstream readouts (routinely done in our lab). We do acknowledge the lack of layer 5 corticospinal motor neurons and propose that our current system resembles human cortical brain regions, in line with our comparison to cortical organoids (**Fig. 2d-f**). As such iNets represent a model for FTLN or ALS/FTLN processes, rather than pure ALS. To summarize, we see the following as strong advantages of iNets compared to other models:

- Longevity, >7 months in culture
- Neuronal maturation and network activity
- Possibility to study neuronal aging in culture
- Self-organized multicellularity, including excitatory & inhibitory neurons and glia
- Highly reproducible cell composition between experiments
- Amenable to subculture at any time point
- Growth in variable surfaces, including for functional analyses (i.e. HD-MEA)
- Easily scalable

We have now added the following sentences in the discussion of our revised manuscript to directly comment on this comparison: *"iNets have the advantage of containing a wide variety of cell types, including excitatory and inhibitory neurons and different types of glia, reflecting the heterogeneity of cellular types present in the human brain and brain organoids. Therefore, iNets are more relevant for CNS drug screening purposes than traditional 2D neural monocultures or co-cultures with a single glial subtype. Additionally, similar to typical 2D cell culture systems, we show that iNets are perfectly suitable for tracking axons and studying network connectivity upon plating on HD-MEAs, in contrast to 3D systems."*

2. The author mention that it is warranted to conduct long-term culture studies for TDP-43 pathology in a human disease context, yet the TDP-43 experiments are conducted on a 2-4 week basis. Based on the ability of the authors to generate long-term in vitro cultures I would have wished that the authors had used patient lines with TARDBP mutations and isogenic corrected controls and investigated pathology long-term (over 6 months) rather than using overexpression of wild-type TDP-43 and analyzing effects short term.

Our response: We thank the referee for this important suggestion, which we have followed (please see also our response to points #1 and #7, reviewer #1). Indeed we have acquired 21 iPSC lines from ALS patients and controls from the AnswerALS initiative¹, as well as 11 edited iPSC lines from the NIH/JAX initiative^{2,3} and an additional iPSC line from a collaborator. We have already successfully generated two novel iCoMoNSC lines, of which the strong morphological resemblance to the control iCoMoNSCs used in the current manuscript at the iCoMoNSC stage and during differentiation reflects the reproducibility of the generation protocol (**Rebuttal Figure 1**). We are currently working on the generation and characterization of more lines and we fully intend to characterize the long-term phenotypes of disease-linked iNets, as the referee recommends. However, these studies are time-consuming and will be the focus of another follow-up study, independent from this manuscript. We feel that the work presented in this manuscript demonstrates the utility of the iCoMoNSCs and we do not want to further delay publication of the study. Most importantly, we think that there is urgency in reporting the role of NPTX2 in human disease.

3. The authors show that after a slight overexpression of TDP-43-HA (wild-type) in a small subset of cells (about 2%), there was a continuous decrease in the number of neurons expressing the TDP-43 protein after 2 and 4 weeks (Figure 2d). The authors describe this as an indication that these neurons degenerated. While this may be the case, it would have been informative to see colocalization experiments of cells with TDP-43-HA and apoptosis markers to more conclusively show that this is what happened and exclude other possible scenarios, for example that the transgenic expression was silenced. This could also have been addressed in the single cell RNA sequencing experiments.

Our response: We appreciate the reviewer's criticism, which we have addressed both bioinformatically and experimentally in our revised manuscript.

Bioinformatic analysis: Regarding the silencing of the vector in our scRNA-seq, we have quantified the expression levels of the 3' long terminal repeats (3'LTRs) of the transfer vector (**Rebuttal Figure 4 top panel**) as well as rtTA, a component of our monocistronic all-in-one Tet-On expression cassette (**Rebuttal Figure 4 bottom panel**), in order to estimate the transduction rate vs TDP-43-HA rate. We found that while all clusters were transduced by the LV (as evidenced by the 3'LTR and rtTA expression in 4 week OFF samples), only cluster 12 cells showed significant increase in expression of rtTA (which is a prerequisite of TDP-43-HA expression) and 3'LTR upon DOX addition. This indicates that cluster 12 originated from transduced neurons of all clusters (**Rebuttal Figure 4**), as cluster 12 cells are exclusively neurons (**Fig. 3d**).

In terms of the continuous decrease of TDP-43-HA expressing cells in DOX ON 4W vs 2W, we detect multiple changes in cell-type specific abundances (cell composition)²⁸ aside from the depletion of cluster 12. Namely, we detect a significant enrichment of maturing excitatory neurons (clusters 11, 4, 9) and a decrease in maturing/apoptotic (cluster 12) (**Rebuttal Figure 5**). This might reflect the gradual replacement of the dying cluster 12 neurons, which originate from all transduced neurons - including maturing excitatory neurons (see **Rebuttal Figure 4**) - by neurons in clusters 11, 4 and 9. - This gap that would have otherwise been filled by cluster 12 neurons and glia/astrocytes (cluster 6) as compared to 2W.

Rebuttal figure 4. Lentiviral vector component expression quantification. Top panel. 3’LTR expression of the LV transfer vector and rtTA expression of our monocistronic all-in-one Tet-On expression cassette (**bottom panel**) was quantified across cells from all clusters and all sample types showing that all cell clusters/cell types were transduced by our LV but only cluster 12 showed increased expression upon DOX induction. Color scale depicts the TDP-43-HA (unnormalized) expression level.

Finally, we note that our single-cell RNA-seq analysis included a filtering step to discard empty droplets, cell debris and potentially dying cells (please see reply to point #5, referee #2, below). The cells retained after this quality control have a minimum gene expression complexity in terms of number of genes detected, and total number of reads per cell. This step is standard and required for downstream analysis. Therefore, we might be specifically selecting against apoptotic, dead or dying (potentially some of the TDP-43-HA-expressing) cells, by filtering them out from the analysis.

Experimental response: Our new data shows that when iNet neurons are transduced by lentiviral vectors expressing either TDP-43 under the control of its full length *TARDBP* 3'UTR⁶ or HA-FUS, there is no neuronal loss over time when compared to both TRE- or EF1 α -driven expression of TDP-43-HA (new Fig. 5c).

Rebuttal figure 5. Changes in cell-type specific abundances. Cell composition of DOX ON 4W vs 2W was quantified and plotted to reveal continuous differences across time. Positive loadings and low adjusted p-values point to increased cell abundances; whereas negative loadings and low adjusted p-values indicate decreased cell abundances.

4. The observation that TDP-43^{p403/404}-positive inclusions emerge and amplify over time (in neurons lacking TDP-43-HA) is a compelling finding (see extended Data figure 8c,d) but also somewhat puzzling. Did this occur in cells other than neurons? Did it occur also in cultures where no TDP-43-HA overexpression was done or is it somehow related to the transgenic overexpression and spread of pathology? It would be nice if the authors could follow up and clarify this matter which is potentially very important in a disease pathology (spread) scenario.

Our response: We absolutely agree with the referee that the emergence and amplification of the TDP-43^{p403/404}-positive inclusions (in neurons lacking TDP-43-HA) is both a compelling and puzzling finding. This observation alone led to initiation of a completely separate project with the main objective to address the spread of TDP-43 pathology within iNets. We think that this signal may emerge due to one of the following mechanisms:

1. As a result of cellular stress due to the presence of neurons with TDP-43-HA overexpression and aggregation within the iNet culture. In this scenario, a sudden loss of synaptic input from TDP-43-HA overexpressing neurons (due to loss of transgenic TDP-43-HA neurons within iNets) or possibly another non-cell autonomous mechanism(s) triggered by loss and/or change of glial cellular identity (see **Rebuttal Figure 5**) could have initiated the elevated p403/404 phosphorylation of TDP-43.
2. Direct spread of pathological seeds i.e. aggregated and/or fragmented TDP-43-HA (**Fig. 3c**), either transsynaptically, or via naked or exosome-mediated secretion from transduced to non-transduced neurons. In this scenario, it is conceivable that we do not capture any TDP-43^{p403/404} positivity in the TDP-43-HA-expressing neurons because the signal might be too transient since the neurons die relatively rapidly. Excitingly, preliminary findings in our aforementioned project suggest that TDP-43^{p403/404}-positive inclusions indeed also form in neurons that have internalized recombinant seeds made of TDP-43 LCD fibrils. While we see extensive recruitment of endogenous TDP-43 by the internalized LCD fibrils, only partial p403/404 phosphorylation of these foci was observed at 2 weeks post seeding (later time points are currently being analyzed), suggesting that the TDP-43^{p403/404}-positive inclusions are formed by endogenous TDP-43. However, these studies are a part of a large, recently initiated, independent project, and will be the focus of another follow-up study. We feel that the work presented in this manuscript demonstrates the utility of the iCoMoNSCs and we do not want to further delay publication of the study. Most importantly, we think that there is urgency in reporting the role of NPTX2 in human disease, while showcasing the advantages in using iNets for the study of TDP-43 proteinopathy.

5. In the single cell RNAseq experiment after TDP-43-HA overexpression (2 or 4 weeks) it would be nice to clarify which neurons were lost with TDP-43 overexpression (as shown in Figure 2d), that is, the specific identity of these cells that were particularly sensitive and the possible pathology which caused death. Currently in Figure (3d-e) it is not evident what cells are missing in the disease-context.

Our response: To our understanding, unveiling the identity of the cells lost upon TDP-43 overexpression could be addressed in two ways: first, by checking the compositional changes (in cell abundances across cell identities) after TDP-43-HA overexpression, as compared to the baseline (that is, transduced, but non-induced cells (“DOX OFF”)); see **Rebuttal Figure 6**); and second, by quantifying the lentiviral transduction efficiency by cell cluster, even before induction, to potentially detect cell identities ‘favorable’ to transduction (see **Rebuttal Figure 4**). As a cautionary note, both approaches might be confounded/biased by the fact that we carried out a computational quality control to retain cells with enough transcriptional complexity (in terms of number of expressed genes and number of reads), potentially filtering out TDP-43-HA overexpressing cells undergoing cell death.

In terms of compositional changes²⁸, we statistically evaluated the changes in cell type abundances (by cell cluster) after TDP-43-HA induction. We detect a significant increase of maturing/apoptotic cells (cluster 12), but we do not detect any targeted decrease of any cluster, except for glia/astrocytes (cluster 15). Hence, we are not able to detect any ‘missing’ neuronal cluster of particular sensitivity to the TDP-43-HA OE (**Rebuttal Figure 6**). However, as mentioned in the previous response, when checking the continuous decrease of TDP-43-HA expressing cells in DOX ON 4W vs 2w, we detected multiple changes in cell-type specific abundances (cell composition) aside from the depletion of cluster 12. Namely, we detect a

significant enrichment of maturing excitatory neurons (clusters 11, 4, 9) and a decrease in maturing/apoptotic (cluster 12) (**Rebuttal Figure 4**).

Rebuttal figure 6. Changes in cell-type specific abundances upon TDP-43-HA overexpression. Cell composition of DOX ON vs DOX OFF was quantified and plotted to reveal continuous differences. Positive loadings and low adjusted p-values point to increased cell abundances; whereas negative loadings and low adjusted p-values indicate decreased cell abundances.

6. The finding that NPTX2 is upregulated and accumulating in neurons that express TDP-43-HA (Figure 4b,d) is a compelling finding. However, it is not yet clear if this upregulation is related to detrimental processes in the cells or compensatory protective mechanisms. As NPTX2 is accumulating in cells that can withstand a higher level of TDP-43 than normal without degenerating for 4 weeks (as shown by the authors in Figure 2b, half the cells that harbour TDP-43-HA overexpression are lost at 4 weeks) this may in fact be beneficial.

Our response: We thank the referee for this important question regarding the consequences of NPTX2 upregulation in response to pathological changes in TDP-43. In order to address the possibility of a potential beneficial effect of NPTX2 upregulation, protecting neurons from TDP-43-HA overexpression-induced toxicity, we performed several experiments (please also see our response to point #6, reviewer #1, above).

1. We directly overexpressed NPTX2-HA in iNet neurons to understand whether this is toxic, indirectly addressing the question of its possible beneficial effect. We found that NPTX2-HA

overexpression leads to its accumulation within neuronal somata and processes in a pattern mimicking NPTX2 misaccumulation downstream of TDP-43-HA overexpression (new **Fig. 5b**). This NPTX2-HA overexpression and subsequent accumulation was toxic to iNet neurons (new **Fig. 5c**). Importantly, the direct neurotoxicity of NPTX2-HA overexpression was lower compared to TDP-43-HA overexpression (new **Fig. 5c**), which is in line with the multiple co-pathologies ongoing within the same neuron, potentially acting synergistically with STMN2, UNC13A and likely other TDP-43 mRNA targets.

2. We have overexpressed TDP-43-HA using different promoters and with or without the control of its 3'UTR and found that there is a significant correlation between TDP-43-HA levels and neuronal toxicity (**Extended Data Fig. 13a,b**).
3. Importantly, we performed rescue experiments in which we corrected NPTX2 levels during TDP-43-HA overexpression in iNets. To this end, we designed NPTX2-targeting shRNAs and co-expressed them in iNets neurons together with TDP-43-HA. We observed that silencing the upregulated NPTX2 in TDP-43-HA+ neurons leads to effective NPTX2 downregulation, resulting in a significant rescue of TDP-43-HA+ neurons (new **Fig. 5e,f**, new **Extended Data Fig. 15c-i**, note that the incompleteness of the rescue effect is likely explained by toxic effects from the dysregulated TDP-43 RNA targets, see above). Taken together, our data clearly indicate that NPTX2 upregulation and misaccumulation is detrimental to neuronal health and that correcting NPTX2 levels may present a promising avenue to alleviate TDP-43-induced neurotoxicity.

7. The authors speculate that NPTX2 malfunction in TDP-43 pathology may increase neuron vulnerability to glutamate toxicity in ALS and FTL and that the NPTX2 increase may trigger a pathological cascade leading to ALS. While this is an interesting thought it is still highly speculative. As the authors have a dynamic in vitro system where this could be studied, they should investigate longitudinally if TDP-43-HA harboring cells have increased levels of NPTX2 prior to degenerating and if this is the cause of the 50% cell loss in their system, and they should also treat the cells with ASOs against NPTX2 to see if neuronal loss can be prevented. Furthermore, they should misexpress NPTX2 alone to see if this is sufficient to induce ALS pathology in neurons without TDP-43 pathology.

Our response: We thank the referee for noting these important points. We performed an array of experiments to address this (please also see our response to points #6 and #10, reviewer #1 and point #6, reviewer #2).

1. Firstly, we checked whether TDP-43-HA overexpressing neurons have increased NPTX2 levels. To this end, we drove the overexpression of TDP-43-HA via different promoters and showed that NPTX2 accumulates in TDP-43-HA expressing neurons prior to their death (**Extended Data Fig. 13a,b**), an event that is correlated to the level of TDP-43-HA overexpression (**Extended Data Fig. 15b**) and which increases over time (**Extended Data Fig. 13a**).
2. Second, we silenced the upregulated NPTX2 via shRNA specifically in neurons that overexpressed TDP-43-HA (**Fig. 5e** and **Extended Data Fig. 15e-h**). Excitingly, we found that correcting just the levels of NPTX2 significantly rescued TDP-43-HA-induced neurotoxicity in

transgenic neurons within iNets (**Fig. 5f**). This result indicates that, even though the misregulation of other TDP-43 targets including STMN2 and UNC13A likely also contribute to neurotoxicity, correcting just NPTX2 levels is beneficial for neurons. These data identify NPTX2 as an important modifier of TDP-43 toxicity in human neurons. Given the known limitations of shRNA-mediated gene silencing¹⁷, including off-target toxicity, we find this consistent and significant rescue of iNets neurons particularly encouraging. Future efforts will focus on testing alternative regimens for correcting NPTX2 in models of TDP-43 proteinopathy, including antisense oligonucleotides.

3. Third, to identify whether direct overexpression of NPTX2 leads to neuronal toxicity, we overexpressed NPTX2-HA in iNets and found that this leads to similar pattern of accumulation of the protein within neurons in soma and neuronal processes, as we observed downstream of TDP-43 overexpression and this is neurotoxic in iNets (**Fig. 5a-c** and **Extended Data Fig. 15a**). We compared the resulting neuronal loss to TDP-43-HA overexpression, with variable levels, achieved with differential promoter or regulatory control. We also used iNets to overexpress HA-FUS, which we found inert under the same experimental conditions and timeframe. Importantly, this direct NPTX2 toxicity was milder compared to TDP-43-triggered toxicity (**Fig. 5c**), which we found to correlate with the levels of TDP-43 overexpression (**Extended Data Fig. 15b**). This finding suggested that NPTX2 is one of several mediators of TDP-43 toxicity, and is potentially acting synergistically with STMN2, UNC13A (and likely other TDP-43 targets) in disease.

Minor comment:

Text on top of cluster in Figure Extended Data Figure 1m is cut a bit (NR2F1)

Our response: We thank the reviewer for noticing and have corrected this in the revised manuscript.

Rebuttal Letter References

1. Baxi, E. G. *et al.* Answer ALS, a large-scale resource for sporadic and familial ALS combining clinical and multi-omics data from induced pluripotent cell lines. *Nat. Neurosci.* **25**, 226–237 (2022).
2. Ramos, D. M., Skarnes, W. C., Singleton, A. B., Cookson, M. R. & Ward, M. E. Tackling neurodegenerative diseases with genomic engineering: A new stem cell initiative from the NIH. *Neuron* **109**, 1080–1083 (2021).
3. Pantazis, C. B. *et al.* A reference human induced pluripotent stem cell line for large-scale collaborative studies. *Cell Stem Cell* **29**, 1685-1702.e22 (2022).
4. Ronchi, S. *et al.* Electrophysiological Phenotype Characterization of Human iPSC-Derived Neuronal Cell Lines by Means of High-Density Microelectrode Arrays. *Advanced Biology* **5**, e2000223 (2021).
5. Trujillo, C. A. *et al.* Complex Oscillatory Waves Emerging from Cortical Organoids Model Early Human Brain Network Development. *Cell Stem Cell* **25**, 558-569.e7 (2019).
6. Polymenidou, M. *et al.* Long pre-mRNA depletion and RNA missplicing contribute to neuronal vulnerability from loss of TDP-43. *Nat. Neurosci.* **14**, 459–468 (2011).
7. Liu, E. Y. *et al.* Loss of Nuclear TDP-43 Is Associated with Decondensation of LINE Retrotransposons. *Cell Rep.* **27**, 1409-1421.e6 (2019).
8. Klim, J. R. *et al.* ALS-implicated protein TDP-43 sustains levels of STMN2, a mediator of motor neuron growth and repair. *Nat. Neurosci.* **22**, 167–179 (2019).

9. Held, A. *et al.* iPSC Motor Neurons with Familial ALS Mutations Capture Gene Expression Changes in Postmortem Sporadic ALS Motor Neurons. *BioRxiv* (2022) doi:10.1101/2022.10.25.513780.
10. Carmen-Orozco, R. P. *et al.* Elevated nuclear TDP-43 induces constitutive exon skipping. *BioRxiv* (2023) doi:10.1101/2023.05.11.540291.
11. Fratta, P. *et al.* Mice with endogenous TDP-43 mutations exhibit gain of splicing function and characteristics of amyotrophic lateral sclerosis. *EMBO J.* **37**, (2018).
12. James, B. D. *et al.* TDP-43 stage, mixed pathologies, and clinical Alzheimer's-type dementia. *Brain* **139**, 2983–2993 (2016).
13. Higashi, S. *et al.* Concurrence of TDP-43, tau and alpha-synuclein pathology in brains of Alzheimer's disease and dementia with Lewy bodies. *Brain Res.* **1184**, 284–294 (2007).
14. Montalbano, M. *et al.* TDP-43 and Tau Oligomers in Alzheimer's Disease, Amyotrophic Lateral Sclerosis, and Frontotemporal Dementia. *Neurobiol. Dis.* **146**, 105130 (2020).
15. Nelson, P. T. *et al.* Limbic-predominant age-related TDP-43 encephalopathy (LATE): consensus working group report. *Brain* **142**, 1503–1527 (2019).
16. Jo, M. *et al.* The role of TDP-43 propagation in neurodegenerative diseases: integrating insights from clinical and experimental studies. *Exp. Mol. Med.* **52**, 1652–1662 (2020).
17. Goel, K. & Ploski, J. E. RISC-y Business: Limitations of Short Hairpin RNA-Mediated Gene Silencing in the Brain and a Discussion of CRISPR/Cas-Based Alternatives. *Front. Mol. Neurosci.* **15**, 914430 (2022).
18. Tollervey, J. R. *et al.* Characterizing the RNA targets and position-dependent splicing regulation by TDP-43. *Nat. Neurosci.* **14**, 452–458 (2011).
19. Lukavsky, P. J. *et al.* Molecular basis of UG-rich RNA recognition by the human splicing factor TDP-43. *Nat. Struct. Mol. Biol.* **20**, 1443–1449 (2013).
20. Pérez-Berlanga, M. *et al.* Loss of TDP-43 oligomerization or RNA binding elicits distinct aggregation patterns. *The EMBO journal* (2023) doi:10.15252/embj.2022111719.
21. van der Ende, E. L. *et al.* Neuronal pentraxin 2: a synapse-derived CSF biomarker in genetic frontotemporal dementia. *J. Neurol. Neurosurg. Psychiatr.* **91**, 612–621 (2020).
22. Sogorb-Esteve, A. *et al.* Differential impairment of cerebrospinal fluid synaptic biomarkers in the genetic forms of frontotemporal dementia. *Alzheimers Res Ther* **14**, 118 (2022).
23. Gómez de San José, N. *et al.* Neuronal pentraxins as biomarkers of synaptic activity: from physiological functions to pathological changes in neurodegeneration. *J. Neural Transm.* **129**, 207–230 (2022).
24. Walker, A. K. *et al.* ALS-associated TDP-43 induces endoplasmic reticulum stress, which drives cytoplasmic TDP-43 accumulation and stress granule formation. *PLoS ONE* **8**, e81170 (2013).
25. Fujita, P. A. *et al.* The UCSC Genome Browser database: update 2011. *Nucleic Acids Res.* **39**, D876-82 (2011).
26. Christmas, M. J. *et al.* Evolutionary constraint and innovation across hundreds of placental mammals. *Science* **380**, eabn3943 (2023).
27. Trapnell, C. *et al.* The dynamics and regulators of cell fate decisions are revealed by pseudotemporal ordering of single cells. *Nat. Biotechnol.* **32**, 381–386 (2014).
28. Petukhov, V. *et al.* Case-control analysis of single-cell RNA-seq studies. *BioRxiv* (2022) doi:10.1101/2022.03.15.484475.

Reviewer Reports on the First Revision:

Referees' comments:

Referee #1 (Remarks to the Author):

The authors have extensively revised their manuscript and have added many additional new analyses, which are compelling. I think that the neuropathology analysis showing NPTX2 up regulation in a neuron w/ TDP-43 pathology but not in a nearby neuron with tau pathology is very elegant. I think that this paper will be of great interest to the field and brings NPTX2 up regulation as a new TDP-43 target gene for study.

I recommend publication of this exciting work in Nature.

I am still a bit perplexed why if the authors think that TDP-43-HA expression induces a subsequent TDP-43 loss of function phenotype, then they cannot detect the cryptic exons in STMN2 or UNC13A, like they do with TDP-43 knockdown. In the revised text they mention that perhaps they cannot detect the cryptic exons because they are rapidly degraded by nonsense mediated decay (NMD). Could the authors consider testing this directly by blocking NMD (with inhibitors or genetically)? I don't think that this is necessarily required for publication, because of the excitement and importance of NPTX2, but the authors might consider, at a minimum, adding to their discussion the possibility that TDP-43-HA expression and TDP-43 knockdown are not completely identical.

Referee #2 (Remarks to the Author):

The study by Hruska-Plochan and colleagues have been improved significantly through the revisions and the majority of my concerns have been addressed satisfactorily. However, I have a few remaining points that I would like the authors to address and/or clarify:

1. Regarding sample size in the reporting summary. It is stated that a minimum of 3 wells were used per condition/experiments for iNets. Please also clarify how many times experiments were repeated, if not evident in the figure legends.

2. The authors write in the reporting summary that all iPSCs and iCoMoNSCs were checked for karyotype. Please clarify if the karyotype was normal in all cases or if there were abnormalities and if so if these are expected to have impacted the results.

3. Please clarify if the scRNA-seq data of iNETs, presented in Figure 2 and Extended Data Figure 5, originate from two independent experiments, that is, two differentiations per time point or if the data originates from one differentiation and if with biological replicates the authors rather refer to two wells/culture dishes within a differentiation (which I would be inclined to call technical replicates).

Furthermore, if the biological replicates do represent distinct differentiations, which would be beneficial, could you please show the variability and/or lack thereof in the cell type proportions and identities generated across differentiations by plotting the two replicates in different colors in a UMAP. The reason for this is to understand how easy it is to reproduce the cell type diversity across experiments in the system.

4. Can the authors please include a sentence regarding that their cultures lack layer 5 corticospinal motor neurons and thus does not constitute a model for pure ALS processes, either in the results related to Figure 2 or in the discussion.

5. Please clarify if the error bars in Figures 3b, 4c, 4d, 5c and 5f, as well as Extended Data Figures 3e, 4f-t, 8g, 9d, 9e, 15d,f,i represent standard deviations (SD) or standard error of the mean (SEM), and include this information in the respective figure legends.

Author Rebuttals to First Revision:

Referees' comments:

Referee #1 (Remarks to the Author):

The authors have extensively revised their manuscript and have added many additional new analyses, which are compelling. I think that the neuropathology analysis showing NPTX2 up regulation in a neuron w/ TDP-43 pathology but not in a nearby neuron with tau pathology is very elegant. I think that this paper will be of great interest to the field and brings NPTX2 up regulation as a new TDP-43 target gene for study.

I recommend publication of this exciting work in Nature.

I am still a bit perplexed why if the authors think that TDP-43-HA expression induces a subsequent TDP-43 loss of function phenotype, then they cannot detect the cryptic exons in STMN2 or UNC13A, like they do with TDP-43 knockdown. In the revised text they mention that perhaps they cannot detect the cryptic exons because they are rapidly degraded by nonsense mediated decay (NMD). Could the authors consider testing this directly by blocking NMD (with inhibitors or genetically)? I don't think that this is necessarily required for publication, because of the excitement and importance of NPTX2, but the authors might consider, at a minimum, adding to their discussion the possibility that TDP-43-HA expression and TDP-43 knockdown are not completely identical.

Our response: We thank the referee for the enthusiastic assessment of our revised manuscript and his/her insightful comment. Indeed, blocking NMD using specific inhibitors during TDP-43-HA overexpression is a particularly interesting experiment that we have considered. However, it will require significant optimization in terms of the experimental timeline. Therefore, we agree with the referee that it is best to not include this experiment to avoid further delaying publication of the current findings. We also agree that there is a possibility that TDP-43-HA expression and TDP-43 KD induce their similar downstream phenotypes (STMN2 and UNC13A downregulation, NPTX2 upregulation) via (partially) distinct mechanisms. To highlight this more clearly, we have added the following sentence to the discussion of our revised manuscript: *“Alternatively, the overexpression and KD paradigm may elicit their similar downstream phenotypes (NPTX2 upregulation, STMN2 and UNC13A downregulation) via distinct mechanisms.”*

Referee #2 (Remarks to the Author):

The study by Hruska-Plochan and colleagues have been improved significantly through the revisions and the majority of my concerns have been addressed satisfactorily. However, I have a few remaining points that I would like the authors to address and/or clarify:

1. Regarding sample size in the reporting summary. It is stated that a minimum of 3 wells were used per condition/experiments for iNets. Please also clarify how many times experiments were repeated, if not evident in the figure legends.

Our response: We agree with the reviewer that replicate numbers of the different datasets could be stated more clearly. To this end, we have now added this information to all the legends of the main figures and extended data figures.

2. The authors write in the reporting summary that all iPSCs and iCoMoNSCs were checked for karyotype. Please clarify if the karyotype was normal in all cases or if there were abnormalities and if so if these are expected to have impacted the results.

Our response: We thank the referee for bringing up this important point. Indeed, the karyotype of fibroblasts, iPSCs and iCoMoNSCs was checked and found to be apparently normal. We have now clarified this in the revised manuscript at the following locations:

1. Main text section, in the sentence: *“iCoMoNSCs were stable across at least 24 passages retaining their characteristic radial morphology in cell clusters and apparently normal karyotype, as well as expression of NSC-specific markers (Fig. 1a, Extended Data Fig. 1b-f).”*
2. Methods section, in the sentence: *“iCoMoNSCs clone 10/80 with apparently normal karyotype was used in the study.”*
3. Supplementary methods section, in the sentences: *“iPSCs were generated from control human early neonatal dermal fibroblasts (Gibco # C0045C) with an apparently normal karyotype (checked at Cell Guidance Systems, according to CellGS fixed sample protocol) via episomal reprogramming using plasmids coding for Oct3/4, Sox2, Klf4 and p53-shRNA^{1,2}” and “Upon further expansion to P12-15, cells were prepared for karyotype check at Cell Guidance Systems (according to CellGS fixed sample protocol) and based on their apparently normal karyotyping results, NiPS 10 was selected for further experiments.”*

Additionally, for the referee’s reference, we include here the karyotype results of the used source fibroblast and derived iPSCs and iCoMoNSCs used in the manuscript (**Rebuttal Figure 1**).

3. Please clarify if the scRNA-seq data of iNETs, presented in Figure 2 and Extended Data Figure 5, originate from two independent experiments, that is, two differentiations per time point or if the data originates from one differentiation and if with biological replicates the authors rather refer to two wells/culture dishes within a differentiation (which I would be inclined to call technical replicates).

Furthermore, if the biological replicates do represent distinct differentiations, which would be beneficial, could you please show the variability and/or lack thereof in the cell type proportions and identities generated across differentiations by plotting the two replicates in different colors in a UMAP. The reason for this is to understand how easy it is to reproduce the cell type diversity across experiments in the system.

Our response: We thank the reviewer for pointing out this important point. We agree that replicates from the same differentiation do not represent true biological replicates and we have corrected this throughout our manuscript. We have also clarified in the legend of Extended Data Fig. 5a that iNet replicates were derived from the same differentiation.

The high reproducibility of iNets is an important advantage of our system and to demonstrate this and address the reviewer's point, we have included new analyses in our updated **Extended Data Fig. 9a-c**. To showcase the replicability of iNets derived from independent differentiations we directly compared two replicates from the same differentiation (r1 and r2, from the middle stage iNets shown in Extended Data Fig. 5a and Fig. 2a) to iNets of the same age from an independent differentiation (shown in Extended Data Fig. 9a and Fig. 3d). Indeed, we found a remarkable consistency between cell types in both differentiations, pointing to the presence of similar cell identities as defined by transcription profiles. To demonstrate this, we have added a joint, unintegrated cell embedding from both differentiations coloring cells by their origin (new **Extended Data Figure 9b**). The good mixing in this UMAP points to low variability in cell identities across experiments. To further quantitate the overlap between the experiments, we have used MetaNeighbor v1.8.0 (<https://www.ncbi.nlm.nih.gov/pmc/articles/PMC5830442/>) to measure cell-type replication across experiments using the annotated clusters. This revealed high pairwise cluster replicability scores for equivalent cell identities across experiments (new **Extended Data Figure 9c**). We describe these results in our revised manuscript as follows: *"We identified 17 clusters (Fig. 3d) with a very similar cell type distribution (apoptotic, glial/astrocytic, maturing inhibitory and excitatory neurons) to our non-transduced middle-aged samples (Fig. 2a,b), pointing to the reproducibility of cell identities across independent iNets, as shown by the mixing of cells from different experiments when plotted on the same UMAP, and by the high pairwise cluster replicability scores for equivalent cell identities across experiments (Extended Data Fig. 9a-c)."*

4. Can the authors please include a sentence regarding that their cultures lack layer 5 corticospinal motor neurons and thus does not constitute a model for pure ALS processes, either in the results related to Figure 2 or in the discussion.

Our response: We apologize for overlooking this in the first revision, and we agree that this is an important point. To this end, we have added the following sentence to the Main text in the section

“Transcriptional maturation of iNets renders them similar to brain organoids”, first paragraph, last sentence: *“Of note, the apparent lack of layer 5 corticospinal motor neurons in iNets indicates that our model does not constitute a pure ALS model”*.

5. Please clarify if the error bars in Figures 3b, 4c, 4d, 5c and 5f, as well as Extended Data Figures 3e, 4f-t, 8g, 9d, 9e, 15d,f,i represent standard deviations (SD) or standard error of the mean (SEM), and include this information in the respective figure legends.

Our response: We agree with the reviewer that this information was missing from some of the Figure legends. Therefore, we have now updated all Figure and Extended Data Figure legends to include information on usage of SD/SEM for each panel containing a graph.